# An EGFR/HER2-targeted conjugate sensitizes gemcitabine-sensitive and resistant pancreatic cancer through different SMAD4-mediated mechanisms

Hongjuan Yao[1], Wenping Song[1,2], Rui Cao[1,3], Cheng Ye[1,4], Li Zhang[1], Hebing Chen [5], Junting Wang[5], Yuchen Shi[6], Rui Li[1], Yi Li[1], Xiujun Liu[1], Xiaofei Zhou[1], Rongguang Shao[1] ✉ & Liang Li [1] ✉

Chemoresistance limits its clinical implementation for pancreatic ductal adenocarcinoma (PDAC). We previously generated an EGFR/HER2 targeted conjugate, dual-targeting ligand-based lidamycin (DTLL), which shows a highly potent antitumor effect. To overcome chemoresistance in PDAC, we aim to study DTLL efficacy when combined with gemcitabine and explore its mechanisms of action. DTLL in combination with gemcitabine show a superior inhibitory effect on the growth of gemcitabine-resistant/sensitive tumors. DTLL sensitizes gemcitabine efficacy via distinct action mechanisms mediated by mothers against decapentaplegic homolog 4 (SMAD4). It not only prevents neoplastic proliferation via ATK/mTOR blockade and NF-κB impaired function in SMAD4-sufficient PDACs, but also restores SMAD4 bioactivity to trigger downstream NF-κB-regulated signaling in SMAD4-deficient tumors and to overcome chemoresistance. DTLL seems to act as a SMAD4 module that normalizes its function in PDAC, having a synergistic effect in combination with gemcitabine. Our findings provide insight into a rational SMAD4-directed precision therapy in PDAC.

Pancreatic ductal adenocarcinoma (PDAC) is the fourth most common cause of cancer-related death and is the most aggressive cancer with a five-year survival of <5%[1,2]. Gemcitabine is the first-line treatment for PDAC[3] but offers little therapeutic value because chemoresistance develops rapidly and PDAC has profound symptomatic effects[4]. Moreover, the newly developed PD-1/PD-L1 immunotherapies even showed a poor effect on pancreatic cancer[5]. Therefore, additional therapeutic strategies for pancreatic cancer are urgently needed.

Typical genetic alterations observed in PDAC include activation of the oncogene KRAS (>90%), inactivation or loss of tumor suppressor genes such as CDKN2A (95%), TP53 (50–75%), SMAD4 (~55%), PTEN (~60%) and mutation of the DNA repair gene BRCA2 (7–19%), contributing to tumor progression or relapse[6–9]. Unfortunately, few of these drivers are currently druggable, thus making it difficult to devise effective therapies against PDAC. Increasingly, molecular profiling of tumor specimens is being utilized to reveal tumor susceptibilities and

[1]Key Laboratory of Antibiotic Bioengineering of National Health and Family Planning Commission (NHFPC), Institute of Medicinal Biotechnology (IMB), Chinese Academy of Medical Sciences and Peking Union Medical College (CAMS & PUMC), NO.1 TiantanXili, Beijing 100050, P.R. China. [2]Department of Pharmacy, Affiliated Cancer Hospital of Zhengzhou University, Henan Cancer Hospital, No.127 Dongming Road, Zhengzhou 450008, China. [3]Academy of Life Science, North China University of Science and Technology, Tangshan 063210, P. R. China. [4]Tianjin Municipal Health Commission, Tianjin 300000, P. R. China. [5]Beijing Institute of Radiation Medicine, Beijing 100850, P.R. China. [6]Dongzhimen Hospital, Beijing University of Chinese Medicine, No.5 Haiyuncang, Beijing 100700, China. ✉e-mail: shaor@imb.pumc.edu.cn; liliang@imb.pumc.edu.cn

accelerate the development of precision medicine. A personalized therapeutic approach targeting key drivers associated with PDAC is likely to be a future trend. In addition, recent studies have demonstrated that these four tumor drivers play roles in gemcitabine susceptibility to pancreatic adenocarcinoma cells[10–13]. A recent study showed that a therapeutic antibody targeting KRAS synergistically increased the antitumor activity of gemcitabine by inhibiting RAS downstream signaling in pancreatic cancer with KRAS mutation[10]. The loss or mutation of TP53 promoted gemcitabine resistance in PDAC[11,12]. P16/CDKN2A inactivated pancreatic cancer cells are 3–4 fold less sensitive to gemcitabine[13]. DPC4/SMAD4 inactivation was modestly less sensitive to gemcitabine[13]. A previous study using mouse models suggested that co-deletion of PTEN and SMAD4 contributes to and mediates resistance to pancreatic cancer[14]. In addition, SMAD4 loss in pancreatic cancer causes alterations to multiple kinase pathways (particularly the phosphorylated ERK/p38/Akt pathways), and increases chemoresistance in vitro[15]. Furthermore, PDAC cells with intact SMAD4 are more sensitive to TGF-β1 inhibitor treatment to reduce cell migration, whereas PDAC cells lacking SMAD4 showed decreased cell motility in response to EGFR inhibitor treatment in PDAC carcinoma tissue[16,17]. Those evidences support that it is necessary and convincing to develop more treatment strategies for PDAC carrying fatal mutations in driver genes.

Antibody-based or ligand-based molecularly targeted drugs including antibody-drug conjugates (ADCs) and fusion proteins have recently become effective drug delivery systems[18]. To date, several EGFR-targeted therapies are available for PDAC treatment[19–23]. In our previous study, DTLL (named as a dual-targeting ligand-based lidamycin) was shown to be a highly potent bispecific antibody-drug conjugate (ADC)-like agent consisting of two oligopeptides against EGFR and HER2, and an enediyne antibiotics lidamycin (LDM)[16]. It was designed not only to inhibit the activities of both EGFR and HER2, but also to combine with the cytotoxic effect of LDM. Our previous studies demonstrated that DTLL was superior to free LDM alone in ovarian carcinoma[24], PDAC[25], and esophageal cancer[26]. Furthermore, DTLL might suppress pancreatic tumor progression by EGFR/HER2-dependent blockage of AKT/mTOR-signaling and PD-L1/PD1-mediated escape from immunosurveillance in PDAC[25].

We previously found that human pancreatic cancer AsPC-1 and MIA PaCa-2 cells with high EGFR/HER2 expression showed the strongest binding affinity to DTLP, the precursor of DTLL, however, the AsPC-1 cell line or xenograft tumor showed intermediate resistance to DTLL, whereas MIA PaCa-2 had stronger affinity and better response[25]. This raised an interesting proposal and encouraged us to explore more effective strategies and underlie relevant mechanisms especially for drug-resistant pancreatic cancers. Other studies have also found the emergence of resistance to EGFR-targeted therapies in PDAC[27,28]. To overcome that, several combination strategies of either targeted agents or checkpoint inhibitors and chemotherapies showed synergistic activities, highlighting the potential of these strategies for pancreatic cancer[29]. For instance, a kinase inhibitor specific to EGFR can sensitize tumors to gemcitabine in PDAC models[30]. Therefore, we hypothesized that it might be possible to improve the chemotherapeutic efficacy of gemcitabine in PDAC when combined with DTLL.

In the present study, we investigated differences in the genetic and expression profiles of the main driver genes in PDAC between the AsPC-1 and MIA PaCa-2 cell lines, followed with evaluation of DTLL efficacy when given in combination with gemcitabine by using gemcitabine-sensitive and gemcitabine-resistant PDAC models. The study aimed to evaluate the effects of the combination treatment and further elucidate the mechanism of this strategy. Our findings demonstrated that DTLL provided a promising synergistic therapeutic strategy but enhanced the susceptibility of both gemcitabine-sensitive and gemcitabine-resistant PDAC tumors through different signaling pathways.

## Results

### Differences in gemcitabine susceptibility and driver profiles in PDAC cells

To stratify different drug responses to gemcitabine of common PDAC cell lines available in the lab and select cell models suitable for our follow-up experiments, we first performed MTS assays using 8 human lines including AsPC-1, MIA PaCa-2, BxPC-3, PANC-1, CFPAC-1, Panc0403, HuPT-3, and SU86.86 cells. As shown in Fig. 1a, these lines showed wide variations in gemcitabine susceptibility. Among them, MIA PaCa-2 and HupT-3 cells showed the most sensitivity to gemcitabine, whereas AsPC-1 and BxPC-3 cells were apparently less responsive to the drug. The viability of AsPC-1 cells after gemcitabine treatment was still over 45% even at 100 μM, implying the strongest resistance to the drug.

Previous studies have demonstrated that KRAS, TP53, CDKN2A, and SMAD4 play key roles in the tumorigenesis and progression of PDAC, as well as gemcitabine susceptibility[10–13]. Therefore, we further detected the expression of these four drivers in these eight cell lines. As shown in Fig. 1b, a lack of KRAS expression was detected in all lines except for CFPAC-1 and Panc0403 cells, and CDKN2A was expressed very little among all eight cell lines. There were no apparent differences in KRAS and CDKN2A expression among the gemcitabine-sensitive or gemcitabine-resistant cell lines. Higher levels of TP53 expression were tested in MIA PaCa-2, BxPC-3, and PANC-1 cells, whereas low expression was observed in other lines, suggesting that the differences in TP53 expression were not predominantly devoted to gemcitabine susceptibility. For instance, HupT-3 cells presented lower levels of TP53 but were sensitive, similar to MIA-PaCa-2 cells, whereas BxPC-3 cells with higher TP53 expression showed less sensitivity than MIA-PaCa-2 cells. Interestingly, SMAD4 was highly expressed in MIA PaCa-2, HupT-3 and PANC-1 cells, whereas little was tested in AsPC-1, BxPC-3, and CFPAC-1 cells, implying that the expression might be potentially related to their susceptibility to gemcitabine. Although very significantly different levels of both TP53 and SMAD4 were observed in MIA PaCa-2 and AsPC-1 cells (Fig. 1b), the strongest expression of TP53 protein in MIA PaCa-2 cells might be attributed to the gain of function (GOF) mutant TP53 proteins and degradation of normally labile protein in MIA-paca-2 cells[31]. Subsequently, we detected the genetic status of these four drivers in eight cell lines (Table 1), and found that the *SMAD4* gene was mutated only in AsPC-1 (R100T mutation) and BxPC-3 (Del) cells. This matched the protein levels in these two lines, as well as the cellular resistance to gemcitabine. Therefore, according to the protein levels and genotypes of those four drivers between MIA PaCa-2/HupT-3 and AsPC-1/BxPC-3 cells, SMAD4 might be the predominant driver contributing to the gemcitabine response in PDAC.

For follow-up studies, we selected AsPC-1 and MIA PaCa-2 cells to detect the expression of numerous proteins including EGFR, HER2, SMAD2, SMAD3, SMAD4, SMAD7, and TGF-β by Western blot analysis. As confirmed in Fig. 1c, SMAD4 protein was highly expressed in MIA PaCa-2 but not AsPC-1 cells. Specifically, the ratios of p-SMAD2/SMAD2, p-SMAD3/SMAD3, SMAD7, p-EGFR/EGFR and p-HER2/HER2 in AsPC-1 cells were obviously higher than those in MIA PaCa-2 cells, showing much more activated TGF-β/SMADs signaling. Consequently, AsPC-1 and MIA PaCa-2 cells were chosen as in vitro models because of their resistant and sensitive responses to gemcitabine with deficient and sufficient SMAD4 expression, as well as mutant/wild-type genetic status, respectively, to further test the hypothesis that SMAD4 might play a key role in chemoresistance of pancreatic cancer. Simultaneously, we observed the antiproliferative effects of gemcitabine and DTLL on the AsPC-1 and MIA PaCa-2 cell lines. As shown in Fig. 1d, AsPC-1 cells were more resistant to gemcitabine than MIA PaCa-2 cells, or DTLL, which was mentioned in our previous study[17].

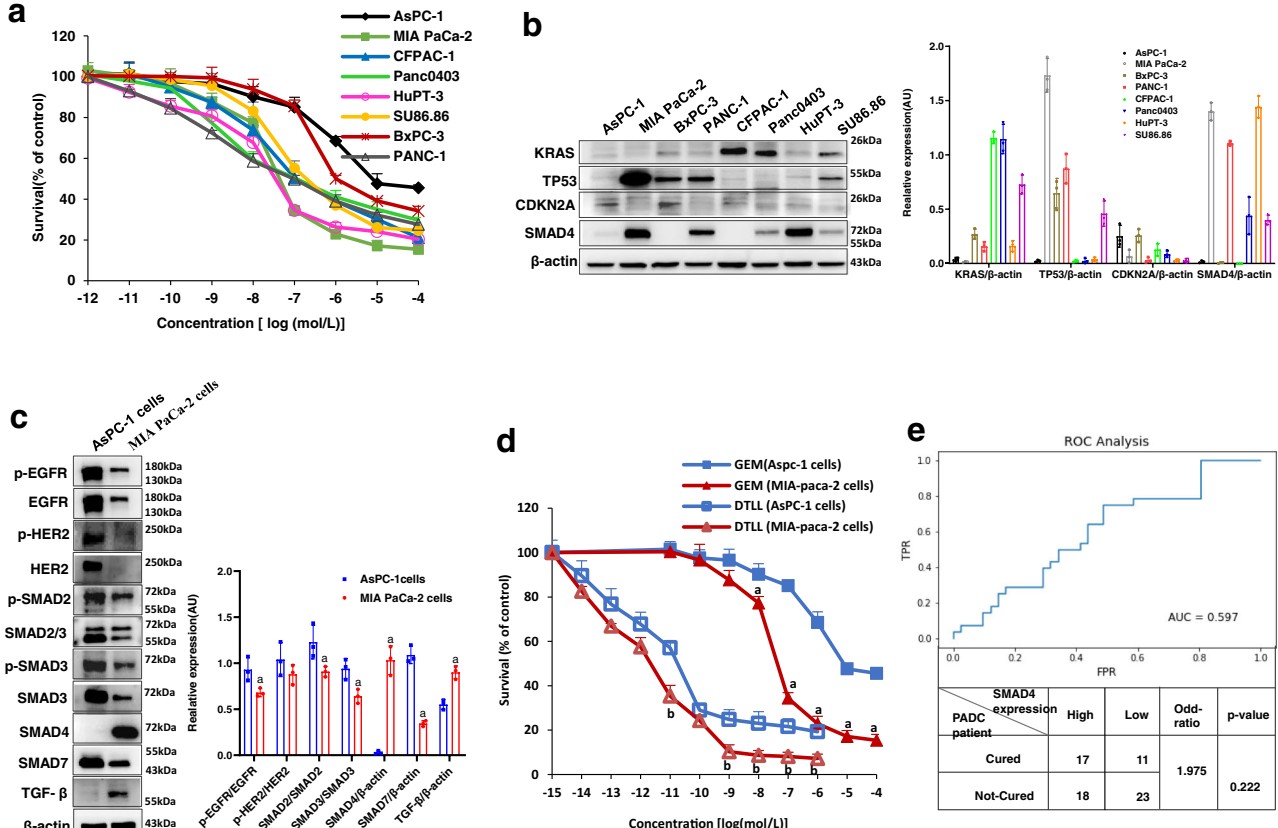

**Fig. 1 | Differences in drug susceptibility and PDAC drivers between AsPC-1 cells and MIA PaCa-2 cells. a** Drug responses to gemcitabine of eight human pancreatic cancer cell lines. AsPC-1, MIA PaCa-2, BxPC-3, PANC-1, CFPAC-1, Panc0403, HuPT-3, and SU86.86 cells were exposed to various concentrations of gemcitabine (GEM) for 72 h and cell viability was measured by the MTS assay. Data are shown as the mean ± SD from three biologically independent experiments ($n = 3$). **b** The protein expression of KRAS, TP53, CDKN2A, and SMAD4 in eight pancreatic cancer cell lines was analyzed by Western blot assay, and quantitative evaluation for each line was performed by using ImageJ software. The protein level of β-actin was used as the internal standard. Data are presented as the mean ± SD ($n = 3$ biologically independent experiments). **c** Differences in protein expression levels in AsPC-1 and MIA PaCa-2 cells. The protein expression of EGFR, HER2, SMAD2, SMAD3, SMAD4, SMAD7, and TGF-β was detected by Western blot analysis. Data are shown as the mean ± SD ($n = 3$ three biologically independent samples). 'a' indicates $p < 0.05$ as compared with AsPC-1 cells. $p < 0.01$ for TGF-β, $p < 0.001$ for SMAD4 and SMAD7. **d** Drug responses to either gemcitabine or DTLL between AsPC-1 and MIA PaCa-2 cells. AsPC-1 and MIA PaCa-2 cells were exposed to various concentrations of GEM

or DTLL for 72 h and cell viability was measured by the MTS assay. The results were obtained from three biologically independent experiments. Data are presented as the mean ± SD ($n = 3$). 'a' indicates $p < 0.05$ as compared with GEM (AsPC-1 cells) and 'b' represents $p < 0.05$ compared with DTLL (AsPC-1 cells). $p < 0.001$ for comparison with GEM at the concentrations of $10^{-7}$, $10^{-6}$, $10^{-5}$ and $10^{-4}$ mol/L, and $p < 0.01$ for comparison with GEM at the concentrations of $10^{-11}$, $10^{-9}$, $10^{-7}$ and $10^{-6}$ mol/L. **e** ROC (receiver operating characteristic) analysis to determine the SMAD4 expression level as a predictive tool for clinical gemcitabine response in pancreatic cancer patients ($n = 108$) from public data (TCGA-PAAD project). An ROC curve was plotted for AUC evaluation with the threshold level for SMAD4 expression at 2.27 (median expression of all the PAAD patients) to define the high (>2.27) or low (<2.27) level, followed with Fisher's exact test for calculation of odds ratio and p-value. Note: For **c** and **d**, statistical significance was determined by using two-sided paired-samples t-test. All specific p values are presented in the Source Data file and only significant values are shown in figures. Source data are provided in the Source Data file.

Furthermore, we downloaded the public data (TCGA-PAAD project) of pancreatic cancer patients from TCGA including clinical information on gemcitabine treatment and SMAD4 expression datasets to determine if SMAD4 expression in a PDAC patient is clinically correlated with gemcitabine response. By using the expression level of SMD4 as the criterion for predicting whether a patient is cured, we plotted an ROC curve with an AUC equal to 0.597 (Fig. 1e). There were 35 patients with high SMAD4 expression, of which 17 (48.57%) samples had a positive drug response, whereas only 34.38% of patients (11 from 32 samples) with a low level of SMAD4 responded positively to gemcitabine. The odds ratio was 1.98 ($p = 0.22$), which indicated that the high level of SMAD4 expression facilitated the gemcitabine response clinically, although the significance level was more than 0.05 perhaps owing to the small sample size.

Therefore, we demonstrated that SMAD4 was associated with not only PDAC occurrence and progression, but also the drug response to gemcitabine treatment both experimentally and clinically.

## DTLL potentiated the inhibitory effects of gemcitabine on cell proliferation and cycle distribution

To investigate whether the combination of the two drugs could impact PDAC cell proliferation, we treated gemcitabine-resistant/SMAD4-deficient AsPC-1 and gemcitabine–sensitive/SMAD4-sufficient MIA PaCa-2 cells with gemcitabine, DTLL and both. The results from both MTS (Fig. 2a) and CyQUANT (Fig. 2b) assays indicated that the combination of gemcitabine with DTLL obviously inhibited AsPC-1 cell growth compared to gemcitabine or DTLL alone. Compared with AsPC-1 cells, gemcitabine alone showed a stronger inhibitory effect in MIA PaCa-2 cells. The combination treatment revealed even more significant inhibition of cell growth in a concentration-dependent manner. The results in both lines showed that the combination treatment had synergistic effects evaluated by the combination index (CI) < 1 (the threshold line) for both (right panels in Fig. 2a, b), indicating that DTLL effectively sensitized the cell response to gemcitabine.

**Table 1 | Genotypic profiles of four drivers in different PDAC cell lines**

| Gene name | Genotype | Cell line name | Exon | Mutation location |
|---|---|---|---|---|
| KRAS | WT | Bxpc-3 | | |
| | | Panc0403 | | |
| | Mut | Su86.86 | exon 2 | G12D |
| | | PANC-1 | | G12D |
| | | Aspc-1 | | G12D |
| | | CF-PANC1 | | G12V |
| | | MIA PaCa-2 | | G12C |
| | | HupT-3 | | G12R |
| TP53 | WT | Aspc-1 | | |
| | | Panc0403 | | |
| | Mut | Su86.86 | exon 6, 9, 10 | G201V,G228V,G321V,G360V,G602,G683T,G962T,G1079T |
| | | PANC-1 | exon 4, 7, 8 | R114H,R141H,R234H,R273H,G341A,G422A,G701A,G818A |
| | | CF-PANC1 | exon 3, 6, 7 | C83R,C110R,C203R,C242R,T247C,T328C,T607C,T724C |
| | | MIA PaCa-2 | exon 3, 6, 7 | R89W,R116W,R209W,R248W,C265T,C346T,C625T,C742T |
| | | HupT-3 | exon 4, 8 | R123W,R150W,R243W,R282W,C367T,C448T,C727T,C844T |
| | | Bxpc-3 | exon 2, 5, 6 | Y61C,Y88C,Y181C,A182G,Y220C,A263G,A542G,A659G |
| CDKN2A | WT | Su86.86 | | |
| | | Bxpc-3 | | |
| | | PANC-1 | | |
| | | Aspc-1 | | |
| | | CF-PANC1 | | |
| | | MIA PaCa-2 | | |
| | | HupT-3 | | |
| | Mut | PANO403 | exon 2 | R99C,R96G,R90C,S73R |
| SMAD4 | WT | Su86.86 | | |
| | | PANC-1 | | |
| | | CF-PANC1 | | |
| | | Panc0403 | | |
| | | MIA PaCa-2 | | |
| | | HupT-3 | | |
| | Mut | Aspc-1 | exon 3 | R100T |
| | | Bxpc-3 | exon 5, 9 | T496C,F166L; Del,Chr18:48584501-48584728:51-178 |

*WT* wild type, *Mut* mutation.

To further determine whether DTLL was capable of sensitizing cells to other drugs, we selected multiple antineoplastic agents widely used for pancreatic cancer treatment to detect the effect of DTLL combined with 5-fluorouracil (antimetabolite), oxaliplatin (DNA cross-linking), paclitaxel (anti-microtubule), irinotecan (topoisomerase I inhibitor), etoposide (topoisomerase II inhibitor) and lapatinib (dual HER2 and EGFR tyrosine kinase inhibitor), followed by evaluation of the synergistic effect by combination index (CI) calculation. The results (shown in Fig. 2c) indicated that the combination of DTLL with all the above agents showed more significant antiproliferative effectiveness than either DTLL or those agents alone in both AsPC-1 and MIA PaCa-2 cells. The drug interactions of those agents and DTLL were shown to be synergistic with CIs <1 at those corresponding drug concentrations, except for each treated point of paclitaxel, 5-fluorouracil, irinotecan or oxaliplatin. The plots in both AsPC-1 and MIA PaCa-2 cells indicated that drugs at much higher doses showed less synergistic effects with CI values >1 (right panels in Fig. 2c). These results suggest that DTLL can effectively sensitize PDAC cells to different types of drugs in addition to gemcitabine.

Subsequently, we detected the cell cycle distribution in AsPC-1 and MIA PaCa-2 cells treated with PI staining. As shown in Fig. 2d, gemcitabine obviously induced the majority (73.54%) of MIA PaCa-2 cells to arrest at S phase compared to control cells at only 31.38% S phase arrest. However, slightly more AsPC-1 cells treated with gemcitabine arrested at both S (51.73% versus 41.50% in control) and G2/M (14.13% versus 4.26% in control), further suggesting that AsPC-1 cells were resistant to this drug. However, G2/M arrest in both lines was induced by DTLL. In the combined treatment group, G2/M arrest was observed in AsPC-1 cells, whereas an increase in S and G2/M phases of MIA PaCa-2 cells was detected.

The above results showed cell cycle arrest and growth inhibition at different levels between the two cell lines by treatments, suggesting the antiproliferative effects of the combination treatment on both lines.

**Effects of DTLL on different signaling pathways in AsPC-1 and MIA PaCa-2 cells**

After demonstrating the antiproliferative effects of the combination treatment, we characterized the potential mechanisms of action in AsPC-1 and MIA PaCa-2 cells. To investigate inhibitory effects on PDAC cell proliferation and phases of drug treatments, we detected the expression of the apoptotic protein p-Bcl2, the DNA damage marker γH2AX and the cell cycle-related protein Cyclin D1. As shown in Fig. 3a, DTLL significantly decreased the expression of p-Bcl2 and Cyclin D1 with increased phosphorylated γH2AX in both AsPC-1 and MIA PaCa-2 cells. Similar observations were obtained from alterations in those proteins after treatment with gemcitabine. Meanwhile, even more remarkable changes in both lines were observed when treated with

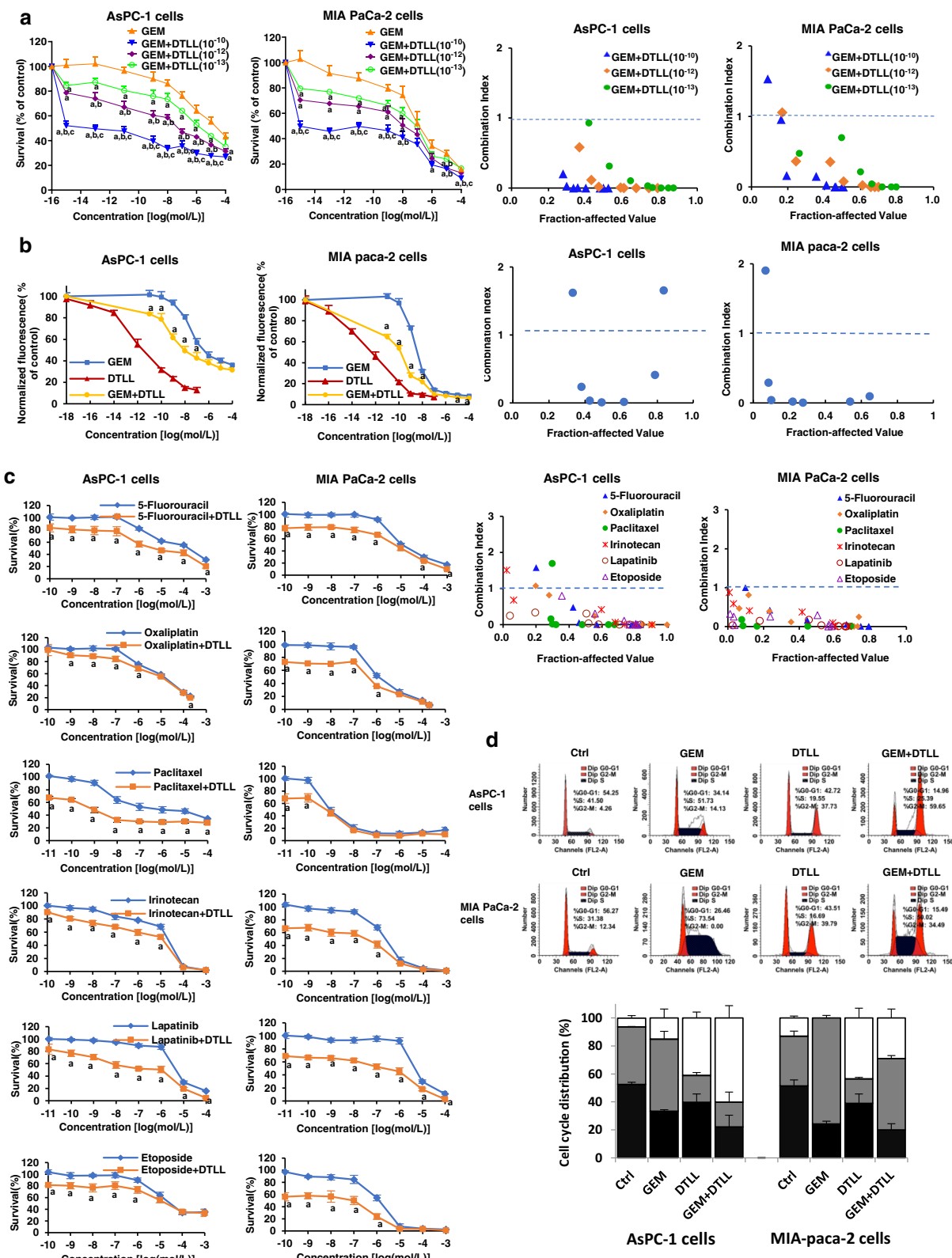

both. The results further demonstrated that the combination treatment could eventually induce cellular apoptosis, cycle arrest, and DNA double-strand breaks in gemcitabine-resistant or gemcitabine-sensitive cells.

Our previous study demonstrated that DTLL enhanced DNA damage via EGFR/HER2-dependent blockage of the AKT/mTOR and PD-L1 signaling pathways in gemcitabine-sensitive MIA PaCa-2 cells.

Hence, to test if these cellular signaling pathways are involved, we treated gemcitabine-resistant AsPC-1 or gemcitabine-sensitive MIA PaCa-2 cells with vehicle, gemcitabine, DTLL or both for 4 h, and further analyzed the ratios of active phosphorylated and total proteins for HER-2, EGFR, ERK1/2, AKT, mTOR, and PD-L1. Interestingly, the responses in the AKT/mTOR and PD-L1 signaling pathways were quite different between AsPC-1 and MIA PaCa-2 cells. After treatment with

**Fig. 2 | DTLL potentiated proliferation inhibition and cell cycle arrest induced by gemcitabine. a** Effects on drug cytotoxicity in AsPC-1 and MIA-paca-2 cells after treatment with GEM, DTLL or both were detected by MTS assays. AsPC-1 and MIA PaCa-2 cells were exposed to a series of concentrations of GEM combined with $10^{-10}$, $10^{-12}$ and $10^{-13}$ mol/L DTLL for 72 h, respectively. The MTS assay was performed at the end of incubation. Data are shown as the mean ± SD ($n = 3$ biologically independent experiments). 'a' indicates $p < 0.05$ as compared with the GEM group, 'b', $p < 0.05$, compared with the 'GEM + DTLL'($10^{-13}$) group, and'c', $p < 0.05$, compared with the 'GEM + DTLL'($10^{-12}$) group. **b** The inhibitory effect of GEM and DTLL on cell proliferation in AsPC-1 and MIA-paca-2 cells was measured based on DNA fluorescence using the CyQUANT proliferation assay. The cells were exposed to a series of concentrations of GEM, DTLL or GEM combined with $10^{-14}$ mol/L DTLL for 72 h, respectively. The assay utilizes a fluorescent dye to measure double strand DNA content with a microplate reader with excitation at 485 nm and emission detection at 530 nm. Data are shown as the mean ± SD ($n = 3$ biologically independent experiments). 'a' indicates $p < 0.05$, compared with the GEM group. **c** Synergistic effect of 5-fluorouracil, oxaliplatin, paclitaxel, irinotecan, lapatinib

and etoposide with DTLL ($10^{-12}$ mol/L) based on MTS assays. The synergistic effect of the combination treatment was evaluated by calculation of the combination index (CI) using the Chou-Talalay method. CI > 1, CI = 1 and CI < 1 indicate antagonistic, additive and synergistic effects, respectively. Data are shown as the mean ± SD (n = 3 biologically independent experiments). 'a' indicates $p < 0.05$, compared with the 5-fluorouracil, oxaliplatin, paclitaxel, irinotecan, lapatinib or etoposide group, respectively. **d** Cell cycle distribution in AsPC-1 and MIA paca-2 cells after treatment with GEM, DTLL or both was detected by flow cytometry assays. In the cell cycle assay, AsPC-1 and MIA PaCa-2 cells were treated with 0.02 μM gemcitabine, 0.1 nM DTLL or their combination for 24 h. The cell cycle distribution was evaluated using propidium iodide (PI) staining and analyzed by flow cytometry. Data are presented as the mean ± SD for biologically triplicate experiments. Note**:** One-way ANOVA with Bonferroni post hoc test was used in **a**. For **b**, **c**, statistical significance was determined by using two-sided paired-samples *t*-test. All specific *p* values are presented in the Source Data file and only significant values are shown in figures. Source data are provided in the Source Data file.

DTLL or both in MIA PaCa-2 cells, the ratios of active phosphorylated and total EGFR, HER-2, AKT and mTOR proteins were obviously decreased (Fig. 3a). Furthermore, there were significant decreases in PD-L1 expression, which confirmed our previous findings regarding to DTLL function[25] and suggested similar AKT/mTOR signaling affected by its combination treatment in MIA PaCa-2 cells. In contrast, the ratios of p-EGFR/EGFR, p-HER-2/HER-2, p-AKT/AKT and p-mTOR/mTOR were significantly increased in AsPC-1 cells. Furthermore, an apparent increase in PD-L1 expression was observed in AsPC-1 cells, indicating that there were obviously different molecular mechanisms by which these two cells differed in drug response to the combination therapy. In addition, no changes in ERK1/2 expression in either of the two cell lines were found.

We further explored whether other molecular signaling pathways were triggered by the combination treatment in AsPC-1 and MIA PaCa-2 cells. Total differences in TGF-β/SMADs signaling proteins were observed between those two lines, including SMAD2, SMAD3, SMAD4, SMAD7, and TGF-β levels. The data shown in Fig. 3b indicated that treatment with DTLL alone or in combination with gemcitabine induced the expression of SMAD2, SMAD3 or SMAD4 in SMAD4-deficient AsPC-1 cells, while a marked decrease in SMAD7 expression was observed. This result indicated that, unlike MIA PaCa-2 cells, DTLL might restore SMAD4 expression by inhibiting SMAD7 in AsPC-1 cells, leading to significant reactivation of SMAD4-dependent signaling. In contrast, DTLL alone or in combination with gemcitabine significantly decreased the expression of SMAD2, SMAD3 or SMAD4 in SMAD4-sufficient MIA PaCa-2 cells, but they markedly increased the expression of SMAD7. We speculated that immediate activation of inhibitory SMADs (SMAD7) in MIA PaCa-2 cells might form a complex with SMAD2/3, thereby interfering with the complex formation between SMAD2/3 and SMAD4[32], and preventing further signal propagation. Therefore, the above results demonstrated that the combination treatment had opposite effects on the TGF-β/SMADs signaling pathways of PDAC cells in a SMAD4-dependent manner.

Subsequently, we detected differences in the effects of DTLL on the translocation of SMAD4 induced from the cytosol to the nucleus in AsPC-1 and MIA PaCa-2 cells. The cytoplasmic and nuclear proteins of AsPC-1 and MIA PaCa-2 cells were extracted after treatments to detect the distribution of SMAD2, SMAD3 and SMAD4 in parallel Western blot assays (Fig. 3c). The results showed that the phosphorylated forms of SMAD2/SMAD3 and SMAD4 were significantly increased in the nucleus of AsPC-1 cells treated with either DTLL or both, but decreased in the cytoplasm. However, there were significant decreases in the ratios of p-SMAD2/SMAD2 and p-SMAD3/SMAD3, as well as decreased SMAD4 in the nucleus of MIA PaCa-2 cells after treatment with DTLL or both. In the MIA PaCa-2 cytoplasm, we found an increase in SMAD4 and enhanced ratios of p-SMAD2/SMAD2 and p-SMAD3/SMAD3 in cells

treated by DTLL or both. As the results from the immunofluorescence assay show in Fig. 3d, the combination treatment, similar to DTLL, significantly increased the nuclear accumulation of SMAD4 in AsPC-1 cells, but suppressed its accumulation in the MIA PaCa-2 nucleus. We demonstrated that the SMAD-dependent signaling pathway was reactivated by DTLL or combination therapy, contributing to increases in the inhibition of SMAD4-deficient AsPC-1 cell growth. However, similar drug responses but controversial alterations in this pathway were observed in SMAD4-sufficient MIA PaCa-2 cells.

## Downregulation or upregulation of SMAD4 altered PDAC susceptibility to gemcitabine

Since we have found a relationship between the gemcitabine response and SMAD4 protein levels, we altered SMAD4 expression by transiently transfecting either a specific siRNA or overexpression vector of SMAD4 into pancreatic cancer cells to detect the effect of SMAD4 expression levels on PDAC cellular susceptibility to gemcitabine. As shown in Fig. 4a, downregulation of SMAD4 at both mRNA and protein levels in MIA PaCa-2 cells that carry the wild-type *SMAD4* gene with high protein levels significantly reduced cell sensitivity to gemcitabine after exposure to various concentrations for 72 h. In contrast, SMAD4 overexpression apparently sensitized AsPC-1 cells carrying the R100T mutation with little SMAD4 protein expression to gemcitabine (Fig. 4b). The results indicated that SMAD4 contributed to the cellular response to gemcitabine in PDAC cells. However, it might be involved in distinct action mechanisms owing to different genotypes and protein levels of SMAD4 in MIA PaCa-2 versus AsPC-1 cells.

## Differences in the protein degradation of wild-type and mutant SMAD4 in PDAC cells and DTLL impact

To further test if the effect of SMAD4 genetic status on its expression, we performed qRT-PCR and Western blot assays to detect differences in *SMAD4* mRNA expression among BxPC-3 (deleted *SMAD4*), MIA PaCa-2 (wild-type *SMAD4*) and AsPC-1 (R100T mutant *SMAD4*) cells, as well as that in protein degradation. *SMAD4* mRNA and protein expression was not detectable in BxPC-3 cells, consistent with its genetic status. With obviously higher levels of *SMAD4* mRNA in MIA PaCa-2 and AsPC-1 cells, there was no significant difference between those two lines (left panel in Fig. 4c). However, SMAD4 protein expression in MIA PaCa-2 cells was much higher than that in AsPC-1 cells (right panel in Fig. 4c). The above result implied that SMAD4 genetic status is responsible for SMAD4 protein levels.

Furthermore, mutant SMAD4 protein in AsPC-1 cells had much shorter half-life of only approximately one hour, showing rapid protein degradation after 24 h of treatment with cycloheximide (CHX, a protein synthesis inhibitor), compared to the prolonged wild-type protein

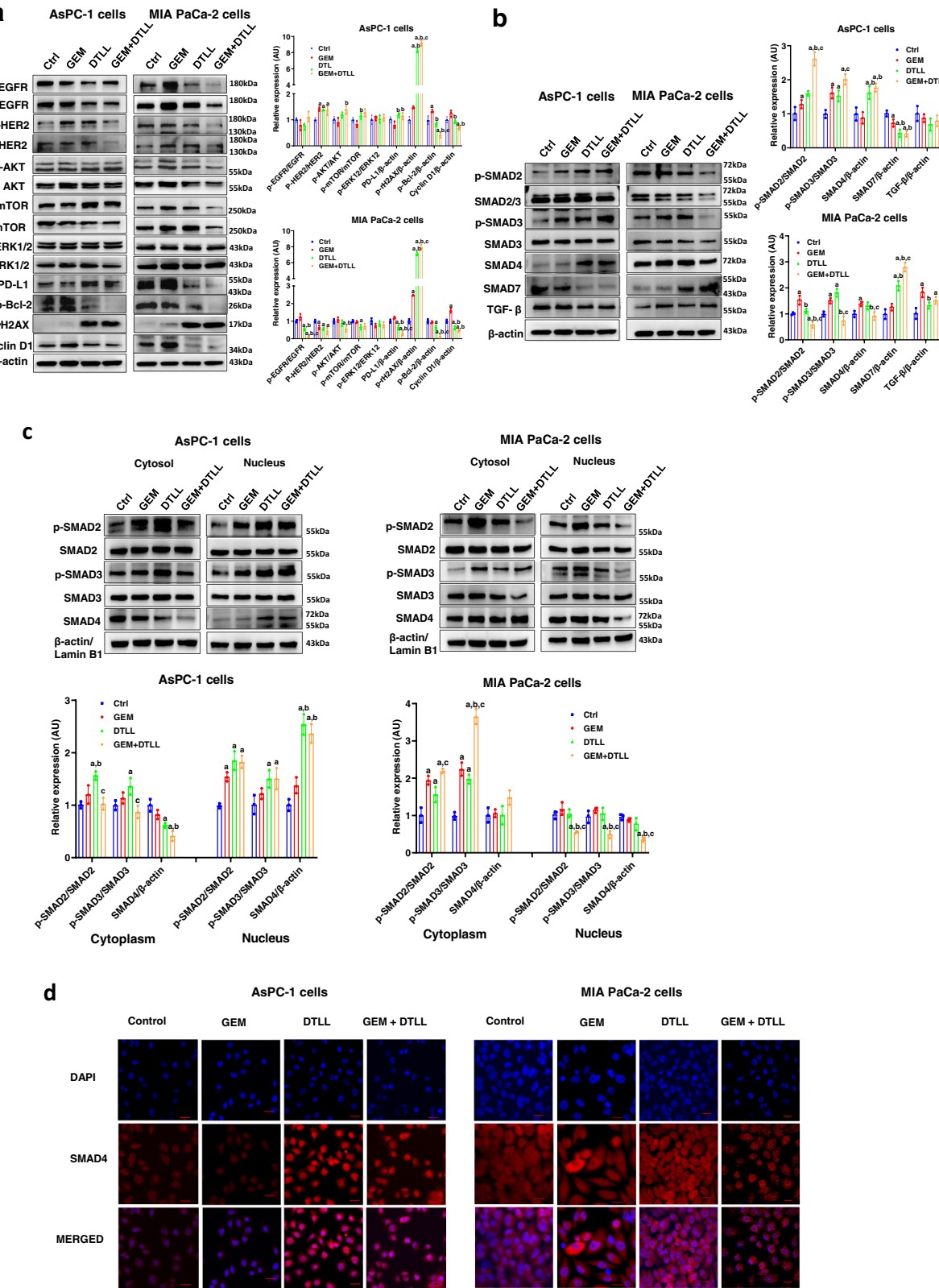

in MIA PaCa-2 cells even after 72 h of CHX treatment (Fig. 4d). Interestingly, with the induction of DTLL, mutant protein degradation of SMAD4 in AsPC-1 cells was significantly retarded, with a dramatic 9-fold-increase in its protein level even after 48 h of CHX treatment compared to the control at the beginning time point. The wild-type SMAD4 protein in MIA PaCa-2 cells also showed a slight increase with the induction of DTLL (Fig. 4e). In addition, SMAD4 protein

degradation was not observed and induced by DTLL in BxPC-3 cells (Fig. 4d, e). This suggests that DTLL might significantly enhance the expression level of SMAD4, especially in SMAD4-deficient AsPC-1 cells, and thus resensitize the cellular response to gemcitabine or other chemotherapeutic agents. In addition, BxPC-3 cells without SMAD4 mRNA and protein expression were suitable for creating wild-type and mutant SMAD4 overexpression vectors in follow-up studies.

**Fig. 3 | Functional characterization of gemcitabine, DTLL or combination therapy in AsPC-1 and MIA PaCa-2 cells.** Protein levels of EGFR/HER2-dependent signaling pathways, PD-L1 signaling pathways, apoptotic protein Bcl2, DNA damage marker γH2AX, cell cycle-related protein Cyclin D1 (a) and TGF-β/SMADs signaling pathways (b) were determined by Western blot assays in AsPC-1 and MIA PaCa-2 cells after treatments with 2 μM gemcitabine, 0.1 nM DTLL and both at 37 °C for 4 h. Data are shown as the mean ± SD ($n = 3$ biologically independent experiments). Note: For **a–c**, 'a' indicates $p < 0.05$ as compared with the control, 'b', $p < 0.05$ compared with the GEM group and 'c', $p < 0.05$ compared with the DTLL group. In AsPC-1 cells, $p \leq 0.001$ ('GEM + DTLL'versus Control) for p-γH2AX, p-Bcl-2, p-SMAD2/SMAD2, p-SMAD3/SMAD3 and SMAD7; $p \leq 0.001$ ('GEM + DTLL'versus GEM) for p-γH2AX, p-Bcl-2, p-SMAD2/SMAD2 and Cyclin D1. $p < 0.001$ ('GEM + DTLL'versus DTLL) for p-SMAD2/SMAD2; $p < 0.001$ (DTLL versus Control) for p-γH2AX and SMAD7; $p < 0.001$ (DTLL versus GEM) for p-γH2AX. In MIA PaCa-2 cells, $p \leq 0.001$ ('GEM + DTLL'versus Control) for p-EGFR/EGFR, PD-L1, p-γH2AX, p-Bcl-2, Cyclin D1 and SMAD7; $p \leq 0.001$ ('GEM + DTLL'versus GEM) for p-EGFR/EGFR, PD-L1, p-γH2AX, p-Bcl-2, Cyclin D1, p-SMAD2/SMAD2, p-SMAD3/SMAD3, and SMAD7; $p \leq 0.001$ (DTLL versus Control) for p-γH2AX, p-SMAD3/SMAD3 and SMAD7; $p \leq 0.001$ (DTLL versus GEM) for p-EGFR/EGFR, p-γH2AX and Cyclin D1; $p \leq 0.001$ (GEM versus Control) for p-γH2AX and Cyclin D1. **c** The nuclear accumulation of SMAD4 in AsPC-1 and MIA PaCa-2 cells after treatment with 2 μM gemcitabine, 0.1 nM DTLL or both at 37 °C for 4 h. After treatment, the cytoplasmic and nuclear protein fractions were detected using nuclear and cytoplasmic extraction kits. Isolation of the cytosol and nuclei was analyzed by Western blot

assay. β-actin and Lamin B1 were used as the loading controls for the cytosol and nucleus, respectively. Band intensities were quantified using ImageJ. Data are shown as the mean ± SD ($n = 3$ biologically independent experiments). In AsPC-1 cells, $p \leq 0.001$ ('GEM + DTLL'versus Control) for SMAD4 in the cytoplasm and p-SMAD2/SMAD2 and SMAD4 in the nucleus; $p \leq 0.001$ ('GEM + DTLL'versus GEM) for SMAD4 in the nucleus; $p \leq 0.001$ (DTLL versus Control) for p-SMAD2/SMAD2 and SMAD4 in the nucleus; $p \leq 0.001$ (DTLL versus GEM) for SMAD4 in the nucleus. In MIA PaCa-2 cells, $p \leq 0.001$ ('GEM + DTLL'versus Control) for p-SMAD2/SMAD2 and p-SMAD3/SMAD3 in the cytoplasm, and SMAD4 in the nucleus; $p \leq 0.001$ ('GEM + DTLL'versus GEM) for p-SMAD3/SMAD3 in the cytoplasm, and p-SMAD2/SMAD2 and SMAD4 in the nucleus; $p \leq 0.001$ ('GEM + DTLL'versus DTLL) for p-SMAD3/SMAD3 in the cytoplasm; $p \leq 0.001$ (DTLL versus Control) for p-SMAD3/SMAD3 in the cytoplasm; $p \leq 0.001$ (GEM versus Control) for p-SMAD2/SMAD2 and p-SMAD3/SMAD3 in the cytoplasm. **d** Immunofluorescence analysis of SMAD4 in AsPC-1 and MIA PaCa-2 cells after treatment with 2 μM gemcitabine, 0.1 nM DTLL or both at 37 °C for 4 h using laser scanning confocal microscopy. Cells were fixed and stained with anti-SMAD4 antibody. Following incubation with fluorescent secondary antibodies, the cells were stained with DAPI. Red denotes the anti-Smad4 (red) antibody, and blue indicates the nucleus of AsPC-1 or MIA-paca-2 cells stained with DAPI. Red scale bars indicate 20 μm. Images are representative of three biologically independent experiments. Note: All specific $p$ values are presented in the Source Data file and only significant values are shown in figures. For **a–c**, statistical significance was determined by using one-way ANOVA with Bonferroni post hoc test. Source data are provided in the Source Data file.

## Evaluation of in vivo efficacy in CDX models from AsPC-1 and MIA PaCa-2 cells

Next, we determined whether there was a difference in the effect of SMAD4 between wild-type and mutant proteins on gemcitabine efficacy in vivo, and how DTLL or its combinational therapy influenced the growth of tumors expressing these two different SMAD4. The above two AsPC-1 and MIA PaCa-2 lines, serving as SMAD4-mutant/deficient and SMAD4-wild-type/sufficient PDAC cells, were used to generate in vivo CDX mouse models for evaluation of the antineoplastic efficacy of vehicle, gemcitabine or DTLL alone, and the combination. As shown in Fig. S1a, AsPC-1 xenografts had much higher expression of p-EGFR and p-SMAD2/SMAD2, p-SMAD3/SMAD3, SMAD7 but lack of SMAD4 and TGF-β expression, as compared to the MIA PaCa-2 xenograft model, further confirming the results from the in vitro observations.

In the AsPC-1 xenograft model (shown in Fig. 5a and Supplementary Table 1), gemcitabine had a minimal inhibitory effect at 26.05% and DTLL slightly repressed tumor growth by 39.80%. However, the combination of gemcitabine and DTLL had remarkable synergistic efficacy with a significant tumor inhibition rate of 66.96% in the AsPC-1 xenograft model. As shown in Fig. 5b and Supplementary Table 1, tumor growth of MIA PaCa-2 CDX models was obviously inhibited in all treatment groups compared with the control. Specifically, gemcitabine or DTLL alone was able to apparently inhibit tumor growth of MIA PaCa-2 models by 59.04% or 79.62%, respectively. The combination treatment significantly repressed tumor growth by 88.15%. In addition, no deaths or significant changes in body weight were observed (Fig. S2a and S2b) in treated mice, and no toxic-pathological organs were observed (Fig. S3a and S3b) as compared to the control group.

After mice were sacrificed, we further detected any alterations in the protein expression of tumor tissue samples in Western blot assays to confirm our results from the above in vitro functional studies, and thereby shed light on its therapeutic efficacy. For SMAD4-sufficient MIA PaCa-2 tumor models, the ratios of p-EGFR/EGFR, p-HER-2/HER-2, p-AKT/AKT, and p-mTOR/mTOR as well as PD-L1 expression were more effectively inhibited after treatment with DTLL or both than in gemcitabine or control-treated groups, whereas slightly decreased SMAD4 and dramatically increased SMAD7 were observed. These data implied that combination treatment enhanced antineoplastic efficacy via EGFR/HER2-dependent blockade of AKT/mTOR and PD-L1 signaling pathways in MIA PaCa-2 xenografts, consistent with the above in vitro results as well as our previous findings of DTLL[25]. In contrast, in either

DTLL or the combination treatment group of SMAD4-deficient AsPC-1 models (shown in Fig. 5c), significantly increased expression of SMAD2/SMAD3 and SMAD4, as well as decreased SMAD7 were tested when compared to the control or gemcitabine, suggesting that DTLL combined with gemcitabine could obviously reactivate SMAD4, and thereby inhibit AsPC-1 tumor growth in a SMAD-dependent manner. Moreover, we detected significant increases in the ratios of active phosphorylated and total proteins for EGFR, HER-2, AKT, and PD-L1, as well as little alteration in p-mTOR/mTOR with gemcitabine or the combination treatment. In contrast to the effect on SMAD4-sufficient MIA PaCa-2 tumors, DTLL inhibited the growth of SMAD4-deficient AsPC-1 tumors but not by blocking the EGFR/AKT/mTOR and PD-L1 signaling pathways.

IHC staining of Ki-67 with semiquantification displayed a remarkable antiproliferative effect on tumor cells in the combination treatment group when compared to gemcitabine or DTLL alone (Fig. 5d), as well as greater apoptosis in the TUNEL assay (Fig. 5e). Our results suggested that the combination treatment showed highly potent efficacy by using in vivo CDX models compared to gemcitabine or DTLL alone.

## Evaluation of in vivo efficacy by PDX models of human pancreatic cancer

PDX models derived from fresh human tumor tissue have been widely used to evaluate the pharmacological efficacy of various therapeutic agents and to predict their clinical implementation in the future as an effective study tool for translational medicine[33]. Therefore, to further investigate the antitumor effects of DTLL in combination with gemcitabine, we selected two PDX models (PA1233 and PA3142) as SMAD4-mutant/deficient and SMAD4-wild-type/sufficient models, based on SMAD4 genetic status obtained from available RNA sequencing datasets (https://models.crownbio.com/pancreatic-cancer/) and protein levels by using Western blot assays (Fig. S1b). Similar to the profiles of AsPC-1 and MIA PaCa-2 tumors (Table 1, Fig.S1a), the *SMAD4* gene in PA1233 tumors was mutated (D537Y and E171D) with obviously lower protein levels whereas higher expression was shown in PA3142 due to the wild-type *SMAD4*. Moreover, higher expression levels of EGFR and HER2 were detected in PA3142 models than in PA1233. These two PDX models had previously shown different responses to gemcitabine validated from the report of the company where PA1233 tumors were resistant to gemcitabine with a very slower growth rate and PA3142

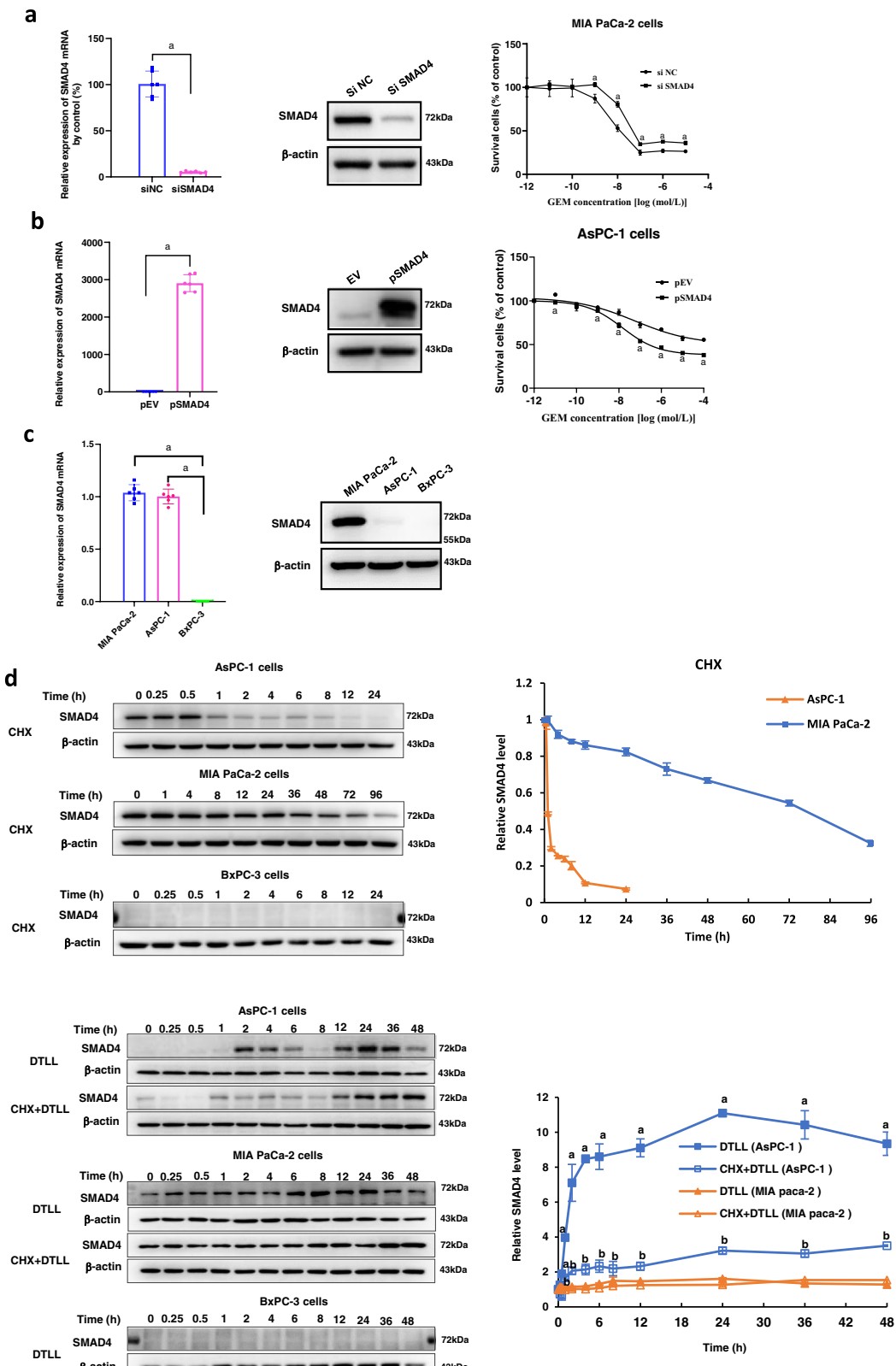

models were sensitive with tumors rapidly developed (https://models. crownbio.com/pancreatic-cancer/). As shown in Supplementary Table 1, the results from the in vivo evaluation of efficacy in PA1233 models indicated that the inhibitory rate of tumor volume on day 28 at the end of the experiment was only 11.65% and 31.54% for gemcitabine and DTLL alone, respectively. However, the combination therapy obviously improved the antineoplastic effect with a 72.25% inhibition

rate. As observed in Fig. 6a, in the SMAD4-deficient PA1233 model with the combination treatment, there were significantly superior anti-neoplastic effects achieved when compared with the gemcitabine or DTLL alone group. In the SMAD4-sufficient PA3142 models, the inhibitory rates of gemcitabine and DTLL were 54.37% and 50.50%, respectively. The combination treatment significantly repressed tumors at 77.17% (Fig. 6b). The results showed that the combination

**Fig. 4 | Downregulation or upregulation of SMAD4 altered PDAC susceptibility to gemcitabine. a** Downregulation of SMAD4 reduced the sensitivity of MIA PaCa-2 cells to gemcitabine. After knocking down SMAD4 with its specific siRNAs, the mRNA and protein levels of *SMAD4* were detected by qRT-PCR and Western blot assays. MIA PaCa-2 cells were exposed to various concentrations of GEM for 72 h and cell viability was measured by MTS assay. The results were obtained from three independent experiments. Data are presented as the mean ± SD ($n = 3$ biologically independent experiments). A pooled siRNA against SMAD4 (siSMAD4) and a scramble siRNA (siNC) indicate siRNA targeting SMAD4 and its negative control, respectively. 'a' indicates, $p < 0.05$, as compared with the control. $p = 1.26 \times 10^{-8}$ in qRT-PCR assay, and $p < 0.001$ at the GEM concentration of $10^{-8}$ mol/L in MTS assay. **b** Upregulation of SMAD4 increased the sensitivity of AsPC-1 cells to gemcitabine. After SMAD4 was upregulated by its overexpression vector, the mRNA and protein levels of *SMAD4* in AsPC-1 cells were detected by qRT-PCR and Western blot assays. The AsPC-1 cell line was exposed to various concentrations of GEM for 72 h, and cell viability was measured by the MTS assay. The results were obtained from three independent experiments. Data are presented as the mean ± SD ($n = 3$ biologically independent experiments). pSMAD4 and pEV indicate SMAD4 overexpression plasmid and its empty vector, respectively. 'a' indicates $p < 0.05$ as compared with the control. $p = 2.42 \times 10^{-11}$ in qRT-PCR assay. In MTS assay $p \leq 0.001$ (pSMAD4 versus pEV) at the concentrations of $10^{-11}$, $10^{-7}$, $10^{-6}$, $10^{-5}$, and $10^{-4}$ mol/L. **c** The differences in S*MAD4* mRNA expression and protein levels in BxPC-3 (SMAD4 deletion), MIA PaCa-2 (wild-type SMAD4) and AsPC-1 (mutant SMAD4) cells. Data are shown as the mean ± SD from three biologically independent samples ($n = 3$). 'a' indicates $p < 0.05$ as compared to BxPC-3 cells with $p = 2.79 \times 10^{-14}$ for AsPC-1 and $4.67 \times 10^{-14}$ for MIA PaCa-2 cells. **d** The protein levels and half-life times of wild-type and mutant SMAD4 in AsPC-1, MIA PaCa-2 and BxPC-3 cells. Data presented are the mean ± SD from three biologically independent samples ($n = 3$). **e** The protein levels of SMAD4 with DTLL induction in AsPC-1, MIA PaCa-2 and BxPC-3 cells. Data are shown as the mean ± SD from three biologically independent samples ($n = 3$). 'a' indicates $p < 0.05$ with comparison of DTLL (AsPC-1) versus DTLL (MIA PaCa-2 cells) and 'b' represents $p < 0.05$ with comparison of CHX + DTLL (AsPC-1) versus CHX + DTLL (MIA PaCa-2 cells). When treated with DTLL, $p < 0.001$ for all time points except for 0, 0.25 and 0.5 h; When treated with CHX + DTLL, $p < 0.001$ for 1, 24, 36, and 48 h. Note: All specific $p$ values are presented in the Source Data file and only significant values are shown in figures. For **a**, **b**, **e**, statistical significance was determined by using two-sided Paired-samples t-test. One-way ANOVA with Bonferroni post hoc test was used in **c**. Source data are provided in the Source Data file.

treatment exhibited highly potent efficacy in both SMAD4-mutant/deficient and SMAD4-wild-type/sufficient PDX models compared to gemcitabine or DTLL alone.

Subsequently, we investigated the drug effects on the expression of relevant proteins in the AKT/mTOR and TGFβ/SMADs signaling pathways in the two PDX models. Similar results were observed (Fig. 6c), in which the combination treatment exhibited similar inhibitory trends in the ratios of p-EGFR/EGFR and pHER2/HER2 in both models. However, the combination significantly upregulated the protein levels of SMAD4 and PD-L1 in PA1233 tumors and downregulated these two proteins in PA3142 models (Fig. 6c). In addition, there were no obvious effects on the p-mTOR/mTOR ratios detected in the PA1233 tumors whereas significantly decreased ratios of p-AKT/AKT and p-mTOR/mTOR were observed in PA3142 tumors treated with DTLL and both. Tumor specimens were tested for IHC staining of Ki-67 and apoptosis in the TUNEL assay with semiquantification (Fig. 6d, e). Similar results were observed with significantly downregulated Ki-67 but increased apoptosis in the combination treatment group as compared to the gemcitabine or DTLL alone group. The above results further demonstrated that the combination treatment showed high inhibitory efficacy with distinct alterations in cell signaling pathways in a SMAD4-dependent manner.

**Evaluation of gemcitabine efficacy in BxPC-3 cells with stably expressing wild-type and mutant SMAD4 in vitro and in vivo**
As we mentioned above, BxPC-3 cells do not express SMAD4 mRNA or protein of owing to gene deletion. Therefore, we used this cell line to generate three types of stably transfected cells containing empty control, mutant, and wild-type SMAD4 overexpression vectors after creating the mutant SMAD4 overexpression vector via site-mutagenesis assay (Fig. 7a) to further test the different effects of wild-type and mutant SMAD4 on drug efficacy. There was no apparent difference in the mRNA level of *SMAD4* in BxPC3-Mut and BxPC3-WT cells, while there was a much lower level of mutant protein than wild-type one, as detected by qRT-PCR and Western blot assays (Fig. 7b, c).

We further evaluated the in vitro inhibitory effects of gemcitabine on the growth of BxPC-3 cells with empty vector (BxPC3-EV), wild-type (BxPC3-WT) and mutant (BxPC3-Mut) overexpression vectors. As shown in Fig. 7d, BxPC3-WT cells were more sensitive to gemcitabine than BxPC3 mock or BxPC3-EV cells, but BxPC3-Mut cells showed little difference from those two. Gemcitabine or DTLL alone, and their combination were tested for proliferative inhibition of BxPC3-EV, BxPC3-WT and BxPC3-Mut cells by using MTS and CyQUANT assays (Fig. 7e, f), respectively. The results indicated that the combination of gemcitabine with DTLL showed the most significant inhibition rate of

tumor cell growth when compared to gemcitabine or DTLL alone, obviously showing a synergistic effect evaluated by CI calculation, not a simple additive effect. In addition, the results from the cytometry assay (Fig. 7g) showed that more G1/S arrest was observed in BxPC3-WT cells treated with gemcitabine than in the other two types of cells, partly explaining the sensitivity of BxPC3-WT cells overexpressing SMAD4 to gemcitabine. However, DTLL or its combination therapy significantly induced an increase in G2/M cells in both BxPC3-WT and BxPC3-Mut lines. This further confirmed that DTLL potentiated the inhibitory effects of gemcitabine on cell proliferation and cycle distribution.

Next, we used the above BxPC3-EV, BxPC3-WT, and BxPC3-Mut stably transfected cell lines to generate in vivo CDX mouse models for evaluation of antineoplastic efficacy of vehicle, gemcitabine or DTLL alone, and the combination. As shown in Fig. 7h and Supplementary Table 2, gemcitabine had a minimal inhibitory effect on BxPC3-Mut tumors at 7.46% and DTLL slightly repressed tumor growth by 47.61%, however, the combination of gemcitabine and DTLL had remarkable synergistic efficacy with a tumor inhibition rate of 78.62% in this gemcitabine-resistant xenograft model. In the BxPC3-WT models, gemcitabine or DTLL alone was able to apparently inhibit tumor growth by 65.33% or 59.19%, respectively, suggesting its sensitivity to both drugs. The combination treatment significantly repressed tumor growth by 82.06%. In addition, there were no deaths or significant changes in body weight observed in treated mice (Fig. S2c), as well as no toxic-pathological organs (Fig. S3c), compared to the control group. This further confirmed the synergistic inhibitory effect of DTLL in combination with gemcitabine on in vivo tumor growth, similar to the above results obtained in both CDX and PDX models.

All the above results demonstrated that different SMAD4 genetic statuses in PDAC cells are responsible for distinct protein levels between mutant and wild-type SMAD4. Expression has been proven to affect cellular susceptibility to gemcitabine by using ex/in vivo MIA PaCa-2, AsPC-1, and BxPC-3 cell lines or CDX/PDX mouse models with either mutant or wild-type SMAD4 expression. We confirmed that DTLL sensitizes gemcitabine efficacy in those models, and further conducted functional characterization in follow-up studies with the above BxPC3-EV, BxPC3-WT. and BxPC3-Mut stably transfected cell lines.

**Distinct protein levels and half-life times of mutant and wild-type SMAD4 with/without DTLL induction in BxPC-3 cells**
There was no apparent difference in *SMAD4* mRNA levels between AsPC-1 vs. MIA PaCa-2 (left panel in Fig. 4c) and BxPC3-Mut vs. BxPC3-WT cells (Fig. 7b), indicating no impact on *SMAD4* mRNA transcription

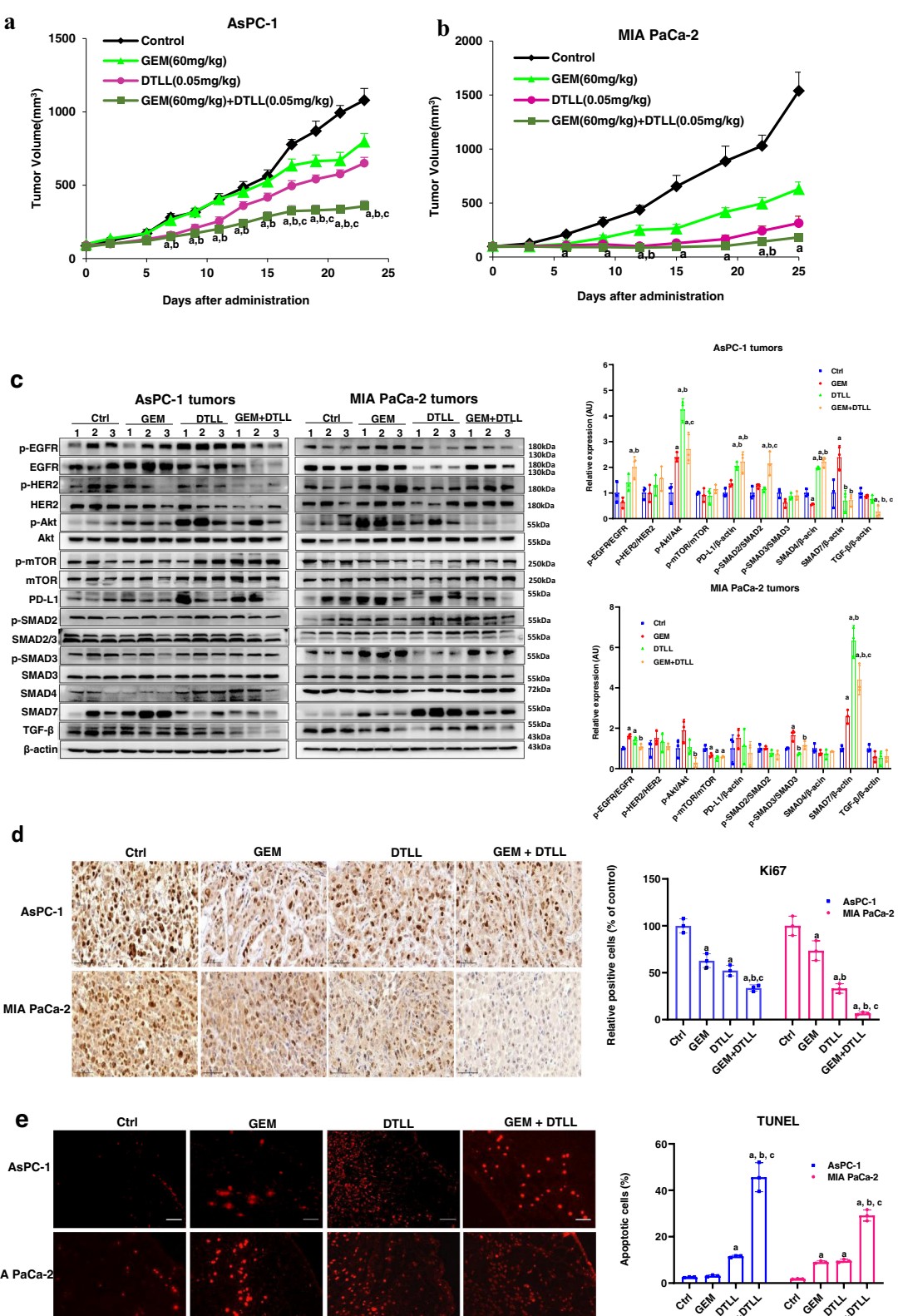

affected by the R100T mutation. Therefore, we tested whether there was a difference in protein degradation between mutant and wild-type SMAD4 in BxPC3-WT and BxPC3-Mut stably transfected cell lines by Western blot assays, followed with DTLL treatment. The results demonstrated that the mutant SMAD4 protein was degraded rapidly (Fig. 8a) but dramatically induced by DTLL (Fig. 8b) in BxPC3-Mut cells, whereas wild-type SMAD4 remained much longer and showed a slight

increase via DTLL treatment in BxPC3-WT cells (Fig. 8a, b), consistent with the results from the AsPC-1 and MIA PaCa-2 lines (Fig. 4d, e). TRIM33 protein, as a specific E3 ligase for SMAD4 ubiquitin-proteasome mediated degradation, was steeply decreased in BxPC3-WT cells after being treated with DTLL from 0, 0.25 till 8 h. However, its level maintained even for 24 h if given DTLL induction and gradually started to decrease in BxPC3-Mut cells (Fig. 8b), which in part

**Fig. 5 | Evaluation of in vivo efficacy by cell line-derived xenograft (CDX) models of human pancreatic cancer.** Inhibitory effect on tumor growth of cells treated with gemcitabine, DTLL and both by using two CDX models of AsPC-1 (**a**) and MIA PaCa-2 (**b**). In both CDX models, the mice received an equal volume of physiological saline (control group), 60 mg/kg gemcitabine intraperitoneally administered once a week (GEM group), 0.05 mg/kg DTLL at the LDM-equivalent dose intravenously administered (DTLL group) every ten days, and the same dosing administration for either DTLL or gemcitabine (GEM + DTLL group). Tumor volume in each group was measured and calculated for comparison between 'GEM + DTLL' versus other groups. Data are representative of biologically independent replicates as the mean ± SEM ($n = 6$ for AsPC-1 and $n = 5$ for MIA PaCa-2 tumors). In AsPC-1 CDX model, $p < 0.001$ ('GEM + DTLL' versus Control) for day 7, 11, 13, 15, 17, 19, 21, and 23; $p < 0.001$ ('GEM + DTLL' versus GEM) for day 11, 13, 15, 17, 19, 21, and 23. In MIA PaCa-2 CDX model, $p < 0.001$ ('GEM + DTLL' versus Control) for day 9, 12, 19, and 22. **c** Protein expression levels in tumor tissue samples from either AsPC-1 or MIA PaCa-2 models after treatment with GEM, DTLL or both were determined by Western blot assay. Band intensities were quantified using Image J. Data are displayed as the mean ± SD ($n = 3$ biologically independent replicates). In AsPC-1 CDX model, $p < 0.001$ for SMAD4 between 'GEM + DTLL' versus Control, 'GEM + DTLL' versus GEM or DTLL versus GEM groups; $p < 0.001$ for p-Akt/Akt between DTLL versus Control groups. In MIA PaCa-2 CDX model, $p < 0.001$ for SMAD7 between DTLL versus Control groups. **d** IHC staining of Ki-67 in paraffin-embedded tumor tissue samples from AsPC-1 and MIA PaCa-2 xenograft models (×100). Tumor sections were deparaffinized, rehydrated and incubated with Ki-67 antibody, followed by incubation with secondary antibody. The results were observed by an inverted microscope using DAB as a chromogenic reagent. Images are representative of three biologically independent replicates with a scale bar representing 50 µm ($n = 3$). In either AsPC-1 or MIA PaCa-2 tumors, $p < 0.001$ between 'GEM + DTLL' versus Control, as well as DTLL versus GEM. In AsPC-1 tumors, $p < 0.001$ between GEM versus Control groups. **e** Apoptotic cells in the AsPC-1 and MIA PaCa-2 models were measured in TUNEL assay (×200). Apoptotic cells in tumor tissue were determined by the TUNEL method according to the manufacturer's instructions. They were observed and photographed under a fluorescence microscope. Red fluorescence represents the positive cells. Images are representative of three biologically independent replicates with a scale bar indicating 25 µm ($n = 3$). In both AsPC-1 tumors and MIA PaCa-2 tumors, $p < 0.001$ when compared 'GEM + DTLL' versus all three Control, GEM and DTLL groups. In MIA PaCa-2 tumors, $p < 0.001$ (DTLL versus Control, or GEM versus Control). Note: For **a–e**, 'a' indicates $p < 0.05$ as compared with the control, 'b', $p < 0.05$ compared with the GEM group and 'c', $p < 0.05$ compared with the DTLL group. All specific $p$ values are presented in the Source Data file and only significant values are shown in figures. For **a-e**, statistical significance was determined by using one-way ANOVA with Bonferroni post hoc test. Source data are provided in the Source Data file.

explained why the degradation velocity of mutant SMAD4 was significantly retarded if given DTLL induction. Next, we tested the protein levels of SMAD4 after cotreatment with CHX to exclude the influence of DTLL induction on protein synthesis and further demonstrated more significant induction of SMAD4 via DTLL in BxPC3-Mut cells than in BxPC3-WT cells (Fig. 8c). This result confirmed that SMAD4 genetic status is responsible for SMAD4 protein expression and that the difference in TRIM33 mediated proteasome degradation between mutant and wild-type proteins partially contributed to their distinguished protein levels. Moreover, DTLL significantly prolonged the half-life time by inhibiting SMAD4 degradation but impacted wild-type and mutant proteins to different degrees, thereby reactivating SMAD4 function with restored proteins, especially in BxPC3-Mut cells, consequently leading to enhanced cellular susceptibility to gemcitabine.

## Different cell signaling pathways affected by mutant and wild-type SMAD4 with/without DTLL induction in BxPC-3 cells

To further explore the mechanism of SMAD4 action by which differences in protein levels of mutant and wild-type SMAD4 in PDAC cells contributed to distinguishing the gemcitabine response, we detected any alterations in the TGF-β/SMAD4 and EGFR/AKT/mTOR signaling pathways by Western blot assay. As shown in Fig. 8d, both the BxPC3-WT and BxPC3-Mut lines had decreased expression of p-EGFR, EGFR, p-HER2, and HER2. Similar to the observation in AsPC-1 and MIA PaCa-2 cells, significant decreases in p-AKT/AKT and p-mTOR/mTOR expression were observed in BxPC3-WT cells, while an increase in p-AKT/AKT was detected in BxPC3-Mut cells. However, there was significantly induced expression of SMAD4, p-SMAD2/SMAD2, p-SMAD3/SMAD3, and TGF-β, but an apparent decrease in SMAD7 in the BxPC3-Mut lines compared to the BxPC3-WT lines (Fig. 8d), further confirming the results from the observations in the AsPC-1 and MIA PaCa-2 lines (Fig. 3a, b).

Next, we tested changes in cell cycle relevant proteins to determine if DTLL affected the expression level of any protein mediated by SMAD4. With DTLL or its combination treatment, the expression levels of cyclin D3, cyclin B1, CDK2, phos-CDC2, phos-Wee1, and P27 were reduced in both the BxPC3-Mut and BxPC3-WT lines (Fig. 8e). However, significantly increased expression of P21, cyclin E2 and CDK4 in the BxPC3-Mut line was observed, while decreases in those proteins were detected in the BxPC3-WT line (Fig. 8e).

Interestingly, we found that the expression levels of phospho-P65, P65, and P50 units of NF-κB, a well-known transcription factor, were induced significantly by DTLL or its combination in the BxPC3-Mut line. Along with increased mutant SMAD4 by DTLL induction, apoptotic proteins including BAX, FADD, and cleaved caspase-8 were significantly induced, while anti-apoptotic Bcl-2 and MCL1 proteins were reduced (Fig. 8f). Moreover, the semiquantification results in BxPC3-Mut cells demonstrated that there were significant decreases in the ratio of Bcl-2/BAX and MCL1/BAX after treatment with DTLL or its combination compared to the control, indicating the consequence of enhanced cell apoptosis by DTLL. On the other hand, in the BxPC3-WT line, all the above proteins were apparently reduced with decreased SMAD4, except for induced cleaved caspase-8 (Fig. 8f). The results from semiquantification in the BxPC3-WT line also demonstrated the consequence of cell apoptosis with significant decreases in the ratio of Bcl-2/BAX and MCL1/BAX after treated by DTLL or its combination compared to the control. The directions of the changes in SMAD4 and NF-κB expression induced by DTLL, as well as the expression of their targeted apoptotic/anti-apoptotic proteins, were opposite between the BxPC3-WT and BxPC3-Mut lines, but eventually led to the consequence of enhanced cell apoptosis and sensitized drug response.

Moreover, to determine if the expression of apoptotic (BAX, FADD) and anti-apoptotic (Bcl-2 and MCL1) proteins was transcriptionally regulated by NF-κB mediated via either mutant or wild-type SMAD4, we tested their mRNA levels in the same cell samples by qRT-PCR. Indeed, the mRNA levels of P50, P65, BAX, and FADD as SMAD4 targets were obviously upregulated along with the same increasing tendency at the *SMAD4* mRNA level in BxPC3-Mut cells when treated with DTLL or the combination, as compared to the control or gemcitabine-treated cells, implying that SMAD4 directly promoted its downstream target NF-κB which further regulated BAX and FADD expression as a regulatory transcription. However, slight decreases in Bcl-2 and MCL1 expression were detected in DTLL-treated cells, independent of NF-κB regulation promoted by SMAD4. On the other hand, along with little change in *SMAD4* mRNA expression in BxPC3-WT cells treated with DTLL or the combination, the mRNA levels of P50, BAX, Bcl-2, and MCL1 were significantly lower than those in the control or gemcitabine-treated cells, (Fig. 8g). The results suggested that, in BxPC3-WT cells, decreased NF-κB was upstream of BAX, Bcl-2, and MCL1 but not transcriptionally regulated by SMAD4, and DTLL altered NF-κB transcriptional activity but did not affect its expression through SMAD4.

These alterations in the mRNA levels of these two lines were largely consistent with the above results from the Western blot assay (Fig. 8f). In addition, all of the above relevant proteins were detected

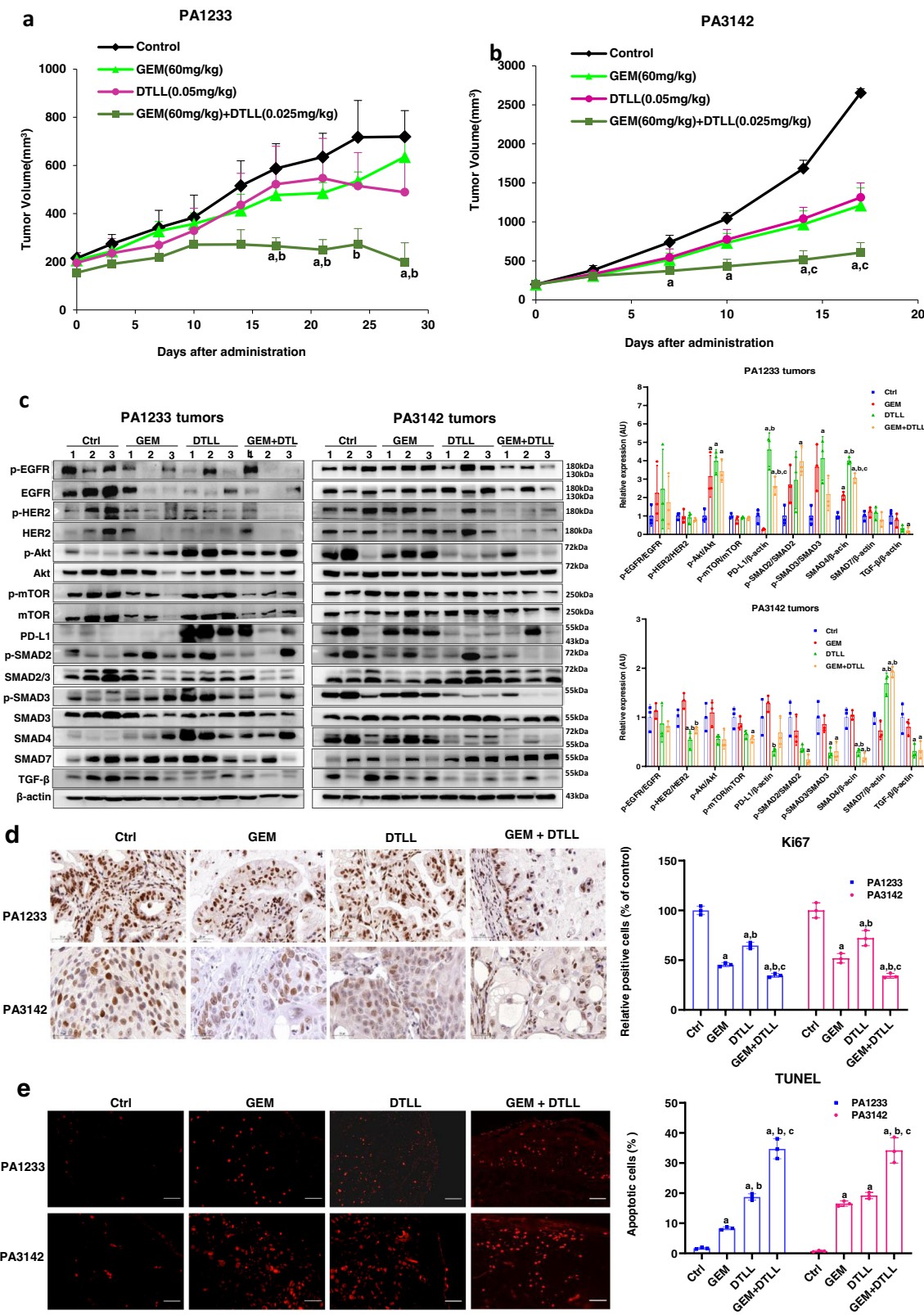

for validating their basic expression profiles in the BxPC3-EV, -WT and -Mut cells (Fig. S4).

**Different protein interactions of mutant or wild-type SMAD4 with/without DTLL induction in BxPC-3 cells**

The results from the co-IP assay further indicated that both mutant and wild-type SMAD4 proteins were able to interact with P50 and P65

(Fig. 8h), which implied that SMAD4 might impact on the transcriptional bioactivity of NF-κB, which was further responsible for regulating the expression of the above mRNAs and proteins in both BxPC3-WT and BxPC3-Mut cells. Moreover, the interactions of wild-type SMAD4 with P50 and P65 were significantly inhibited in BxPC3-WT cells after treated by DTLL but were apparently increased in DTLL-treated BxPC3-Mut cells (Fig. 8i). This might explain the reason for the difference

**Fig. 6 | Evaluation of in vivo efficacy by patient-derived xenograft (PDX) models of human pancreatic cancer.** Inhibitory effect of gemcitabine, DTLL and both in two PDX models of PA1233 (**a**) and PA3142 (**b**) on tumor growth. In both PDX models, the mice were received equal volumes of physiological saline (control group), 60 mg/kg gemcitabine every four days (GEM group), 0.05 mg/kg DTLL once a week (DTLL group) and 60 mg/kg gemcitabine combined with 0.025 mg/kg DTLL (GEM + DTLL group). Data are representative of biologically independent replicates as the mean ± SEM (*n* = 3 for PA1233 and *n* = 4 for PA3142 tumors). In PA3142 model, *p* < 0.001 ('GEM + DTLL'versus Control) for day 14 and 17. **c** Protein expression levels in tumor tissue samples from PA1233 or PA3142 models after treatments with GEM, DTLL or both were determined by Western blot assays. Band intensities were quantified using Image J. Data are displayed as the mean ± SD (*n* = 3 biologically independent replicates). In PA1233 PDX model, *p* < 0.001 for PD-L1 between DTLL versus Control or DTLL versus GEM; *p* < 0.001 for SMAD4 between 'GEM + DTLL' versus Control, DTLL versus Control or DTLL versus GEM. In PA3142 PDX model, *p* < 0.001 for SMAD4 between 'GEM + DTLL' versus Control, 'GEM + DTLL' versus GEM, DTLL versus Control or DTLL versus GEM; p < 0.001 for SMAD7 between 'GEM + DTLL' versus Control, 'GEM + DTLL' versus GEM or DTLL versus GEM. **d** IHC staining of Ki-67 in paraffin-embedded tumor tissue samples from either PA1233 or PA3142 models (×100). Tumor sections were deparaffinized, rehydrated and incubated with Ki-67 antibody, followed by incubation with

secondary antibody. The results were observed by an inverted microscope using DAB as a chromogenic reagent. Images are representative of three biologically independent replicates with a scale bar representing 50 μm (*n* = 3). In either PA1233 or PA3142 tumors, *p* < 0.001 between 'GEM + DTLL' versus Control, 'GEM + DTLL' versus DTLL or GEM versus Control. In PA1233 tumors, *p* < 0.001 between DTLL versus Control or DTLL versus GEM. **e** Apoptosis in the PA1233 and PA3142 models was measured in TUNEL assay (×200). Apoptotic cells in tumor tissue were determined by the TUNEL method according to the manufacturer's instructions. They were observed and photographed under a fluorescence microscope. Red fluorescence represents the positive cells. Images are representative of biologically independent replicates with a scale bar representing 25 μm (*n* = 3). In either PA1233 or PA3142 tumors, *p* < 0.001 between 'GEM + DTLL' versus Control, 'GEM + DTLL' versus GEM, 'GEM + DTLL' versus DTLL or DTLL versus Control. In PA1233 tumors, *p* < 0.001 between DTLL versus Control. In PA3142 tumors, *p* < 0.001 between GEM versus Control. Note: For **a**–**e**, 'a' indicates *p* < 0.05 as compared with the control, 'b', *p* < 0.05 compared with the GEM group and 'c', *p* < 0.05 compared with the DTLL group. All specific *p* values are presented in the Source Data file and only significant values are shown in figures. For a–b, statistical significance was determined by using paired-samples *t*-test was two-sided. For **c**–**e**, statistical significance was determined by using one-way ANOVA with Bonferroni post hoc test. Source data are provided in the Source Data file.

intendancies in the downstream gene expression affected by NF-κB transcriptional activity between those two cell lines after treated by DTLL. Owing to the increased bioactivity of mutant SMAD4-mediated NF-κB given DTLL, apoptotic BAX and FADD proteins were significantly induced and resulted in cell death (increased cleaved caspase-8). However, the transcriptional bioactivity of NF-κB promoted by wild-type SMAD4 was inhibited by DTLL, and thus anti-apoptotic proteins (Bcl-2 and MCL1) were downregulated, mainly responsible for the increase in programmed cell death (enhanced cleaved caspase-8).

In accordance with the results of TRIM33 alteration over the time course shown in Fig. 8b, the mRNA and protein levels of TRIM33 in BxPC3-Mut cells were decreased but not as significantly as those in BxPC3-WT cells after treatment with DTLL for 24 h (Fig. 8f, g). This result suggested that TRIM33 was not regulated by SMAD4, and seemed to be the feedback of either mutant or wild-type SMAD4 expression. We also demonstrated the interaction of TRIM33 with either mutant or wild-type SMAD4 proteins (Fig. 8h). The interaction between wild-type SMAD4 and TRIM33 was significantly inhibited in BxPC3-WT cells after treatment with DTLL, while their interaction was apparently increased in DTLL-treated BxPC3-Mut cells (Fig. 8i). Different levels of TRIM33 expression in both cell lines were decreased but to different degrees. Therefore, we speculated that, in BxPC3-WT cells with high levels of wild-type SMAD4 expression, TRIM33 was responsible for its degradation to a certain degree and interacted with SMAD4 to maintain its expression balance. If given DTLL or its combination treatment, TRIM33 was reduced rapidly within 8 h with its apparently impaired interaction with wild-type SMAD4 to retard its degradation and maintain its expression balance. However, little expression of mutant SMAD4 in BxPC3-Mut cells was accompanied by a low level of TRIM33 which mediates its proteasome degradation. With DTLL or its combination treatments, TRIM33 was decreased at slower and lower velocities, thus resulting in very slow degradation of mutant SMAD4 and increased accumulation of SMAD4 protein expression. Consequently, the increased accumulation of mutant SMAD4 expression is accompanied by its significantly enhanced interaction with TRIM33 by DTLL to compensate for its degradation for a balance.

**Different proteomic and gemcitabine pharmacokinetic profiles in mutant and wild-type SMAD4 cells with/without DTLL induction in BxPC-3 cells**
In addition, we further performed a proteomic study to determine if SMAD4 has effects on the expression of gemcitabine-relevant transporters or pharmacokinetic enzymes. All the proteomic raw data and

the results files have been uploaded to the iProX Consortium (https://www.iprox.org/) with PXD identifiers (PXD031977). As shown in the heatmap (left panel of Fig. 8j) and Supplementary Data 3, the proteomic profiles of BxPC3-Mut and -WT cells without any treatments were quite distinct and showed fewer changes after gemcitabine treated both lines, respectively. DTLL or its combination obviously altered the profile of BxPC3-Mut cells compared to the control or gemcitabine-induced profile, unlike BxPC3-WT cells. Next, we found a total of 12 proteins of gemcitabine-relevant transporters or pharmacokinetic enzymes detectable (Supplementary Table 3), including ABCC1, CDA, CDC5L, CMPK1, DCK, DCTD, NT5C, NT5C2, NT5C3A, RRM1, RRM2, and SLC29A1. Except for DCK, DCTD, NT5C2 and NT5C3A, all the other nine proteins were significantly altered, as shown in the expression heatmap (right panel of Fig. 8j) between BxPC3-Mut versus BxPC3-WT cells after treatments. The expression of SMAD4 was included as a positive control, which confirmed that SMAD4 expression alterations (Ratio $_{Mut/WT}$ = 0.017, $p = 7.3 \times 10^{-5}$), similar to the results in Western blot assay. Specifically, the expression of RRM2 was much lower in BxPC3-Mut cells than in BxPC3-WT cells (Ratio $_{Mut/WT}$ = 0.22, $p = 2.24 \times 10^{-6}$), and was significantly induced after treated with DTLL and its combination, with a similar altered trend of SMAD4 in those cells. The proteomic results implied that SMAD4 status significantly impacted the proteomic profiles as a result of chemoresistance properties that DTLL might also affect the proteomic profiles of both lines, and that it altered cellular susceptibility via RRM2 induction regulated by SMAD4 in BxPC3-Mut cells.

All the above results suggested different action mechanisms of DTLL enhancing cellular susceptibility on the basis of SMAD4 genetic or expression status in BxPC3-Mut and BxPC3-WT cells, respectively, thereby shedding light on its synergistic therapeutic efficacy.

## Discussion
To date, improving the efficacy of first-line chemotherapeutic agents through targeted agents, gene therapy, and combinational treatments has important clinical significance. Gemcitabine is considered as the first-line drug for PDAC and has a poor response rate of ~20% with a median survival of 6 months[34,35]. Even if gemcitabine was combined with other agents to improve efficacy for PDAC, the results are still very limited with a few weeks-months of increase in patient survival time generally[36–38]. Most failures in gemcitabine-based regimens in PDAC were attributed to drug resistance and subsequent relapse or metastasis[39–42]. In light of this, it would provide promising insights into clinical applications to develop therapeutic strategies for pancreatic cancer.

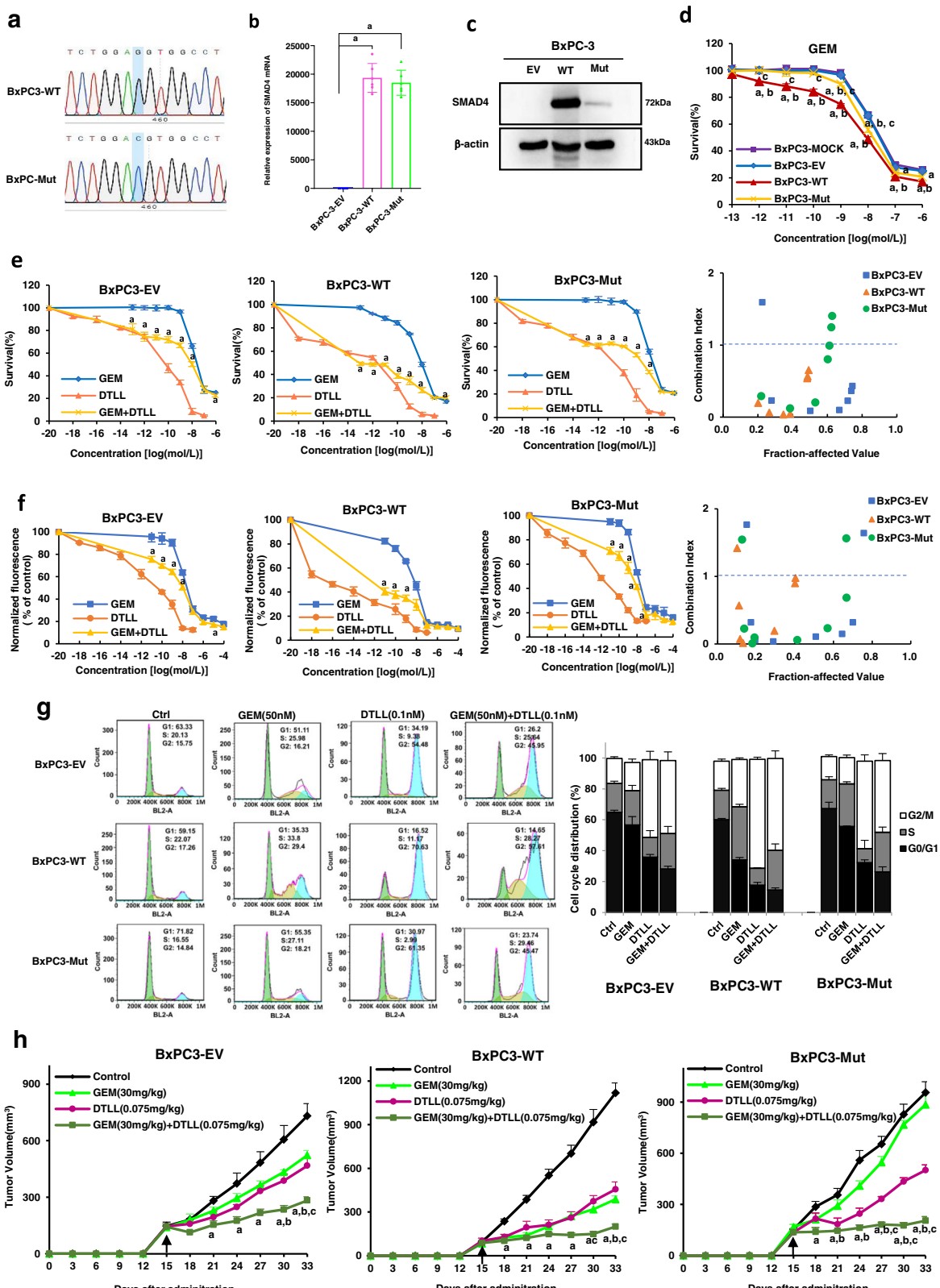

In the present study, we first reported that DTLL was able to improve the gemcitabine efficacy of pancreatic cancer and overcome its resistance as an ADC-like immunotherapeutic agent. Our laboratory had previously reported that DTLL exhibited potent effectiveness in a variety of cells and xenograft tumors including ovarian, esophageal, and pancreatic cancer[24–26]. However, the use of DTLL for combinational therapy has not been explored. Our results showed that the

combination of gemcitabine plus DTLL had synergistic effects on both gemcitabine-resistant and gemcitabine-sensitive mouse models of PDAC (Figs. 2, 5, 6). Furthermore, we identified a critical role of SMAD4 in mediating the responses of pancreatic cancer cells to gemcitabine chemotherapy. Based on differences in genetic status and protein expression of SMAD4 as well as drug susceptibilities between AsPC-1 and MIA PaCa-2 cells, we chose these two lines as models of SMAD4-

**Fig. 7 | Evaluation of gemcitabine efficacy in BxPC-3 cells stably expressing wild-type and mutant SMAD4 in vitro and in vivo. a** Sequence of wild-type Smad4 and mutant Smad4 R100T constructs. **b** Real-time qRT-PCR analysis of the mRNA level of SMAD4 in BxPC3-EV, BxPC3-WT and BxPC3-Mut stably transfected cells. BxPC3-EV, BxPC3-WT and BxPC3-Mut indicate that BxPC3 cells were stably transfected with the empty vector, wild-type SMAD4 or mutant SMAD4 (R100T) overexpression vectors. Data are shown as the mean ± SD from three biologically independent samples ($n = 3$). 'a' indicates $p < 0.001$ as compared with BxPC3-EV with $p = 1.38 \times 10^{-10}$ for BxPC3-Mut and $7.24 \times 10^{-11}$ for BxPC3-WT cells. **c** Western blot analysis of SMAD4 in BxPC3-EV, BxPC3-WT and BxPC3-Mut stably transfected cells. The results were obtained from three biologically independent experiments as the mean ± SD ($n = 3$). **d** Effects of overexpression of wild type and mutant SMAD4 on the sensitivity of BxPC-3 cell lines to gemcitabine. The BxPC-3 cell line was exposed to various concentrations of GEM for 72 h and cell viability was measured by the MTS assay. Data are shown as the mean ± SD from three biologically independent replicates ($n = 3$). 'a' indicates $p < 0.05$, compared with BxPC3-MOCK; **b** $p < 0.05$, compared with the BxPC3-EV; **c** $p < 0.05$, compared with the BxPC3-WT. In BxPC3-WT cells, $p < 0.001$ for $10^{-10}$, $10^{-9}$ and $10^{-8}$ mol/L of GEM as compared to either BxPC3-MOCK or BxPC3-EV; and $p < 0.001$ at $10^{-6}$ mol/L as compared to BxPC3-MOCK. In BxPC3-Mut cells, $p < 0.001$ at $10^{-8}$ mol/L of GEM as compared to BxPC3-MOCK. **e** Effects on drug cytotoxicity in BxPC3-EV, BxPC3-WT and BxPC3-Mut stably transfected cells after treatment with GEM, DTLL or GEM combined with $10^{-14}$ mol/L DTLL were detected by MTS assays. Data are shown as the mean ± SD from three biologically independent replicates ($n = 3$). 'a' indicates $p < 0.05$ as compared with the GEM group. In BxPC3-EV cells, $p < 0.001$ at $10^{-12}$, $10^{-11}$, $10^{-10}$, $10^{-9}$ and $10^{-8}$ mol/L of GEM between 'GEM + DTLL' versus GEM. In either BxPC3-WT or BxPC3-Mut cells, $p < 0.001$ between 'GEM + DTLL' versus GEM at all concentrations of GEM except for $10^{-7}$ and $10^{-6}$ mol/L of GEM. **f** The inhibitory effect of GEM and DTLL on cell proliferation in BxPC3-EV, BxPC3-WT and BxPC3-Mut stably transfected cells was measured by a CyQuant proliferation assay. The cells were exposed to a series of concentrations of GEM, DTLL or GEM combined with $10^{-14}$ mol/L DTLL for 72 h, respectively. Data are shown as the mean ± SD from three biologically independent replicates (n = 3). As compared 'GEM + DTLL' to GEM, $p < 0.001$ for $10^{-9}$ mol/L of GEM in BxPC3-EV cells, $p < 0.001$ at $10^{-13}$ mol/L in BxPC3-WT cells, and $p < 0.001$ at $10^{-11}$, $10^{-12}$ and $10^{-13}$ mol/L in BxPC3-Mut cells. **g** Cell cycle distribution in BxPC3-EV, BxPC3-WT and BxPC3-Mut stably transfected cells after treated with GEM, DTLL or both was detected by flow cytometry assays. Cells were treated with 0.005 μM gemcitabine, 0.1 nM DTLL or their combination for 24 h. The cell cycle distribution was evaluated using propidium iodide (PI) staining and analyzed by flow cytometry. Data are representative of biologically independent triplicates ($n = 3$). **h** Inhibitory effect on tumor growth of treatments with gemcitabine, DTLL and both in these three CDX models derived from BxPC3-EV, BxPC3-WT and BxPC3-Mut cells. In those models, the mice received an equal volume of physiological saline (control group), 30 mg/kg gemcitabine intraperitoneally administered every four days (GEM group), 0.075 mg/kg DTLL at the LDM-equivalent dose intravenously administered once a week (DTLL group) and 30 mg/kg gemcitabine combined with 0.075 mg/kg DTLL ('GEM + DTLL'group). Data are presented as the mean ± SEM for biologically independent replicates as the mean ± SD ($n = 5$). In BxPC3-EV CDX model, $p < 0.001$ ('GEM + DTLL'versus Control) for day 27, 30 and 33. In BxPC3-WT CDX model, $p < 0.001$ ('GEM + DTLL'versus Control) for day 21, 24, 27, 30, and 33. In BxPC3-Mut CDX model, $p < 0.001$ ('GEM + DTLL' versus Control) for day 24, 27, 30, and 33; $p < 0.001$ ('GEM + DTLL' versus GEM) for day 27, 30, and 33; $p < 0.001$ ('GEM + DTLL'versus DTLL) for day 30. One-way ANOVA with Bonferroni post hoc test was used in **b**, **d**, **h**; Paired-samples $t$-test was two-sided and used in **e** and **f**. All specific $p$ values are presented in the Source Data file and only significant values are shown in figures. Source data are provided in the Source Data file.

deficient/gemcitabine-resistant and SMAD4-sufficient/gemcitabine-sensitive PDAC, respectively. The results demonstrated that the combination treatment produced a significantly synergistic inhibitory effect on tumor cell growth, especially on SMAD4-defficient/gemcitabine-resistant AsPC-1 cells or xenografted tumors (Figs. 2, 5). To predict the enhanced effect of DTLL on gemcitabine sensitivity for clinical implementation in the future, two PA1233 and PA3142 PDX models based on their SMAD4 profiles were utilized to evaluate pharmacological efficacy (Fig. 6), confirming our observations from the CDX models. The in vitro and in vivo results from BxPC-3 stably transfected cells containing empty, mutant and wild-type SMAD4 overexpression vectors further demonstrated that SMAD4-deficient/sufficient cells showed distinct resistant/sensitive responses to gemcitabine (Fig. 7 and Supplementary Table 2), because of differences in wild-type and mutant SMAD4 protein expression. Another report also demonstrated that MIA-PaCa-2 tended to be the cell line with the highest intrinsic chemosensitivity, whereas PANC-1 and AsPC-1 were the cell lines with the most distinct intrinsic chemoresistance. Their results further confirmed our findings[43].

SMAD4 (also known as DPC4) was identified on the basis of frequent homozygous deletions and mutations[44], serves as the central mediator of TGF-β/SMADs signaling and functions as a tumor suppressor with over 50% of SMAD4 loss or inactivation in PDAC[45,46], among which homozygous deletion and inactivation of SMAD4 were found in ~30% and 20% of pancreatic cancer patients, respectively. SMAD4 loss was directly associated with poor prognosis[47,48], short survival rate[49], metastasis[50], and radioresistance[51] in pancreatic carcinoma patients. SMAD4 was associated with drug resistance to 5'-fluorouracil in colon cancers[52] or gemcitabine in hepatocellular carcinoma[53]. However, few studies have reported the correlation between SMAD4 and gemcitabine resistance in pancreatic cancer. The impact of SMAD4 loss on chemoresistance has not been fully characterized. Consequently, few therapeutic agents have been developed to overcome chemoresistance.

Interestingly, in the present study, we found that SMAD4 expression and bioactivity of AsPC-1/BxPC3-Mut and MIA PaCa-2/BxPC3-WT cells were distinct although DTLL or its combination treatment had inhibitory effects on both lines. In AsPC-1 cells deficient in SMAD4, we observed reactivation of SMAD4 with nuclear translocation to form a SMAD2/3/4 complex in the nucleus, and a decrease in SMAD7 expression (Fig. 3b–d), similar to the findings in Fig. 8D. The SMAD complex in the nucleus then regulates the expression of different genes through interaction with DNA and DNA-binding proteins and acts as a tumor suppressor, eventually inhibiting tumor growth[54,55]. This result indicated that the combination treatment significantly inhibited PDAC tumor growth in a SMAD4-deficient/gemcitabine-resistant model predominantly through a SMAD4-dependent signaling pathway.

In contrast, the combination treatment obviously inhibited SMAD4 activity in MIA PaCa-2 cells (Fig.3b–d), consistent with the results verified in BxPC3-WT cells (Fig. 8d). Our findings demonstrated that the combination treatment enhanced antineoplastic activities via blockage of AKT/mTOR signaling pathways in gemcitabine-sensitive PDAC models. A previous study demonstrated that SMAD4 inhibited tumor metastasis in patients with colon cancer through multiple processes including the inhibition of apoptosis, epithelial–mesenchymal transition (EMT), and the PI3K/AKT signaling pathway[56]. Their findings indicated an association between SMAD4 and the AKT pathway. In the present study, we also observed decreases in the expression of both SMAD4 and AKT/mTOR signaling (Fig. 3a, b), suggesting that AKT/mTOR signaling blockade by DTLL or the combination treatment in SMAD4-sufficient PDAC cells might be attributed to not only blockade of EGFR/HER2 signaling, but also decreased SMAD4.

As shown in the schematic illustration model in Fig. 9, we investigated different action mechanisms of mutant and wild-type SMAD4 on the PDAC drug response with/without DTLL induction (shown in Fig. 8). Our findings demonstrated that the *SMAD4* genetic status of PDAC is responsible for SMAD4 protein levels, which determine different cellular susceptibilities. Moreover, DTLL significantly altered the half-life time and protein levels of mutant and wild-type SMAD4 by inhibiting protein degradation at different velocities and changing the interaction of SMAD4 with TRIM33 (Fig. 8a–c). In addition, DTLL predominantly enhanced the expression of apoptotic proteins (BAX, FADD) in BxPC3-Mut cells but promoted decreases mainly in

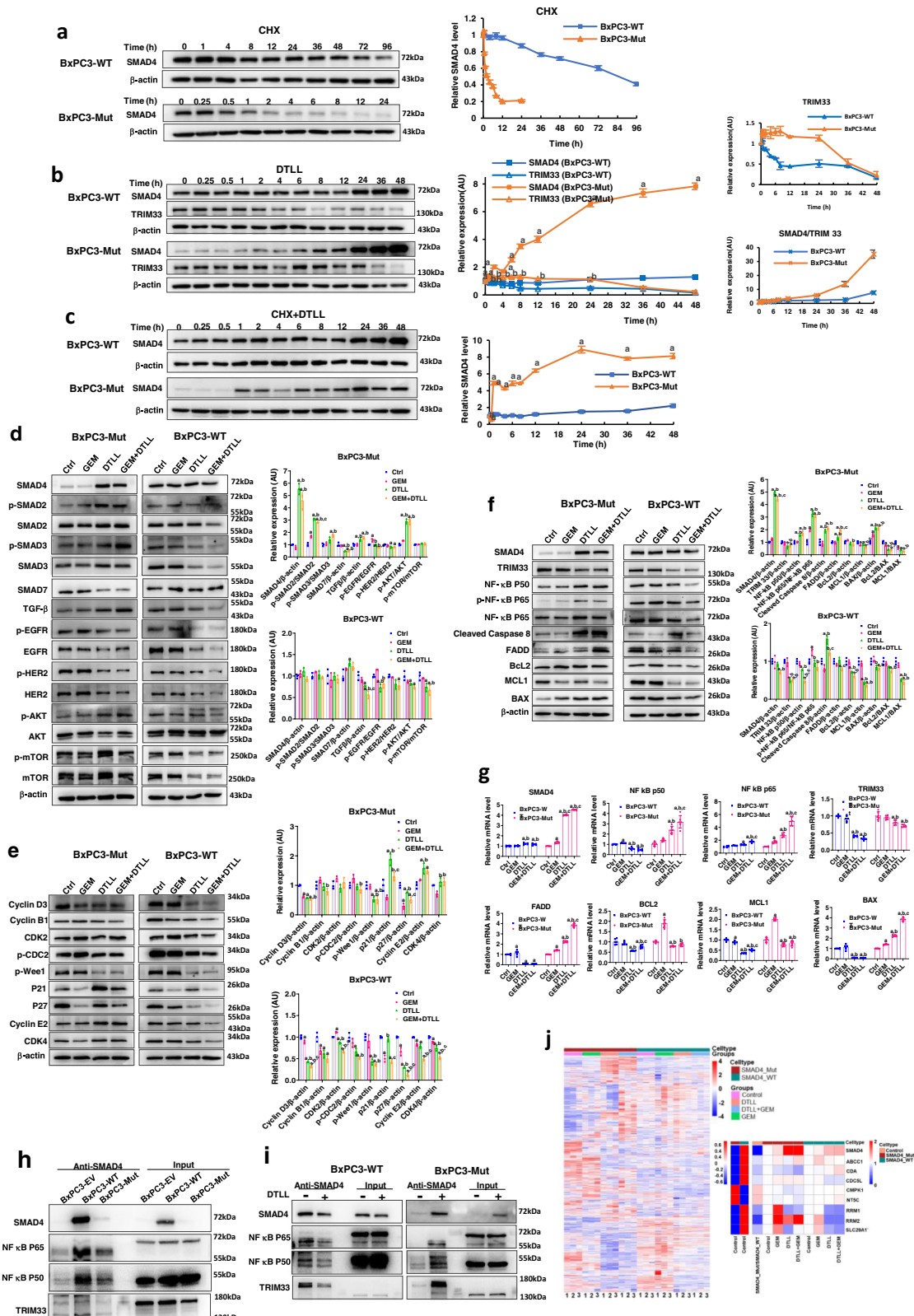

antiapoptotic proteins (Bcl-2 and MCL1) of the BxPC3-WT line (Fig. 8d–g) through the regulation of upstream NF-κB transcriptional activity. Moreover, NF-κB expression was controlled as a target of either mutant or wild-type SMAD4, and the interaction of those two proteins in BxPC3-Mut and BxPC3-WT cell lines exhibited differently in an opposite direction (Fig. 8h, i). The trends in protein expression affected by DTLL (including NF-kB, TRIM33 and apoptosis-relevant

proteins) were mainly attributed to SMAD4 protein levels based on its genetic status. In addition, *SMAD4* genetic status significantly impacted the difference in proteomic profiles between BxPC3-Mut and BxPC3-WT cells (Fig. 8j). RRM2 protein expression was obviously reduced along with decreased mutant SMAD4 in BxPC3-Mut cells, which is also partially responsible for gemcitabine resistance. DTLL alone or its combination treatment significantly triggered an increase

**Fig. 8 | Different mechanisms of the gemcitabine response in BxPC-3 cells affected by wild-type and mutant SMAD4 along with DTLL. a** The protein levels and half-life times of wild-type and mutant SMAD4 in BxPC3-WT and BxPC3-Mut stably transfected cells. Data are shown as the mean ± SD from three biologically independent samples ($n = 3$). **b** Different SMAD4 and TRIM33 levels of mutant and wild-type SMAD4 with DTLL induction in BxPC3-WT and BxPC3-Mut stably transfected cells. Data are shown as the mean ± SD from three biologically independent samples ($n = 3$). 'a' indicates $p < 0.05$ with comparison SMAD4 of BxPC3-Mut versus BxPC3-WT cells, 'b' represents $p < 0.05$ with comparison TRIM33 of AsPC-1 versus MIA PaCa-2 cells. When treated with DTLL, $p < 0.001$ for SMAD4 at all time points except for 0, 0.5, and 1 h, $p < 0.001$ for TRIM33 at 1, 2, 6, 8, 12, and 24 h. **c** Distinct protein levels and half-life times of mutant and wild-type SMAD4 with DTLL induction. Data are shown as the mean ± SD from three biologically independent samples ($n = 3$). 'a' indicates $p < 0.05$ with comparison of BxPC3-Mut versus BxPC3-WT cells. When treated with CHX + DTLL, $p < 0.001$ for SMAD4 at all time points except for 0, 0.25 and 0.5 h. **d** Different cell signaling pathways affected by mutant and wild-type SMAD4 with DTLL induction. Data are shown as the mean ± SD from three biologically independent samples ($n = 3$). In BxPC3-Mut cells, $p < 0.001$ ('GEM + DTLL' versus Control) for SMAD4, p-SMAD2/SMAD2, SMAD7, TGFβ and p-AKT/AKT; $p < 0.001$ ('GEM + DTLL' versus GEM) for SMAD4, p-SMAD2/SMAD2, TGFβ, p-EGFR/EGFR and p-AKT/AKT; $p < 0.001$ (DTLL versus Control) for SMAD4, p-SMAD2/SMAD2, SMAD7 and p-AKT/AKT; $p < 0.001$ (DTLL versus GEM) for SMAD4, p-SMAD2/SMAD2, p-EGFR/EGFR and p-AKT/AKT. In BxPC3-WT cells, $p < 0.001$ ('GEM + DTLL' versus Control) for TGFβ; $p < 0.001$ (DTLL versus GEM) for p-EGFR/EGFR. **e** Cell cycle-related proteins were determined by Western blot assays in BxPC3-Mut and BxPC3-WT stably transfected cells after treatments with 2 μM gemcitabine, 0.1 nM DTLL or its combination at 37 °C for 4 h. Data are shown as the mean ± SD from three biologically independent samples ($n = 3$). In BxPC3-Mut cells, $p < 0.001$ ('GEM + DTLL' versus Control) for Cyclin D3 and p27; $p < 0.001$ (DTLL versus Control) for Cyclin D3 and p21; $p < 0.001$ (DTLL versus GEM) for p21 and p27. In BxPC3-WT cells, $p < 0.001$ ('GEM + DTLL' versus Control) for Cyclin D3, Cyclin B1, CDK2, p-CDC2, p-Weel, p21, p27, Cyclin E2 and CDK4; $p < 0.001$ ('GEM + DTLL' versus GEM) for Cyclin D3, CDK2, p-Weel, p27, Cyclin E2 and CDK4; $p < 0.001$ ('GEM + DTLL' versus DTLL) for CDK2, p21, Cyclin E2 and CDK4; $p < 0.001$ (DTLL versus Control) for Cyclin D3, CDK2, p-Weel, p27, Cyclin E2 and CDK4; $p < 0.001$ (DTLL versus GEM) for Cyclin D3, CDK2, p21and p27; $p < 0.001$ (GEM versus Control) for CDK2, p21, p27and Cyclin E2. **f** TRIM33, NF-κB and apoptosis-related proteins were determined by Western blot assays in BxPC3-Mut and BxPC3-WT stably transfected cells after treatment with gemcitabine and DTLL. Data are shown as the mean ± SD from three biologically independent samples ($n = 3$). In BxPC3-WT cells, $p < 0.001$ ('GEM + DTLL' versus Control) for SMAD4, NF-kB p50, p-NF-kB p65/NF-kB

p65, Cleaved Caspase 8, FADD, MCL1, BAX, BCL2/BAX and MCL1/BAX; $p < 0.001$ ('GEM + DTLL' versus GEM) for SMAD4, NF-kB p50, p-NF-kB p65/NF-kB p65, Cleaved Caspase 8, FADD; $p < 0.001$ ('GEM + DTLL' versus DTLL) for SMAD4 and FADD; $p < 0.001$ (DTLL versus Control) for SMAD4, p-NF-kB p65/NF-kB p65, Cleaved Caspase 8, BAX, BCL2/BAX and MCL1/BAX; $p < 0.001$ (DTLL versus GEM) for SMAD4, p-NF-kB p65/NF-kB p65 and Cleaved Caspase 8; $p < 0.001$ (GEM versus Control) for p-NF-kB p65/NF-kB p65 and MCL1/BAX. In BxPC3-WT cells, $p < 0.001$ ('GEM + DTLL' versus Control) for TRIM33, NF-kB p50, BCL2, MCL1, and MCL1/BAX; $p < 0.001$ ('GEM + DTLL' versus GEM) for NF-kB p50, BCL2, MCL1 and MCL1/BAX; $p < 0.001$ ('GEM + DTLL' versus DTLL) for NF-kB p50; $p < 0.001$ (DTLL versus Control) for Cleaved Caspase 8, MCL1, and MCL1/BAX; $p < 0.001$ (DTLL versus GEM) for Cleaved Caspase 8 and MCL1. **g** The mRNA levels of *SMAD4, TRIM33, P50, P65, FADD, Bcl-2, MCL1,* and *BAX* were determined in BxPC3-Mut and BxPC3-WT cells after treatments with gemcitabine, DTLL and their combination by using qRT-PCR. Data are shown as the mean ± SD from three biologically independent samples ($n = 3$). In BxPC3-WT cells, $p < 0.001$ ('GEM + DTLL' versus Control) for NF-kB p50, NF-kB p65, TRIM 33, FADD, BcL2, MCL1, and Bax; $p < 0.001$ ('GEM + DTLL' versus GEM) for NF-kB p50, NF-kB p65, TRIM 33, MCL1and Bax; $p < 0.001$ ('GEM + DTLL' versus DTLL) for NF-kB p65 and MCL1; $p < 0.001$ (DTLL versus Control) for SMAD4, NF-kB p50, NF-kB p65, TRIM 33, BcL2, MCL1, and Bax; $p < 0.001$ (DTLL versus GEM) for SMAD4, NF-kB p50, TRIM 33, BcL2, MCL1, and Bax. In BxPC3-Mut cells, $p < 0.001$ ('GEM + DTLL' versus Control) for SMAD4, NF-kB p50, NF-kB p65, TRIM 33, FADD and Bax; $p < 0.001$ ('GEM + DTLL' versus GEM) for SMAD4, NF-kB p50, NF-kB p65, TRIM 33, FADD, BcL2, MCL1, and Bax; $p < 0.001$ ('GEM + DTLL' versus DTLL) for SMAD4, NF-kB p65, FADD, and Bax; $p < 0.001$ (DTLL versus Control) for SMAD4, NF-kB p50, NF-kB p65, FADD, and Bax; $p < 0.001$ (DTLL versus GEM) for SMAD4, NF-kB p50, NF-kB p65, FADD, BcL2, MCL1, and Bax; $p < 0.001$ (GEM versus Control) for SMAD4, FADD, BcL2, MCL1, and Bax. **h** Specific interactions of SMAD4 with NF-κB and TRIM33 in BxPC3-EV, BxPC3-WT, and BxPC3-Mut stably transfected cells. The results were obtained from three biologically independent experiments. **i** Different protein interactions of mutant or wild-type SMAD4 with DTLL induction in BxPC3-WT and BxPC3-Mut stably transfected cells. The results were obtained from three biologically independent experiments. **j** Different proteomic and gemcitabine pharmacokinetic profiles in mutant and wild-type SMAD4 cells after treatment with GEM, DTLL or both. Data are shown as the mean ± SD from three biologically independent samples ($n = 3$). For **d**–**g**, 'a' indicates $p < 0.05$ as compared with the control, 'b', $p < 0.05$ compared with the GEM group and 'c', $p < 0.05$ compared with the DTLL group. One-way ANOVA with Bonferroni post hoc test was used in **d**–**g**; Paired-samples $t$-test was two-sided and used in **b** and **c**. All specific $p$ values are presented in the Source Data file and only significant values are shown in figures. Source data are provided in the Source Data file.

in RRM2 expression, interpreting its mechanism of action on enhanced BxPC3-Mut cellular susceptibility mediated by increased SMAD4.

Consequently, all the above findings suggested that DTLL is capable of sensitizing either gemcitabine-resistant or gemcitabine-sensitive cells to gemcitabine efficacy (or even other chemotherapeutic agents) through distinct action mechanisms according to different SMAD4 profiles. As shown in the schematic illustration model (Fig. 9), the mechanistic studies implied that DTLL combinational treatment might not only prevent neoplastic proliferation via blockage of ATK/mTOR signaling and anti-apoptotic proteins (Bcl-2 and MCL1) mediated by impaired NF-κB function if given to SMAD4-sufficient/gemcitabine-sensitive PDAC cells, but also restore the bioactivity of SMAD4 as a tumor suppressor to trigger its downstream NF-κB-regulated signaling of cell apoptosis in SMAD4-deficient/gemcitabine-resistant tumors. The inhibitory effectiveness of DTLL alone or in combination with gemcitabine revealed that it functions as a double-edged sword in PDAC tumor growth, with totally different drug responsive behaviors and molecular action mechanisms on synergistic efficacy between gemcitabine-resistant and gemcitabine-sensitive PDAC. We speculate that DTLL seems to act as a module of the SMAD4 driver that normalizes its function as a tumor suppressor according to SMAD4 protein level and genetic status in PDAC, and then adjusts NF-κB, TRIM33, gemcitabine-metabolic enzymes (such as RRM2) and proteomic profiles of PDAC cells, which explains the synergistic effect on tumor growth of DTLL in combination with

gemcitabine. Further deeper mechanistic studies are necessary to verify our findings in the future.

In the current precision medicine treatment era, diverse genetic alterations in cancer subclones with unique hydrogenous signatures are responsible for the therapeutic resistance of pancreatic cancers as one of the biggest challenges. The most common mutations in PDAC occur in *KRAS, TP53, CDKN2A,* and *SMAD4* driver genes[57–60]. In contrast to other oncogenes (such as mutated BRAF or EGFR), effective therapies that directly target those four driver genes are still unavailable in PDAC. Consequently, there is an urgent need to develop biomarkers or therapeutic approaches for the selection of adaptive PDAC patients. Given the critical effect of p53-related pathways on pancreatic cancers, reactivation of p53 has been investigated to sensitize tumors to chemotherapy in a number of preclinical studies[61–64]. A study on cell lines suggested that the mutant p53 activator, PRIMA-1, accelerated apoptosis and sensitized tumor cells to chemotherapy[65]. The combination of p53 transduction by Ad5CMV-p53 with two genotoxic drugs (gemcitabine and cisplatin), under a correct schedule of administration, appears to be a very promising therapy for human pancreatic cancer[66]. Researchers recently revealed a p53-dependent interaction between cancer cells and CAFs during the response to gemcitabine/abraxane, also suggesting that combining biomarkers of GOF (gain of function) mutant p53 with high HSPG2 stromal deposition may be used in the future to identify patients[67]. Undoubtedly, activation of tumor suppressors for the treatment of human cancer has been a long sought, yet

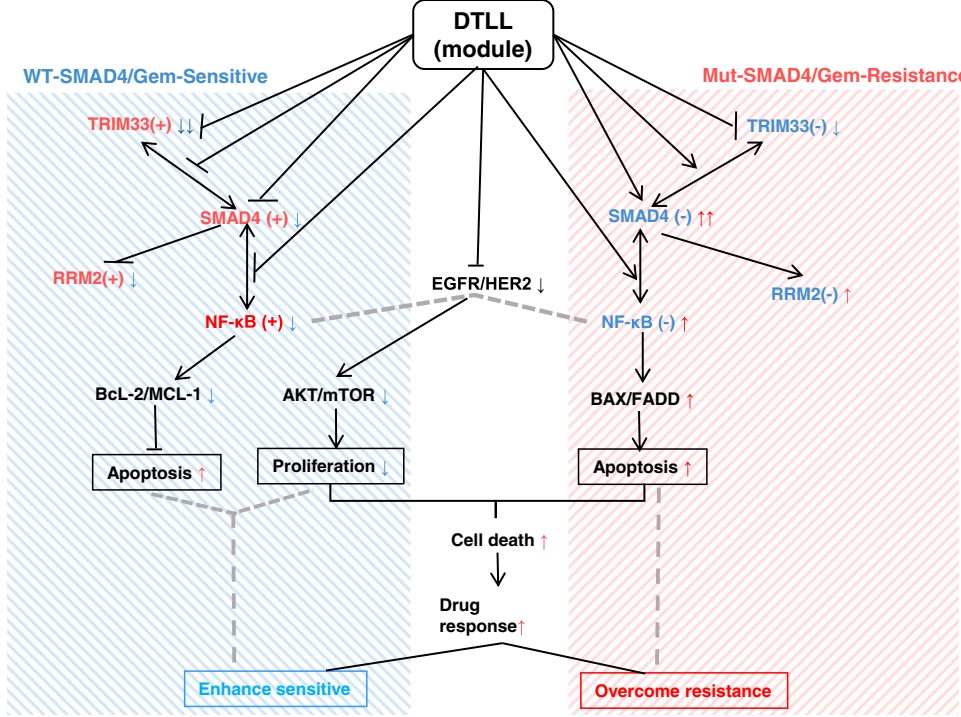

**Fig. 9 | Schematic illustration model of the DTLL action mechanism along with mutant and wild-type SMAD4 in PDAC cells.** There were different action mechanisms of mutant and wild-type SMAD4 on the PDAC drug response with/without DTLL induction. The SMAD4 genetic status of PDAC is responsible for SMAD4 protein levels which determine different cellular susceptibilities. DTLL seems to act as a module of the SMAD4 driver that normalizes its function as a tumor suppressor according to SMAD4 level and genetic status in PDAC, and then adjusts NF-κB, TRIM33, gemcitabine-metabolic enzymes (such as RRM2) and proteomic profiles of PDAC cells, which explains the synergistic effect on tumor growth of DTLL in combination with gemcitabine. Specifically, DTLL contributes to decreases mainly in AKT/mTOR signaling and anti-apoptotic proteins (Bcl-2 and MCL1) by impaired NF-κB function in WT-SMAD4/GEM-sensitive PDAC cells.

However, DTLL predominantly enhances the expression of apoptotic BAX and FADD by restoring SMAD4 bioactivity to trigger downstream NF-κB-regulated signaling in Mut-SMAD4/GEM-resistant cells. Moreover, NF-κB expression was regulated as a SMAD4 target in these two types of cells, and the interaction between these two proteins was different in an opposite direction. In addition, SMAD4 status significantly impacted the proteomic profiles, especially RRM2 expression. As a result, DTLL in combination with chemotherapeutic agents shows synergistic effects by either enhancing sensitivity or overcoming resistance in PDAC cells via wild-type or mutant SMAD4-mediation, respectively. Note: WT indicates wild type and Mut represents mutant (R100T mutation) SMAD4. (+), sufficient expression; (−), deficient expression.

elusive, strategy. A recent study also provided an example of a "tumor suppressor reactivation" approach. They found that the tumor suppressor PTEN was reactivated for cancer treatment through inhibition of the MYC-WWP1 pathway, and further identified a derivative of cruciferous vegetables (I3C) as a potent WWP1 inhibitor via PTEN reactivation, leading to suppression of tumorigenesis[68]. Our findings with a similar therapeutic approach for PDAC might provide important insight into clinical implementation via SMAD4 reactivation induced by DTLL combination treatment. Moreover, regardless of whether SMAD4 expression is sufficient, the synergistic effect of DTLL combination treatment on PDAC might enhance the therapeutic benefits of gemcitabine for patients. Furthermore, our findings may expand the translational avenue for cancer treatments. A previous study demonstrated a precision treatment strategy for EZH2-aberrant tumors that was based on tumor-intrinsic MLL1 expression and concurrent inhibition of epigenetic crosstalk and feedback MAPK activation[69]. For clinical application in the future, our approach might also require biomarkers to identify possible PDAC responders based on tumor-intrinsic SMAD4 status for rational combination regimens with either AKT or TGF-β inhibitors.

Inevitably, there are several limitations in our study. First, we investigated the role of SMAD4 in mediating the cellular response in both SMAD4-deficient and SMAD4-sufficient PDAC. Mechanistically, we only studied the EGFR, AKT, NF-kB, and SMAD4-mediated signaling pathways, but more key drivers (such as PTEN, TP53, and KRAS) and

their potential crosstalk need to be further investigated. Second, DTLL or the combination treatment increased the expression of PD-L1 in mutant SMAD4 tumors but reduced its expression in wild-type tumors, which might promote or inhibit immune escape in vitro and in vivo, respectively. However, whether the exact mechanisms of tumor immune escape and SMAD4-mediated activation or inhibition occur is unclear, which needs further investigation. Third, based on differences in SMAD4 expression and its genetic status, distinct molecular mechanisms were observed. More functional studies are needed to evaluate and characterize the effects of the combination treatment on autophagy, angiogenesis, and epithelial-mesenchymal transition (EMT), in addition to the influences on cell proliferation, apoptosis and transcriptional activity. CRIPS/Cas9 depleted cells or transgenic mouse models would be more helpful in the future to further confirm and deeply interpret SMAD4 roles in PDAC chemoresistance. Furthermore, SMAD4 might regulate its downstream genes via a SMAD4-miRNA-downstream gene axis (such as AKT, E-cadherin, and Vimentin)[56], further influencing PDAC. Fourth, due to financial limitations, we chose only three to four mice in the PDX models, as well as BxPC-3 stably transfected cell lines or CDX models for functional characterization instead of PDX models. More cell lines or PDX models need to be further selected. In addition, deeper investigation is necessary to fully elucidate the molecular mechanisms of SMAD4 in the functional regulation of NF-κB-mediated signaling, TRIM33-mediated self-control of protein degradation and the RRM2-relevant pharmacokinetic pathway

in PDAC progression and drug resistance. More specifically, further functional studies such as splicing regulation with both *trans-* and *cis*-regulatory effects, need to be investigated. We demonstrated that SMAD4 status significantly impacted the proteomic profiles of BxPC3-Mut/gemcitabine-resistant and BxPC3-WT/gemcitabine-sensitive cells. The crosstalk between cancer cells and the tumor microenvironment plays an important role in resistance to therapy. More studies need to be conducted to determine whether SMAD4-mediated chemo-sensitization is involved in the tumor microenvironment or immunodeficiency. Finally, clinical validation to identify the molecular profiles of SMAD4 and its relevant proteins and to predict the drug response of PDAC patients is essential in the future.

Taken together, we demonstrated the synergism of gemcitabine and DTLL in both gemcitabine-sensitive and gemcitabine-resistant PDAC and identified the critical role of SMAD4 in mediating the cellular responses of pancreatic cancer to chemotherapy. SMAD4 genetic status is responsible for SMAD4 protein levels, and its R100T mutation contributes to loss of SMAD4 protein and function with rapid protein degradation, leading to resistance to gemcitabine of PDAC cells. DTLL altered the protein half-life time and level of mutant and wild-type SMAD4 by inhibiting protein degradation at different velocities and changing the interaction of SMAD4 with TRIM33 distinctly. Further studies implied that DTLL combinational treatment might not only prevent neoplastic proliferation via blockage of ATK/mTOR signaling and anti-apoptotic proteins mediated by impaired NF-κB function in SMAD4-sufficient/gemcitabine-sensitive PDAC cells, but also restore the bioactivity of SMAD4 as a tumor suppressor to trigger its downstream NF-κB-regulated signaling of cell apoptosis and enhance the gemcitabine-related enzyme RRM2 in SMAD4-deficient/gemcitabine-resistant tumors. Our findings may reveal a potential combination therapy for overcoming resistance of PDAC mediated by SMAD4, which paves the way toward a long-sought tumor suppressor reactivation approach and lays the foundation for future precision medicine as the first step.

## Methods

### Cell lines and plasmids
Human pancreatic carcinoma cell lines AsPC-1, MIA PaCa-2, BxPC-3, PANC-1, CFPAC-1, Panc0403, HuPT-3, and SU86.86 were obtained from Dr. Liewei Wang (Department of Molecular Pharmacology of Experimental Therapeutics, Mayo Clinic). AsPC-1, BxPC-3, HuPT-3, and SU86.86 cells were cultured in RPMI 1640 (Gibco, Life Technologies) supplemented with 10% fetal bovine serum (FBS, Gibco, Life Technologies), 100 U/ml penicillin, and 100 μg/ml streptomycin. MIA PaCa-2, CFPAC-1, Panc0403 and PANC-1 cells were cultured in DMEM under the same conditions. The cells were maintained in a humidified incubator at 37 °C and with 5% $CO_2$.

The pcDNA3-Flag-SMAD4 plasmid (#80888) for overexpressing wild-type SMAD4, and an empty FNpCDNA3 vector (#45346) were purchased from Addgene (Cambridge, MA, USA).

### Driver mutation analysis
Eight human pancreatic carcinoma cell lines were used to prepare for genomic DNA extraction according to the manufacturer's instructions (Tiangene Biotech Co., Beijing) Specifically, a panel that contains 207 primers that target 50 tumor genes and tumor suppressor genes for hotspot mutations in tumors was designed (Nuoshai Biotech Co., Beijing). DNA library preparation was performed by two-step PCR in a thermal instrument (BIO-RAD, T100TM) followed by quantification and paired-end sequencing on HiSeqX Ten sequencers (Illumina, San Diego, CA). Raw reads were filtered by removing adaptor sequences if reads were contained by cutadapt (v 1.2.1) and low quality bases from reads 3' to 5' ($Q < 30$) by PRINSEQ-lite (v 0.20.3). After data QC and SNP calling, the remaining clean data were mapped to the reference genome by BWA (version 0.7.13-r1126) with default parameters. A Perl

script was written to calculate each genotype of the target site. Annovar (2018-04-16) was used to detect genetic variants.

### Preparation of DTLL
We prepared DTLL according to a previous approach (Patent Publication No. CN101497666A). The producer clones of DTLP (the precursor of DTLL) fusion protein in this study have been deposited it into the China Pharmaceutical Culture Collection (CPCC) with the Accession Number CCPC 101501 (http://www.cpcc.ac.cn/). DTLL is a bispecific fusion protein consisting of ligand-based and antibody-based oligopeptides against EGFR and HER2, and an enediyne antibiotics lidamycin[25].

### MTS cell proliferation colorimetric assay
Cells were seeded into 96-well plates at 5000 cells/well, and treated with gemcitabine, DTLL or both at 37 °C for 72 h. Cell viability was measured by the MTS (Cell Titer 96 AQueous One Solution Cell proliferation assay, Promega Corporation, USA) assay according to the manufacturer's manuals. The assay was performed in triplicate. Cell viability was detected to create drug-response curves and CIs were calculated by the Chou-Talalay method[70,71].

### CyQUANT cell viability assay
In parallel to the above MTS assay, cell proliferation was confirmed using a CyQUANT® NF cell proliferation assay kit (Thermo Fisher Scientific, Waltham, MA, USA) according to the manufacturer's protocol. This assay is based on the measurement of cellular DNA content via fluorescent dye binding. Cells were seeded in a 96-well plate (3000 cells/well) and then exposed to a series of concentrations of GEM, DTLL, or GEM + DTLL ($10^{-14}$ mol/L) for 72 h. Culture media was removed and 100 uL of the CyQUANT working solution was applied to the cells and incubated for 60 min at 37 °C. The sample fluorescence was measured using a multimodal microplate reader (SpectraMax iD5, Molecular Devices) at 480 nm excitation and 520 nm emission. The assay was performed in triplicate. Cell viability was detected to create drug-response curves and CIs were calculated by the Chou-Talalay method[70,72].

### Calculation of the combination index (CI)
After analysis of cell proliferation, the combination index (CI) at each drug concentration was calculated by using the Chou-Talalay method. The equation for CI calculation of Chou-Talalay is as follows:

$$CI = (D1/Dm1) + (D2/Dm2) + \alpha(D1*D2/Dm1*Dm2)$$

The combination index (CI) concept was introduced initially by Chou T.C. and Talalay P[70,72]. The derived combination index equation for the two drugs is:

$$CI = (D)1/(Dx)1 + (D)2/(Dx)2$$

where (Dx) 1, (Dx) 2 = the concentration of the tested substance 1 and the tested substance 2 used in the single treatment that was required to decrease the cell number by x% and (D) 1, (D) 2 = the concentration of the tested substance 1 in combination with the concentration of the tested substance 2 that together decreased the cell number by x%. The CI value quantitatively defines synergism (CI < 1), additive effect (CI = 1) and antagonism (CI > 1).

### Cell cycle arrest
After treatment with 0.02 μM gemcitabine, 0.1 nM DTLL or their combination for 24 h, AsPC-1 and MIA PaCa-2 cells were fixed with pre-cold 70% ethanol at 4 °C overnight and washed with cold PBS. Then the cells were incubated with 100 μg/ml RNase A (Beyotime Technologies,

Shanghai, China) and 50 µg/ml PI at 37 °C for 30 min, and the cell-cycle distribution was analyzed on a FACScan flow cytometer.

### Immunofluorescence
After being treated with 2 µM gemcitabine, 0.1 nM DTLL and both at 37 °C for 4 h, cells were fixed with 4% paraformaldehyde for 30 min and permeabilized with 0.5% Triton X-100. Then the cells were blocked with 1% BSA for 1 h and incubated with anti-SMAD4 (1:800, Cell Signaling Technology) at 4 °C overnight. Cells were incubated with Alexa Fluor-conjugated secondary antibodies (1:500, Invitrogen) and 5 µg/ml DAPI (Sigma-Aldrich Inc.) for 30 min. Then, the cells were observed with a laser scanning confocal microscope.

### Transfection with plasmid or specific siRNA
For plasmid transfection, cells were transfected with the aforementioned empty (pcDNA3-Flag) or SMAD4 overexpression (pcDNA3-Flag-SMAD4) vector using Lipofectamine 2000 transfection reagent (Invitrogen, Carlsbad, California, USA) following the protocols of the manufacturer.

Specific siRNAs against SMAD4 were purchased from RiboBio (Guangdong, China) together with a scrambled siRNA. For siRNA transfection, cells were transfected with a pooled sample of four SMAD4-specific siRNA oligos at a final concentration of 10 nmol/L using Lipofectamine™ RNAiMAX (Invitrogen, Carlsbad, California, USA) as the transfection reagent according to the manufacturer's instructions. A scrambled siRNA was applied as a negative control. Approximately 6 h after transfection, the medium was replaced and the cells were further cultured for 48 h for gene expression analysis by Western blot and qRT-PCR assays. The oligonucleotide sequences of four SMAD4-specific siRNAs and scramble siRNA are supplied in Supplementary Table 6.

### Site-directed mutagenesis
Plasmids for human SMAD4 overexpression with a point mutant (R100T) were constructed with a wild-type SMAD4 plasmid (PcDNA3-Flag-SMAD4, Addgene) as a DNA template by using a fast mutagenesis system (TransGen, China) according to the manufacturer's instructions. The resulting mutant SMAD4 plasmid was confirmed by sequencing (Shenggong, Shanghai, China) to verify the presence of the introduced mutations and the absence of additional unwanted mutations.

### Stable transfection for SMAD4 overexpression
Lentivirus vectors were used to establish stably transfected cell lines. Lentiviruses were produced in HEK293T cells for the stable transfection of the cell lines in accordance with the manufacturer's instructions. The complete coding sequence of the human wild-type SMAD4 or mutant SMAD4 (R100T) gene from the aforementioned mutant or wild-type SMAD4 overexpression vector was cloned into pLV-EF1-CMV- EGFP/T2A/Puro vectors. An empty vector was transfected as control. A total of $3 \times 10^5$ BxPC-3 cells in 1 mL of medium with 5 µg/mL polybrene were infected with 1 mL of lentivirus supernatant. After 72 h, 2 µg/ml puromycin was used for selection of a stable cell pool. The efficiency of overexpression in a stable cell line was validated by qRT-PCR and Western blot assays.

### Real-time quantitative reverse transcription-PCR (real-time qRT-PCR)
Cells were trypsinized, washed in PBS, and collected with centrifugation at $1000 \times g$ for 2 min. Total RNA was extracted using an RNeasy Mini Kit (Qiagen, Valencia, CA) according to the manufacturer's protocol. RNA concentration and purity were estimated by Nano-300 Micro-Spectrophotometer (Allsheng Co.,Ltd, Hangzhou, China). cDNA was synthesized with a Transcriptor First Strand cDNA Synthesis Kit (Roche, Mannheim, Germany). Real-time qRT-PCR was conducted with a SYBR® Premix Ex Taq™ II kit (TaKaRa, Kusatsu, Japan) according to the manufacturer's recommendations. mRNA expression was quantified using the $2^{-\Delta\Delta Ct}$ method. GAPDH served as an internal control. The PCR conditions were as follows: 42 °C for 5 min, 95 °C for 10 s followed by 40 cycles of 95 °C for 5 s and 60 °C for 34 s. The primers used for real-time qRT-PCR are listed in Supplementary Table 7.

### Western blot assay
Cells or tissue samples were processed and lysed in RIPA buffer containing protease inhibitors (Sigma-Aldrich, Carlsbad, CA, USA). Equal amounts of protein were loaded in each lane of an SDS−8−12% polyacrylamide gel. Western blot assays were performed by the standard method using the following primary antibodies: phosphorylated EGFR, EGFR, phosphorylated HER2, HER2, phosphorylated AKT, AKT, phosphorylated mTOR, mTOR, phosphorylated ERK1/2, ERK1/2, PD-L1, phosphorylated Bcl-2, Cyclin D1, phosphorylated γH2AX, phosphorylated SMAD2, SMAD2, phosphorylated SMAD3, SMAD3, SMAD4, TGF-β, TPIM33, Cyclin D3, Cyclin B1, Cyclin E2, CDK2, CDK4, P50, phosphorylated P65, P65, phosphorylated CDC2, phosphorylated Wee1, P21, P27, Cleaved Caspase 8, FADD, BCL-2, MCL1, BAX and β-actin were purchased from Cell Signaling (Danvers, MA, USA). SMAD7 and Lamin B were purchased from Proteintech (Rosemont, IL, USA). The protein levels were quantified using ImageJ software.

### Co-immunoprecipitation (Co-IP) assay
Cells with/without DTLL induction were washed with PBS and lysed with cell lysis buffer (Cell Signaling Technology, Beverly, MA) containing protease inhibitors on ice for 30 min. After centrifugation at $12,000 \times g$ for 15 min at 4 °C, the supernatant was collected and incubated with SMAD4 antibody (w/w, antibody: total protein = 1:500) or rabbit normal IgG control at 4 °C overnight with rotation, and then further incubated with 20 µL of prewashed protein A/G agarose beads at 4 °C for 2 h. After incubation, the immune complexes were washed with lysis buffer three times, and the proteins were eluted by the addition of SDS sample buffer and incubated at 100 °C for 5 min. These eluates were subjected to Western blot analysis using primary antibodies as mentioned above.

### Nuclear and cytoplasmic protein extractions
After treatment, the cytoplasmic and nuclear protein fractions were obtained using nuclear and cytoplasmic extraction kits (Thermo Scientific) according to the manufacturer's instructions. Briefly, after treatment with 2 µM gemcitabine and/or 0.1 nM DTLL for 4 h, cells were harvested and washed, followed by the addition of ice-cold cytoplasmic extraction reagents (CER I and CER II). Then the samples were centrifuged, and the supernatant was collected to obtain the cytoplasmic fraction. The nuclei were resuspended in nuclear protein extraction agent (NER). After centrifugation, the supernatant was collected as the nuclear fraction.

### In vivo therapeutic efficacy
We used both pancreatic cancer cell line-derived xenograft (CDX) and patient-derived xenograft (PDX) mouse models for the in vivo evaluation of the combination of gemcitabine with DTLL efficacy. PDX models were obtained from the HuPrime® allograft PDX platform of Crown Bioscience Inc (Beijing, China) in which RNA sequencing datasets were available for each selected mouse model. The PDX models were developed and established in immune-deficient mice in Crown Bioscience, under the approval of the Institutional Review Boards of the hospitals and the informed consent from patients. Five-week-old female BALB/c nude mice (Huafukang Bioscience Co, Beijing, China) were subcutaneously inoculated with AsPC-1, MIA PaCa-2, and BxPC-3 cells with stable expression of wild-type and mutant SMAD4 or human pancreatic tumor tissue samples. All experiments were conducted under specific pathogen-free (SPF) conditions. Mice were housed in

groups of 5 mice per individually ventilated cage in a 12 h light/dark cycle, with a temperature of $20 \pm 2\,°C$) and relative humidity of $55\% \pm 15$. All mice had access to food and water ad libitum. When tumors reached 150–180 mm³ in volume, mice were treated with gemcitabine, DTLL and both in CDX models and PDX models. Mice were weighed, and tumors were measured every three or every four days. At the end of the experiments, the mice were sacrificed, and then tumors plus major organs, including the heart, liver, spleen, lung and kidney, were examined histologically. Animal protocols were approved by the Institutional Animal Care and Use Committee of the Institute of Medicinal Biotechnology (IMB), Chinese Academy of Medical Sciences & Peking Union Medical College (CAMS & PUMC).

### TUNEL assay

Paraffin-embedded sections of tumors were fixed in 4% paraformaldehyde and dewaxed. Then the TUNEL assay (Beyotime Biotechnology, China) was performed according to the instructions of the manufacturer. Finally, the slides were visualized by fluorescence microscopy.

### Immunohistochemistry (IHC)

Tumor sections of AsPC-1, MIA PaCa-2, PA1233, and PA3142 were deparaffinized, rehydrated, and treated with citrate buffer to retrieve antigens. Subsequently, sections were incubated with Ki-67 antibody (1:400 diluted, Abcam, Cambridge, UK) at $4\,°C$ overnight, followed by incubation with secondary antibody. The results were observed by an inverted microscope using DAB as a chromogenic reagent.

### Proteomic analysis

The proteomic analysis of the cell samples was conducted using the combination of DIA and a data-dependent acquisition (DDA)-based ion library. After treatment with 2 µM gemcitabine, 0.1 nM DTLL and both at $37\,°C$ for 4 h, cells were harvested and lysed with lysis buffer (8 M urea, 100 mM Tris-HCl pH 7.6, 1 mM PMSF), and then centrifuged at $18,000 \times g$ for 15 min at $4\,°C$. The extracted protein in the supernatant was quantified using a BCA protein assay kit (Beyotime Biotechnology, China) and digested in trypsin (Promega, Madison, WI) after reduction and alkylation using the FASP (filter aided sample preparation) method. The concentration of digested peptides was determined by measuring the absorbance at 280 nm using a NanoDrop 2000 instrument (Thermo Fisher Scientific, Waltham, MA, USA). An EASY-nLC 1200 chromatography system and an Q Exactive HF mass spectrometer (Thermo Fisher Scientific, Waltham, MA, USA) were used for mass spectrometry acquisition and analysis. The DIA analysis was performed according to a previous study[71]. All results were filtered by a Q-value cutoff of 0.01 (corresponding to an FDR of 1%). The $P$-value estimator was performed by the Kermel Density Estimator. The area was used for protein quantification. Every peptide was validated with at least three fragments.

### Bioinformatics analysis

We downloaded a file (TCGA-PAAD.htseq_fpkm.tsv) from the TCGA-PAAD Project of TCGA public datasets on the website (https://portal.gdc.cancer.gov/repository) including clinical information on gemcitabine treatment and SMAD4 expression data of pancreatic cancer patients to determine whether SMAD4 expression is clinically correlated with gemcitabine response. Among the 108 patients, 69 samples had a gemcitabine response available. Responsive data were defined as four different types of gemcitabine response data: complete response, partial response, stable disease and clinical progressive disease. First, we analyzed the relationship between SMAD4 expression and gemcitabine response. Second, we plotted the ROC (receiver operating characteristic) curve evaluated with an AUC (area under the curve) and took 2.27 (median expression of all the PAAD patients) as the threshold for SMAD4 expression to define the high (>2.27) or low (<2.27) level,

followed with Fisher's exact test for calculation of the odds ratio and $p$-value.

### Statistical analysis

Data are presented as the mean ± standard deviation (SD). Statistical analysis was performed using SPSS software. A paired-samples $t$-test with two-sided was applied for statistical analyses between two groups, and one-way ANOVA with Bonferroni post hoc test was used for multiple comparisons. A value of $p < 0.05$ was considered to be statistically significant.

### Reporting summary

Further information on research design is available in the Nature Research Reporting Summary linked to this article.

## Data availability

A reporting summary for this article is available as Supplementary Information file. The main data supporting the findings of this study are available within the article and its Supplementary Figures. The source data underlying Figs. 1a–d, 2–7 & 8a–i, Table 1, Supplementary Figs. 1–4, and Supplementary Tables 1, 2 are provided as a Source Data file. The source data underlying Fig. 1e in this study were downloaded from the TCGA-PAAD Project of TCGA public datasets on the website (https://portal.gdc.cancer.gov/repository), including clinical information on gemcitabine treatment and SMAD4 expression data of pancreatic cancer patients, which are provided in Supplementary Data 1 (TCGA-PAAD + clinical_drug) and Supplementary Data 2 (TCGA-PAAD.htseq_fpkm). All the source data of Fig. 8j and Supplementary Table 3, and the results files in this study have been deposited in the iProX Consortium database under accession code PXD031977, as provided in Supplementary data 3 (Proteomic profile). Specific data $P$ values are also included within the Source Data file. Source data are provided with this paper.

The producer clones of DTLP (the precursor of DTLL) fusion protein in this study have been deposited into the China Pharmaceutical Culture Collection (CPCC) under Accession Number CCPC101501. Additional details on datasets and protocols that support the findings of this study will be made available by the corresponding author upon reasonable request.

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

## Acknowledgements

This work was supported by the National Natural Science Foundation of China (NSFC 81472787, 81773671, 81828010), the CAMS Innovation Fund for Medical Sciences (CIFMS 2021-I2M-1-070), the Drug Innovation Major Project of China (2018ZX09711001-007-002) and the Innovation Team and Talents Cultivation Program of the National Administration of Traditional Chinese Medicine. (ZYYCXTD-C-202205). We gratefully thank all of the contributors of data to TCGA, generously providing cyber-resources for the scientists for the allowing exploration, visualization and analysis of the multidimensional genomics data of cancer. We gratefully appreciate the support for the proteomic analysis from the State Key Laboratory of Proteomics, the Beijing Proteome Research Center, the National Center for Protein Sciences (Beijing), Beijing Institute of Lifeomics, Beijing 102206, China.

## Author contributions

L.L. and S.R. conceived, designed, and supervised the study. L.L. and Y.H. wrote the paper with data acquisition and statistical analysis. Y.H., L.L., and Z.L. performed in vitro experiments. C.R., S.W., Y.C., and L.R. prepared animal experiments and IHC staining. C.H. and W.J. performed bioinformatics TCGA data and proteomic analysis. L.X. and L.Y. contributed to pathological analysis. Z.X. and S.Y. advised the analysis and helped data interpretation. All authors read and approved the final manuscript.

## Competing interests

The authors declare no competing interests.
