## [Peer Review File · Nature Communications]

Title: An EGFR/HER2-targeted conjugate DTLL sensitizes gemcitabine-sensitive/resistant pancreatic cancer through different SMAD4-mediated mechanismsReviewers' comments:

Reviewer #1 (Remarks to the Author); mouse models and PDAC therapy:

In this paper Hongjuan Yao and colleague's report that combination of DTLL with gemcitabine result is a synergic antitumoral effect in pancreatic cancer through a molecular mechanism involving SMAD4 expression.

The effect of DTLL on pancreatic cancer was previously published by the same group.

The work is not very original and the mechanistic hypothesis, particularly SMAD4 involvement in sensitivity to gemcitabine, is not adequately demonstrated. Some data presented in this paper is unnecessary and some others experiments need to be performed.

Response to reviewer's comment: We appreciate your comment. We reported the antineoplastic effect of DTLL alone on pancreatic cancer by using a human pancreatic cancer MIA PaCa-2 cell line and corresponding CDX tumor models. "We previously found that human pancreatic cancer AsPC-1 and MIA PaCa-2 cells with high EGFR/HER2 expression showed the strongest binding affinity to DTLP, the precursor of DTLL; however, AsPC-1 cell line or xenograft tumor showed intermediate resistance to DTLL whereas MIA PaCa-2 had stronger affinity and better response [25]. This raised an interesting proposal and encouraged us to explore more effective strategies and underlie relevant mechanisms especially for drug-resistant pancreatic cancers.....For instance, a kinase inhibitor specific to EGFR can sensitize tumors to gemcitabine in PDAC models [30] (Page 5, Line 118-129)." Our hypothesis in the present study was made based on distinguished effects on PDAC tumor inhibition of DTLL in combination with gemcitabine, not only regarding to DTLL effect as a single antineoplastic agent but also particularly focusing on how to re-sensitize gemcitabine efficacy in SMAD4-deficient/resistant PDAC. We hypothesized that it might be possible to improve chemotherapeutic efficacy of gemcitabine in PDAC when combined with DTLL, and indeed found that that SMAD4 deficiency was associated with gemcitabine resistance, and that DTLL in combination with gemcitabine sensitized those resistant PDAC cells to gemcitabine, followed with studies to underlay its action mechanisms.

According to the reviewer suggestion, we further performed mechanism studies to functionally characterize the effect of SMAD4 expression on gemcitabine response, and differences in DTLL action mechanism of between mutant/resistant and wild-type SMAD4/sensitive PDAC cells.

Experimentally, we demonstrated that upregulated or downregulated SMAD4 expression by its specific siRNAs or overexpression vectors in SMAD4-deficient AsPC-1 and SMAD4-sufficient MIA-PaCa-2 cells altered PADC cellular susceptibility to gemcitabine, respectively. DTLL in combination with gemcitabine potentiated inhibitory effects of gemcitabine on *in vitro* cell viability, and synergistically repressed *in vivo* tumor growth of AsPC-1 and MIA PaCa-2 CDX models. Furthermore, we generated three types of stably transfected BxPC-3 cells containing empty, mutant and wild-type SMAD4 overexpression vectors and their corresponding CDX models to investigate the effect of DTLL, and to explore its action mechanism on sensitization of gemcitabine response along

with SMAD4 genetic or expression status, respectively. The results have shed light on its therapeutic efficacy *in vitro* and *in vivo*.

“Functional studies indicated that SMAD4 genetic status is responsible for SMAD4 protein level which determines different cellular susceptibility of PDAC. R100T mutation contributes to loss of SMAD4 protein and function with rapid protein degradation, leading to resistance to gemcitabine in PDAC cells. Moreover, DTLL significantly altered the protein half-life time and level of mutant and wild-type SMAD4 by inhibiting protein degradation at different velocities and distinctly changing the interaction of SMAD4 with TRIM33. Mechanism studies implied that DTLL combinational treatment might not only prevent from neoplastic proliferation via blockage of ATK/mTOR signaling and anti-apoptotic proteins (Bcl-2 and MCL1) mediated by impaired NF- κ B function in SMAD4-sufficient/gemcitabine-sensitive PDAC cells, but also restore the bioactivity of SMAD4 as a tumor suppressor to trigger its downstream NF- κ B-regulated signaling of cell apoptosis in SMAD4-deficient/gemcitabine-resistant tumors.....In conclusion, SMAD4 is the key central mediator of not only the occurrence and development but also susceptibility in PDAC. DTLL sensitized gemcitabine efficacy via distinct action mechanisms based on SMAD4 profiles in SMAD4-sufficient/gemcitabine-sensitive and SMAD4-deficient/-resistant PDAC, respectively. Our findings provide insight into a rational SMAD4-directed precision treatment strategy and reveal a promising DTLL combination therapy to overcome chemoresistance in gemcitabine-resistant PDAC (Page 2, Line 43-61).”

Those functional experiments verified our hypothesis and we re-addressed our findings in detail in the text of Results section (Page 22, Line 548 – 587; Page 26, Line 688 – Page 33, Line 894) and Discussion sections (Page 34, Line 898 - Page 40, Line 1085). Our point-by-point responses to the comments and suggestions are provided below. All changes in our manuscript have been highlighted by using the track changes mode in MSWord. A clean version of the manuscript has also been uploaded for your reference.

Major comments:

1. It is not clear the rationale to study the expression of KRAS, TP53, CDKN2A and SMAD4 in the resistance to gemcitabine. Are the authors suggesting that genes playing a role in the tumorigenesis process of pancreatic cancer are also involved in chemo resistance? If yes, please give the arguments.

Response to reviewer’s comment: Thank you for pointing out this important question. We appreciate your comment and modified the text to clarify the rationale in the sections of Introduction, results and Discussion, as described in below. As described in the section of Introduction (Page 4, Line 81-105), “Typical genetic alterations observed in PDAC include activation of oncogene KRAS (>90%), inactivation or loss of tumor suppressor genes such as CDKN2A (95%), TP53 (50-75%), SMAD4 (~55%), PTEN (~60%) and mutation of DNA repair genes BRCA2 (7-19%), contributing to tumor progression or relapse [6-9]. Unfortunately, few of these drivers are currently druggable, thus making it difficult to devise effective therapies against PDAC. Increasingly, molecular profiling of tumor specimens is being utilized to reveal tumor susceptibilities and accelerate the development

of precision medicine. A personalized therapeutic approach targeting key drivers associated with PDAC is likely to be a future trend. In addition, recent studies have demonstrated that these four tumor drivers play roles in gemcitabine susceptibility to pancreatic adenocarcinoma cells [10-13]. A recent study showed that a therapeutic antibody targeting KRAS synergistically increased the antitumor activity of gemcitabine by inhibiting RAS downstream signaling for pancreatic cancer with KRAS mutation [10]. The loss or mutation of TP53 promoted gemcitabine resistance in PDAC [11, 12]. P16/CDKN2A was inactivated pancreatic cancer cells were 3–4 fold less sensitive to gemcitabine [13]. DPC4/SMAD4 inactivation was modestly less sensitive to gemcitabine [13]. A previous study using mouse models suggested that co-deletion of PTEN and SMAD4 contributes to and mediates resistance to pancreatic cancer [14]. Additionally, SMAD4 loss in pancreatic cancer causes alterations to multiple kinase pathways (particularly the phosphorylated ERK/p38/Akt pathways), and increases chemoresistance in vitro [15]. Furthermore, PDAC cells with intact SMAD4 are more sensitive to TGF- β 1 inhibitor treatment to reduced cell migration; whereas PDAC cells lacking SMAD4 showed decreased cell motility in response to EGFR inhibitor treatment in PDAC carcinoma tissue [16, 17]. Those evidences support that it is necessary and convincing to develop new treatment strategies for PDAC carrying those fatal mutations in driver genes.”

In the section of Result (Page 16, Line 379-382), we also mentioned that “Previous studies have demonstrated that KRAS, TP53, CDKN2A and SMAD4 play key roles in the tumorigenesis and progression of PDAC, as well as gemcitabine susceptibility [10-13]. Therefore, we further detected the expression of these 4 drivers in these eight cell lines.”

In the section of Discussion (Page 37, Line 1015-Page 38, 1041), we further addressed a number of preclinical studies on currently developing novel therapeutic approaches that target those driver mutations in PDAC, especially for the relationship of p53 and resistance to conventional chemotherapy in pancreatic cancer patients, as shown below. “In the current precision medicine treatment era, diverse genetic alterations in cancer subclones with unique hydrogenous signatures are responsible for the therapeutic resistance of pancreatic cancers as one of the biggest challenges. The most common mutations in PDAC occur in KRAS, TP53, CDKN2A, and SMAD4 driver genes [60-63]. In contrast to other oncogenes (such as mutated BRAF or EGFR), effective therapies that directly target those four driver genes are still unavailable in PDAC. Consequently, there is an urgent need to develop novel biomarkers or therapeutic approaches for the selection of adaptive PDAC patients. Given the critical effect of p53-related pathways on pancreatic cancers, reactivation of p53 has been investigated to sensitize tumors to chemotherapy in a number of preclinical studies [64-67]. A study on cell lines suggested that the mutant p53 activator, PRIMA-1, accelerated apoptosis and sensitized the tumor cells to chemotherapy [68]. The combination of p53 transduction by Ad5CMV-p53 with two genotoxic drugs (gemcitabine and cisplatin), under a correct schedule of administration, appears to be a very promising therapy for human pancreatic cancer [69]. Researchers recently revealed a p53-dependent interaction between cancer cells and CAFs during the response to gemcitabine/abraxane, also suggesting that combining biomarkers of GOF (gain of function) mutant p53 with high HSPG2 stromal deposition may be used in the future to identify patients [70]. Undoubtedly, activation of tumor suppressors for the treatment of

human cancer has been a long sought, yet elusive, strategy. A recent study also provided an example of a “tumor suppressor reactivation” approach. They found that PTEN tumor suppressor was reactivated for cancer treatment through inhibition of the MYC-WWP1 pathway, and further identified a derivative of cruciferous vegetables (I3C) as a potent WWP1 inhibitor via PTEN reactivation, leading to suppression of tumorigenesis [71]. Our findings with a similar therapeutic approach for PDAC might provide important insight into clinical implementation via SMAD4 reactivation induced by DTLL combination treatment.”

2. The phenotyping pancreatic cancer cells used in this study must include the analysis of expression of transporters and enzymes participating in the metabolism of gemcitabine.

Response to reviewer’s comment: Thank you for your constructive suggestion. According to your comment, we performed a proteomic study to determine if SMAD4 has effects on the expression of gemcitabine-relevant transporter or pharmacokinetic enzymes, as shown in Supplementary Table 1. We added and modified the relevant text in the sections of Methods, results and Discussion.

As described in the section of Methods (Page 13-14, Line 332 -347), “Proteomic Analysis: The proteomic analysis of the cell samples was conducted using the combination of DIA and a data dependent acquisition (DDA)-based ion library. After treatment with 2 μ M gemcitabine, 0.1 nM DTLL and both at 37°C for 4 h, cells were harvested and lysed with lysis buffer (8 M urea, 100 mM Tris-HCl pH 7.6, 1 mM PMSF), and then centrifuged at 18000 g for 15 min at 4°C. The extracted protein in the supernatant was quantified using a BCA protein assay kit (Beyotime Biotechnology, China) and digested in trypsin (Promega, Madison, WI) after reduction and alkylation using the FASP (filter aided sample preparation) method. The concentration of digested peptides was determined by measuring the absorbance at 280 nm using a NanoDrop 2000 instrument (Thermo Fisher Scientific, Waltham, MA, USA). An EASY-nLC 1200 chromatography system and an Q Exactive HF mass spectrometer (Thermo Fisher Scientific, Waltham, MA, USA) were used for mass spectrometry acquisition and analysis. The DIA analysis was performed according to a previous study [33]. All results were filtered by a Q-value cutoff of 0.01 (corresponding to an FDR of 1%). The P-value estimator was performed by the Kernel Density Estimator. Area was used for protein quantification. Every peptide was validated with at least three fragments.”

In the section of Results (Page 33, Line 870-889), “As shown in the heatmap (left panel of Fig. 8J), proteomic profiles of BxPC3-Mut and -WT cells without any treatments were quite distinct and showed fewer changes after gemcitabine treated both lines, respectively. DTLL or its combination obviously altered the profile of BxPC3-Mut cells compared to the control or gemcitabine-induced profile, unlike BxPC3-WT cells. Next, we found a total of 12 proteins of gemcitabine-relevant transporters or pharmacokinetic enzymes detectable (Supplementary Table 1B), including ABCC1, CDA, CDC5L, CMPK1, DCK, DCTD, NT5C, NT5C2, NT5C3A, RRM1, RRM2 and SLC29A1. Except for DCK, DCTD, NT5C2 and NT5C3A, all the other nine proteins were significantly altered, as shown in the expression heatmap (right panel of Fig. 8J) between BxPC3-Mut versus

BxPC3-WT cells after treatments. The expression of SMAD4 was included as a positive control, which confirmed that SMAD4 expression alterations ($\text{Ratio}_{\text{Mut/WT}} = 0.017$, $p = 7.3 \times 10^{-5}$), as same as the results in Western blot assay. Specifically, the expression of RRM2 was much lower in BxPC3-Mut than BxPC3-WT cells ($\text{Ratio}_{\text{Mut/WT}} = 0.22$, $p = 2.24 \times 10^{-6}$), and significantly induced after treated with DTLL and its combination, with a similar altered trend of SMAD4 in those cells. The proteomic results implied that SMAD4 status significantly impacted on the proteomic profiles as a result of chemoresistance properties that DTLL might also affect the proteomic profiles of both lines, and that it altered cellular susceptibility via RRM2 induction regulated by SMAD4 in BxPC3-Mut cells.”

As described in Discussions section, “In addition, SMAD4 status significantly impacted the difference in proteomic profiles between BxPC3-Mut and BxPC3-WT cells (Fig. 8J). In particular, RRM2 protein expression was obviously reduced along with decreased mutant SMAD4 in BxPC3-Mut cells, which is also responsible for gemcitabine resistance partially. DTLL alone or its combination treatment significantly triggered an increase in RRM2 expression, interpreting its action mechanism on enhanced BxPC3-Mut cellular susceptibility mediated by increased SMAD4. (Page 37, Line 987-994).” “We demonstrated that SMAD4 status significantly impacted the proteomic profiles of BxPC3-Mut/gemcitabine-resistant and BxPC3-WT/gemcitabine-sensitive cells. The crosstalk between cancer cells and the tumor microenvironment plays an important role in resistance to therapy. More studies need to be conducted to determine whether SMAD4-mediated chemo-sensitization is involved in tumor microenvironment or immunodeficiency (Page 40, Line 1077-1083).”

3. Is the effect of DTLL specific of gemcitabine? Please test different drugs currently utilized in pancreas cancer treatment.

Response to reviewer’s comment: Thank you for pointing out this important question. We studied DTLL interactions with other drugs according to your comment and modified the text in the sections of Results as the below (Page 18, Line 448-463). “To further determine whether DTLL was capable of sensitizing cells to other drugs, we selected multiple antineoplastic agents widely used for pancreatic cancer treatment to detect the effect of DTLL combined with 5-fluorouracil (antimetabolite), oxaliplatin (DNA cross-linking), paclitaxel (anti-microtubule), irinotecan (topoisomerase I inhibitor), etoposide (topoisomerase II inhibitor) and lapatinib (dual HER2 and EGFR tyrosine kinase inhibitor), followed by evaluation of synergistic effect by combination index (CI) calculation. The results (shown in Fig. 2C) indicated that the combination of DTLL with all the above agents showed more significant antiproliferative effectiveness than either DTLL or those agents alone in both AsPC-1 and MIA PaCa-2 cells. The drug interactions of those agents and DTLL were shown to be synergistic with CIs < 1 at those corresponding drug concentrations, except for each one treated point of paclitaxel, 5-fluorouracil, irinotecan or oxaliplatin, respectively. The plots in both AsPC-1 and MIA PaCa-2 cells indicated that drugs at very higher doses showed less synergistic effect with CI values more than 1 (right panels in Fig 2C). These results suggest that DTLL is able to effectively sensitize PDAC cells to different types of drugs in addition to gemcitabine.”

4. Figure 2A. Effect of DTLL on gemcitabine is very similar in both MiaPaCa2 sensitive and AsPC1 resistant cells. It is not clear why authors said that effect of combination of DTLL with gemcitabine results in synergism and not a simple additive effect.

Response to reviewer's comment: Thank you so much for the comment. We evaluated the interaction of DTLL with gemcitabine by using Chou-Talalay method to calculate combination index (CI), which was derived from the median-effect principle with CompuSyn software.

We also addressed the text in the Methods section (Page 8-9, Line 193-206) to calculate combination index (CI) for evaluation of drug interaction at each drug concentration by using the Chou-Talalay method. "Calculation of combination index (CI): After analysis for cell proliferation, the combination index (CI) at each drug concentration was calculated by using the Chou-Talalay method. The equation for CI calculation of Chou-Talalay is as follows:

$$CI = (D1/Dm1) + (D2/Dm2) + \alpha(D1 * D2 / Dm1 * Dm2)$$

The combination index (CI) concept was introduced initially by Chou T.C. and Talalay P [31, 32]. The derived combination index equation for two drugs is:

$$CI = (D)1 / (Dx)1 + (D)2 / (Dx)2,$$

where (Dx) 1, (Dx) 2 = the concentration of the tested substance 1 and the tested substance 2 used in the single treatment that was required to decrease the cell number by x% and (D) 1, (D) 2 = the concentration of the tested substance 1 in combination with the concentration of the tested substance 2 that together decreased the cell number by x%. The CI value quantitatively defines synergism (CI<1), additive effect (CI=1) and antagonism (CI>1)."

After we performed CyQUANT and MTS assays, we tested CI values to determine whether the anti-proliferative effects of DTLL combination with gemcitabine in the both MIA PaCa-2 sensitive and AsPC-1 resistant cell lines were synergistic or simply additive. In addition, we further added the relevant text in the sections of Results (Page 18, Line 439-447). "The results from both MTS (Fig. 2A) and CyQUANT (Fig. 2B) assays indicated that the combination of gemcitabine with DTLL obviously inhibited AsPC-1 cell growth compared to gemcitabine or DTLL alone. Compared with AsPC-1 cells, gemcitabine alone did show a stronger inhibitory effect in MIA PaCa-2 cells. The combination treatment revealed even more significant inhibition of cell growth in a concentration-dependent manner. The results in the both lines showed that the combination treatment had synergistic effects evaluated by the combination index (CI) less than 1 (the threshold line) for both (right panels in Fig. 2A and 2B), indicating that DTLL effectively sensitized the cell response to gemcitabine."

We further demonstrated the synergistic effect of combination of DTLL and gemcitabine on BxPC-3 stably transfected cells expressing wild-type and mutant SMAD4 proteins. "Gemcitabine or DTLL alone, and their combination were tested for proliferative inhibition of BxPC3-EV, BxPC3-WT and BxPC3-Mut cells by using MTS and CyQUANT assays (Fig. 7E and Fig. 7F), respectively. The results indicated that the

combination of gemcitabine with DTLL showed the most significant inhibition rate of tumor cell growth when compared to gemcitabine or DTLL alone, obviously showing a synergistic effect evaluated by CI calculation, not a simple additive effect (Page 27, Line 701-707).”

5. Figure 2B. The cytometry study was performed after 72 h of treatment, when most of the MiaPaCa2 sensitive and AsPC1 resistant cells died or are dying (Figure 1A). The analysis was apparently performed only in living cells (no SubG1 cells were presented in the pictures). Please present these cells.

Response to reviewer’s comment: Thank you so much for the reviewer’s comment. We understood that most of the MIA PaCa-2 sensitive and AsPC-1 resistant cells died or are dying after 72 h of treatment, which hardly reflects biological behaviors for all the sub-populations of G1, S, G2 and M cells if we detect them at 72 h of post-treatment. Therefore, we performed flow cytometry for the cell cycle assay after treated the both cells with drugs for 24 h actually, not for 72 h. Under this condition that was consistent with immunoblotting test time-point at 24 h of post-treatment, cell cycle assay was performed to test all living cells at different stages of cell phases, as presented in the Figure 2D (Previous Figure 2B). In fact, to investigate effects on drug cytotoxicity, we performed the cell viability assay for AsPC-1 and MIA PaCa-2 cells with drug exposure for 72 h, respectively, in Figure 2A.

6. Figure 3A. Authors suggest that the effect DTLL on gH2AX, BCL2 and cyclin D1 could explain DTLL mechanism but it is similar in both MiaPaCa2 sensitive and AsPC1 resistant cells.

Response to reviewer’s comment: Thank you so much for the comment. DTLL was designed to be an EGFR/HER2 bispecific antibody drug conjugate (ADC)-like immunotherapeutic agent, named as dual-targeting ligand-based lidamycin (DTLL). Lidamycin is well-known to function as an antitumor cytotoxic agent directly through DNA damaging mechanism. As the lidamycin derivative, DTLL also showed similar cytotoxic effect through promoting DNA damage in AsPC-1 and MIA PaCa-2 cells, and inhibited induced growth inhibition and G2/M arrest experimentally. Therefore, our results from Western blot assay further verified that DTLL altered downstream signaling marker proteins, such as DNA damage-related protein gH2AX, apoptosis-related protein Bcl-2 and cell cycle regulatory protein cyclin D1. The similar effect of DTLL on cell signaling only indicates that those downstream proteins could be reflective outcomes in AsPC-1 and MIA PaCa-2 cells. However, we found that the action mechanisms of DTLL in combination with gemcitabine showed quite different responsive changes in upstream cell signaling of these proteins (gH2AX, Bcl-2 and cyclin D1) mediated by SMAD4 in gemcitabine-sensitive and gemcitabine-resistant cells. Actually, the expression and bioactivity of SMAD4, together with its target proteins (NF-kB and downstream apoptotic signaling molecules), were in totally opposite directions between these two lines.

As described in the section of Results (Page 30, Line 787-804), “Interestingly, we

found that the expression levels of phospho-P65, P65 and P50 units of NF- κ B, a well-known transcription factor, were induced significantly by DTLL or its combination in the BxPC3-Mut line. Along with increased mutant SMAD4 by DTLL induction, apoptotic proteins including BAX, FADD and cleaved caspase-8 were significantly induced, while anti-apoptotic Bcl-2 and MCL1 proteins were reduced (Fig. 8F). Moreover, the semiquantification results in BxPC3-Mut cells demonstrated that there were significant decreases in the ratio of Bcl-2/BAX and MCL1/BAX after treatment with DTLL or its combination compared to the control, indicating the consequence of enhanced cell apoptosis by DTLL. On the other hand, in the BxPC3-WT line, all of the above proteins were apparently reduced with decreased SMAD4, except for induced cleaved caspase-8 (Fig. 8F). The results from semiquantification in the BxPC3-WT line also demonstrated the consequence of cell apoptosis with significant decreases in the ratio of Bcl-2/BAX and MCL1/BAX after treated by DTLL or its combination compared to the control. The directions of the changes in SMAD4 and NF- κ B expression induced by DTLL, as well as the expression of their targeted apoptotic/anti-apoptotic proteins, were opposite between the BxPC3-WT and BxPC3-Mut lines, but eventually led to the consequence of enhanced cell apoptosis and sensitized drug response.”

As described in the section of Discussion, “Interestingly, in the present study, we found that SMAD4 expression and bioactivity of AsPC-1/BxPC3-Mut and MIA PaCa-2/BxPC3-WT cells were distinct although DTLL or its combination treatment had inhibitory effects on both lines. In AsPC-1 cells deficient in SMAD4, we observed reactivation of SMAD4 with nuclear translocation to form a SMAD2/3/4 complex in the nucleus and a decrease in SMAD7 expression (Fig. 3B-D), similar to the findings in Fig. 8D (Page 35, Line 948-953).” “In contrast, the combination treatment obviously inhibited SMAD4 activity in MIA PaCa-2 cells (Fig.3B-D), consistent with the results verified in BxPC3-WT cells (Fig. 8D). Our findings demonstrated that the combination treatment enhanced antineoplastic activities via blockage of AKT/mTOR signaling pathways in gemcitabine-sensitive PDAC models (Page 36, Line 959-963).” “As shown in the schematic illustration model in Fig. 9, we investigated different action mechanisms of mutant and wild-type SMAD4 on the PDAC drug response with/without DTLL induction (shown in Fig. 8). Our findings demonstrated that the SMAD4 genetic status of PDAC is responsible for SMAD4 protein levels, which determine different cellular susceptibilities. Moreover, DTLL significantly altered the half-life time and protein levels of mutant and wild-type SMAD4 by inhibiting protein degradation at different velocities and changing the interaction of SMAD4 with TRIM33 (Fig. 8A-C). In addition, DTLL predominantly enhanced the expression of apoptotic proteins (BAX, FADD) in BxPC3-Mut cells but promoted decreases mainly in antiapoptotic proteins (Bcl-2 and MCL1) of the BxPC3-WT line (Fig. 8D-G) through the regulation of upstream NF- κ B transcriptional activity. Moreover, NF- κ B expression was controlled as a target of either mutant or wild-type SMAD4, and the interaction of those two proteins in BxPC3-Mut and BxPC3-WT cell lines exhibited differently in an opposite direction (Fig. 8H and 8I). The trends in protein expression affected by DTLL (including NF- κ B, TRIM33 and apoptosis-relevant proteins) were mainly attributed to SMAD4 protein levels on the basis of its genetic status. In addition, SMAD4 status significantly impacted the difference in proteomic profiles between BxPC3-Mut and BxPC3-WT cells

(Fig. 8J). In particular, RRM2 protein expression was obviously reduced along with decreased mutant SMAD4 in BxPC3-Mut cells, which is also responsible for gemcitabine resistance partially. DTLL alone or its combination treatment significantly triggered an increase in RRM2 expression, interpreting its action mechanism on enhanced BxPC3-Mut cellular susceptibility mediated by increased SMAD4 (Page 36, Line 972-994).”

As described in the section of Conclusion (Page 40, Line 1088-1105), “Taken together, we first demonstrated the synergism of gemcitabine and DTLL in both gemcitabine-sensitive and gemcitabine-resistant PDAC and identified the critical role of SMAD4 in mediating the cellular responses of pancreatic cancer to chemotherapy. SMAD4 genetic status is responsible for SMAD4 protein levels, and its R100T mutation contributes to loss of SMAD4 protein and function with rapid protein degradation, leading to resistance to gemcitabine of PDAC cells. DTLL altered the protein half-life time and level of mutant and wild-type SMAD4 by inhibiting protein degradation at different velocities and changing the interaction of SMAD4 with TRIM33 distinctly. Further studies implied that DTLL combinational treatment might not only prevent from neoplastic proliferation via blockage of ATK/mTOR signaling and anti-apoptotic proteins mediated by impaired NF- κ B function if given in SMAD4-sufficient/gemcitabine-sensitive PDAC cells, but also restore the bioactivity of SMAD4 as a tumor suppressor to trigger its downstream NF- κ B-regulated signaling of cell apoptosis and enhance the gemcitabine-relative enzyme RRM1 in SMAD4-deficient/gemcitabine-resistant tumors. Our findings may unravel a potential combination therapy for overcoming resistance of PDAC mediated by SMAD4, which paves the way toward a long-sought tumor suppressor reactivation approach and lays the foundation for future precise medicine as the first step.”

7. In Figure 3B authors found a switch in the expression of SMAD4 after treatment with DTLL and therefore they hypothesize that this is the mechanism of sensitization to gemcitabine by DTLL. This is only an association between expression of SMAD4 and gemcitabine effect. This hypothesis needs to be validated by overexpressing SMAD4 in AsPC1 resistant cells and knockdown Smad4 in MiaPaCa2 sensitive cells.

Response to reviewer’s comment: Thank you so much for the comment. According to the reviewer’s advice, we conducted research by overexpressing SMAD4 in AsPC-1 resistant cells and knocking down SMAD4 in MIA PaCa-2 sensitive cells to validate our hypothesis, as described in the section of Results (Page 22, Line 546-560).

“Down regulation or upregulation of SMAD4 altered PDAC susceptibility to gemcitabine:

Since we have found a relationship between the gemcitabine response and SMAD4 protein levels, we altered SMAD4 expression by transiently transfecting either a specific siRNA or overexpression vector of SMAD4 into pancreatic cancer cells to detect the effect of SMAD4 expression levels on PDAC cellular susceptibility to gemcitabine. As shown in Fig. 4A, downregulation of SMAD4 at both mRNA and protein levels in MIA PaCa-2 cells that carry wild-type SMAD4 gene with high protein levels significantly reduced cell sensitivity to gemcitabine after exposure to various concentrations for 72 h. In contrast, SMAD4 overexpression apparently sensitized AsPC-1 cells carrying R100T mutation at little

protein expression of SMAD4 gene to gemcitabine (Fig. 4B). The results indicated that SMAD4 contributed to cellular response to gemcitabine in PDAC cells. However, it might be involved in distinct action mechanisms owing to different genotypes and protein levels of SMAD4 in MIA PaCa-2 versus AsPC-1 cells.”

8. Is SMAD4 expression correlated to gemcitabine sensitivity in data available from public databases? Please analyze this possibility.

Response to reviewer’s comment: Thank you so much for the comment. We performed a bioinformatics analysis and addressed more detailed information in the section of Methods (Page 14, Line 348-362). “Bioinformatics analysis: We downloaded a file (TCGA-PAAD.htseq_fpkms.tsv) from the TCGA-PAAD Project of TCGA public datasets on the website (<https://portal.gdc.cancer.gov/repository>) including clinical information on gemcitabine treatment and SMAD4 expression data of pancreatic cancer patients to determine whether SMAD4 expression is clinically correlated with gemcitabine response. Among the 108 patients, 69 samples had a gemcitabine response available. Responsive data were defined as four different types of gemcitabine response data: complete response, partial response, stable disease and clinical progressive disease. First, we analyzed the relationship between SMAD4 expression and gemcitabine response. Second, we plotted the ROC (Receiver Operating Characteristic) curve evaluated with an AUC (area under curve) and took 2.27 (median expression of all the PAAD patients) as the threshold for SMAD4 expression to define the high (>2.27) or low (<2.27) level, followed with Fisher-exact test for calculation of odd ratio and p-value.”

As described in the section of Results (Page 17-18, Line 419-434), “Furthermore, we downloaded the public data (TCGA-PAAD project) of pancreatic cancer patients from TCGA including clinical information on gemcitabine treatment and SMAD4 expression datasets to determine if SMAD4 expression in a PDAC patient is clinically correlated with gemcitabine response. By using the expression level of SMD4 as the criterion for predicting whether a patient is cured, we plotted an ROC curve with an AUC equal to 0.597 (Fig. 1E). There were 35 patients with high SMAD4 expression, of which 17 (48.57%) samples had a positive drug response, whereas only 34.38% of patients (11 from 32 samples) with a low level of SMAD4 responded positively to gemcitabine. The odds ratio was 1.98 ($p = 0.22$), which indicated that the high level of SMAD4 expression facilitated the gemcitabine response clinically, although the significance level was more than 0.05 perhaps owing to the small sample size.

Therefore, we demonstrated that SMAD4 was associated with not only PDAC occurrence and progression, but also the drug response to gemcitabine treatment experimentally and clinically.”

9. The experiment performed on PDTX from 2 different patients, SMAD4 mutant and wildtype, is interesting but not enough convincing to proof the hypothesis. Like in point 8, rescue of SMAD4 expression in PA1233 and or knocking down its expression in PA3142 are necessary to validate the hypothesis, otherwise, it remains only a hypothesis.

Response to reviewer's comment: Thank you so much for the comment. According to the reviewer's suggestion, we evaluated *in vivo* gemcitabine efficacy by using CDX models derived from BxPC-3 cells with stable expression of wild-type and mutant SMAD4, instead of PDX models owing to technical and financial issues. We have also added text regarding relevant experiments in the section of Methods, including "Transfection with plasmid or specific siRNA, Site-mutagenesis assay, Stable transfection for SMAD4 overexpression, Real time quantitative reverse transcription-PCR (real time qRT-PCR) and *In vivo* therapeutic efficacy (Page 9-11, Line 221-268)."

As we described in the section of Results (Page 26, Line 688-693), "BxPC3 cells do not express SMAD4 mRNA or protein owing to gene deletion. Therefore, we used this cell line to generate three types of stably transfected cells containing empty control, mutant and wild-type SMAD4 overexpression vectors after creating the mutant SMAD4 overexpression vector via site-mutagenesis assay (Fig. 7A) to further test the different effects of wild-type and mutant SMAD4 on drug efficacy."

"Next, we used the above BxPC3-EV, BxPC3-WT and BxPC3-Mut stably transfected cell lines to generate *in vivo* CDX mouse models for evaluation of antineoplastic efficacy of vehicle, gemcitabine or DTLL alone, and the combination. As shown in Fig. 7H and Table 3, gemcitabine had a minimal inhibitory effect on BxPC3-Mut tumors at 7.46% and DTLL slightly repressed tumor growth by 47.61%, however, the combination of gemcitabine and DTLL had remarkable synergistic efficacy with a tumor inhibition rate of 78.62% in this gemcitabine-resistant xenograft model. In the BxPC3-WT models, gemcitabine or DTLL alone was able to apparently inhibit tumor growth by 65.33% or 59.19%, respectively, suggesting its sensitivity to both drugs. The combination treatment significantly repressed tumor growth by 82.06%. In addition, there were no deaths or significant changes in body weight observed in treated mice (Fig. S2C), as well as no toxic-pathological organs (Fig. S3C), compared to the control group. This further confirmed the synergistic inhibitory effect of DTLL in combination with gemcitabine on *in vivo* tumor growth, similar to the above results obtained in both CDX and PDX models (Page 27-28, Line 714-728)."

10. Please explain why authors studied expression of PDL1 in response to the treatment with DTLL. This data was previously reported.

Response to reviewer's comment: Thank you so much for the reviewer comment. As mentioned in the section of Introduction (Page 5, Line 115-124), "DTLL might suppress pancreatic tumor progression by EGFR/HER2-dependent blockage of AKT/mTOR-signaling and PD-L1/PD1-mediated escape from immunosurveillance in PDAC [25]." "We previously found that human pancreatic cancer AsPC-1 and MIA PaCa-2 cells with high EGFR/HER2 expression showed the strongest binding affinity to DTLL, the precursor of DTLL, however, AsPC-1 cell line or xenograft tumor showed intermediate resistance to DTLL whereas MIA PaCa-2 had stronger affinity and better response [25]. This raised an interesting proposal and encouraged us to explore more effective strategies and underlie relevant mechanisms especially for drug-resistant pancreatic cancers." Consequently, in the present study, we aimed to explore how and what the difference in

signaling pathways causes distinguished drug responses in gemcitabine-sensitive and gemcitabine-resistant PDAC cells.

Therefore, we described the data with regarding to PD-L1 expression in response to the treatment with DTLL in the section of Results, as follow. “Our previous study demonstrated that DTLL enhanced DNA damage via EGFR/HER2-dependent blockage of the AKT/mTOR and PD-L1 signaling pathways in gemcitabine-sensitive MIA PaCa-2 cells. Hence, to test if these cellular signaling pathways are involved, we treated gemcitabine-resistant AsPC-1 or gemcitabine-sensitive MIA PaCa-2 cells with vehicle, gemcitabine, DTLL or both for 4 h, and further analyzed the ratios of active phosphorylated and total proteins for HER-2, EGFR, ERK1/2, AKT, mTOR and PD-L1. Interestingly, the responses in the AKT/mTOR and PD-L1 signaling pathways were quite different between AsPC-1 and MIA PaCa-2 cells. After treatment with DTLL or the both in MIA PaCa-2 cells, the ratios of active phosphorylated and total proteins for EGFR, HER-2, AKT and mTOR were obviously decreased (Fig. 3A). Furthermore, there were significant decreases in PD-L1 expression, which confirmed our previous findings regarding to DTLL function [25] and suggested similar AKT/mTOR signaling affected by its combination treatment in MIA PaCa-2 cells. In contrast, the ratios of p-EGFR/EGFR, p-HER-2/HER-2, p-AKT/AKT and p-mTOR/mTOR were significantly increased in AsPC-1 cells. Furthermore, an apparent increase in PD-L1 expression was observed in AsPC-1 cells, indicating that there were obviously different molecular mechanisms by which these two cells differed in drug response to the combination therapy (Page 20, Line 490-507).” Moreover, we also observed the same results by using in vivo CDX and PDX models. “For SMAD4-sufficient MIA PaCa-2 tumor models, the ratios of p-EGFR/EGFR, p-HER-2/HER-2, p-AKT/AKT and p-mTOR/mTOR as well as PD-L1 expression were more effectively inhibited after treatment with DTLL or both than in gemcitabine or control treated group, whereas slightly decreased SMAD4 and dramatically increased SMAD7 were observed. These data implied that combination treatment enhanced antineoplastic efficacy via EGFR/HER2-dependent blockage of AKT/mTOR and PD-L1 signaling pathways in MIA PaCa-2 xenografts, consistent with the above in vitro results as well as our previous findings of DTLL [25] Moreover, we detected significant increases in the ratios of active phosphorylated and total proteins for EGFR, HER-2, AKT, and PD-L1, as well as little alteration in p-mTOR/mTOR with gemcitabine or the combination treatment. In contrast to the effect on SMAD4-sufficient MIA PaCa-2 tumors, DTLL inhibited the growth of SMAD4-deficient AsPC-1 tumors but not by blocking EGFR/AKT/mTOR and PD-L1 signaling pathways (Page 24, Line 615-633).” “Subsequently, we investigated the drug effects on the expression of relevant proteins in the AKT/mTOR and TGF- β /SMADs signaling pathways in the two PDX models.....the combination significantly upregulated the protein levels of SMAD4 and PD-L1 in PA1233 tumors and downregulated these two proteins in PA3142 models (Fig. 6C) (Page 26, Line 670-676).”

“DTLL or the combination treatment increased the expression of PD-L1 in mutant SMAD4 tumors but reduced its expression in wild-type tumors, which might promote or inhibit immune escape in vitro and in vivo, respectively. However, whether the exact mechanisms of tumor immune escape and SMAD4-mediated activation or inhibition

occur is unclear and needs further deeper investigation.” To avoid unnecessary confusion, we restated the part of PD-L1 expression in the section of Discussion (Page 39, Line 1055-1060).

11. Please quantify Ki67 and TUNEL staining.

Response to reviewer’s comment: Thank you so much for the comment. According to the reviewer’s suggestion, we performed semi-quantification analysis for Ki67 and TUNEL staining in tumor tissue samples from AsPC-1 and MIA PaCa-2 CDX models (Fig. 5D and 5E), as well as PA1233 and PA3142 PDX models (Fig. 6D and 6E).

Reviewer #2 (Remarks to the Author); antibody-drug conjugates therapy:

This paper describes synergistic effects between a dual-targeted toxin-linked monoclonal antibody in gemcitabine -sensitive and -resistant human cell line and PDX models of human pancreatic adenocarcinoma. Extensive molecular characterisation of the cancer cells utilised is presented, and this is very interesting data. The fact that the combination is more effective than either treatment alone, and in both gemcitabine-sensitive and resistant lines, seems promising. I think it deserves to be published, but I would like the authors would state clearly, be prepared to share their monoclonal with other labs- under an appropriate Materials Transfer Agreement of course - which have the capacity for toxin labelling (My own lab does not have this capacity, so this is a matter of principle, not out of self-interest) - I suggest that this should be a condition of publication, because unless others can replicate and confirm the interesting findings presented here, the paper is of limited utility.

Response to reviewer’s comment: Thank you so much for the comment. In our study, DTLL, as a novel bispecific antibody-drug conjugate (ADC)-like agent, was designed and generated in our laboratory and has previously been applied for a pair of patents in China. Certainly, under an appropriate Materials Transfer Agreement, we would like to share our DTLL samples with other laboratories to replicate and verify our findings if they are interested in the promising effect of DTLL on antitumor activity or even overcoming chemoresistance.

The paper is well written, the work seems well planned and conducted.

1. I would, however think that it should be a requirement for publication to remove "on the basis of SMAD-4 profiles" from both title and abstract -perhaps replace by something like "in both gemcitabine-sensitive and -resistant phenotypes

Response to reviewer’s comment: Thank you so much for the comment. According to the reviewer’s suggestion, we decided to choose the title “An antibody drug conjugate-like agent DTLL sensitizes gemcitabine-sensitive/resistant pancreatic cancer on basis of SMAD4 profiles” for our manuscript, as well as the running title “DTLL sensitizes gemcitabine-sensitive/resistant PDAC based on SMAD4 profiles”. For the reason of maintaining SMAD4, we answer the reviewer’s questions 1 & 2 in detail together.

2. Why? The data on SMAD4 is interesting and should be retained in the paper, but a causal link as suggested in the title is not warranted as yet, because it is suggested by some, but not all of the cell line results

Response to reviewer's comment: Thank you so much for the comment. We performed more experiments and re-addressed the relevant text to clarify the causal link between SMAD4 and DTLL effects on the PDAC response more clearly, according to the review's suggestion.

As we mentioned in the section of Introduction, "Our previous studies demonstrated that DTLL was superior to free LDM alone in ovarian carcinoma [24], PDAC [25], and esophageal cancer [26].....We previously found that human pancreatic cancer AsPC-1 and MIA PaCa-2 cells with high EGFR/HER2 expression showed the strongest binding affinity to DTLP, the precursor of DTLL, however, AsPC-1 cell line or xenograft tumor showed intermediate resistance to DTLL whereas MIA PaCa-2 had stronger affinity and better response [25]. This raised an interesting proposal and encouraged us to explore more effective strategies and underlie relevant mechanisms especially for drug-resistant pancreatic cancers (Page 5, Line 113-124)."

Our hypothesis in the present study was based on distinguished effects on PDAC tumor inhibition of DTLL in combination with gemcitabine, not only regarding the DTLL effect as a single antineoplastic agent but also particularly focusing on how to re-sensitize gemcitabine efficacy in SMAD4-deficient/resistant PDAC. We hypothesized that it might be possible to improve the chemotherapeutic efficacy of gemcitabine in PDAC when combined with DTLL and indeed found that SMAD4 deficiency was associated with gemcitabine resistance and that DTLL in combination with gemcitabine sensitized those resistant PDAC cells to gemcitabine and further elucidated its action mechanisms. Furthermore, we performed mechanism studies to functionally characterize the effect of SMAD4 expression on gemcitabine response, and differences in DTLL action mechanism between mutant/resistant and wild-type SMAD4/sensitive PDAC cells.

Experimentally, we have demonstrated that upregulated or downregulated SMAD4 expression by its specific siRNAs or overexpression vectors in SMAD4-deficient AsPC-1 and SMAD4-sufficient MIA-paca-2 cells altered PADC cellular susceptibility to gemcitabine, respectively. DTLL in combination with gemcitabine potentiated the inhibitory effects of gemcitabine on *in vitro* cell viability and synergistically repressed the *in vivo* tumor growth of AsPC-1 and MIA PaCa-2 CDX models. Furthermore, we generated three types of stably transfected BxPC-3 cells containing empty, mutant and wild-type SMAD4 overexpression vectors and their corresponding CDX models to investigate the effect of DTLL, and to explore its action mechanism on sensitization of gemcitabine response along with SMAD4 genetic or expression status, respectively. The results have shed light on its therapeutic efficacy *in vitro* and *in vivo*.

"As shown in the schematic illustration model in Fig. 9, we investigated different action mechanisms of mutant and wild-type SMAD4 on the PDAC drug response with/without DTLL induction (shown in Fig. 8). Our findings demonstrated that SMAD4 genetic status of PDAC is responsible for SMAD4 protein levels, which determine

different cellular susceptibilities. Moreover, DTLL significantly altered the half-life time and protein levels of mutant and wild-type SMAD4 by inhibiting protein degradation at different velocities and changing the interaction of SMAD4 with TRIM33 (Fig. 8A-C). In addition, DTLL predominantly enhanced the expression of apoptotic proteins (BAX, FADD) in BxPC3-Mut cells but promoted decreases mainly in antiapoptotic proteins (Bcl-2 and MCL1) of the BxPC3-WT line (Fig. 8D-G) through the regulation of upstream NF- κ B transcriptional activity. Moreover, NF- κ B expression was controlled as a target of either mutant or wild-type SMAD4, and the interaction of those two proteins in BxPC3-Mut and BxPC3-WT cell lines exhibited differences in the opposite direction (Fig. 8H and 8I). The trends in protein expression affected by DTLL (including NF- κ B, TRIM33 and apoptosis-relevant proteins) were mainly attributed to SMAD4 protein levels on the basis of its genetic status. In addition, SMAD4 status significantly impacted the difference in proteomic profiles between BxPC3-Mut and BxPC3-WT cells (Fig. 8J). In particular, RRM2 protein expression was obviously reduced along with decreased mutant SMAD4 in BxPC3-Mut cells, which is also responsible for gemcitabine resistance partially. DTLL alone or its combination treatment significantly triggered an increase in RRM2 expression, interpreting its action mechanism on enhanced BxPC3-Mut cellular susceptibility mediated by increased SMAD4 (Page 36, Line 972-994).”

Based on the above consideration, we created a causal link between SMAD4 and DTLL effects on the PDAC response more clearly by using two different types (gemcitabine-resistant/sensitive or SMAD4-mutant/wild-type) of PDAC cells and CDX/PDX models in our study, such as AsPC-1 versus MIA PaCa-2, and BxPC3-Mut versus BxPC3-WT.

Apart from those 2 conditions, I am enthusiastic about this paper and support its acceptance.

Response to reviewer’s comment: Thank you so much for the consideration of our manuscript.

Martin Clynes

Reviewer #3 (Remarks to the Author); TGF- β signalling:

In this manuscript Yao et al suggest that an antibody drug conjugate-like based on a EGFR/HER2 dual targeting ligand-based lidamycin (DTLL) sensitizes pancreatic cancer cells to gemcitabine. In addition, they suggest that this process is commonly occurring in SMAD4-depleted and expressing cells. However, in both cases, the authors suggest by a series of protein expression associations that is dependent on SMAD4 activity. Overall, this is an interesting application for this recently published drug conjugate. In addition, the authors present treatment effects in two cell line models and two PDX to supporting their hypothesis. However, the current manuscript is largely descriptive, does not mechanistically support any of the findings and lacks any genetic or functional validation of any of the findings.

Response to reviewer's comment: Thank you so much for the comment. According to your suggestion and in order to verify our hypothesis, we performed mechanism studies to functionally characterize the effect of SMAD4 expression on gemcitabine response, and the difference in DTLL action mechanism of between mutant SMAD4/resistant and wild-type SMAD4/sensitive PDAC cells. We demonstrated that SMAD4 upregulated or downregulated expression by using its specific overexpression vector/or siRNA in PDAC cells altered cellular susceptibility to gemcitabine, and that DTLL re-sensitized its efficacy in both resistant AsPC-1 and sensitive MIA PaCa-2 cells, respectively. "As shown in Fig. 4A, downregulation of SMAD4 at both the mRNA and protein levels in MIA PaCa-2 cells that carry wild-type SMAD4 gene at high protein levels significantly reduced cell sensitivity to gemcitabine after exposure to various concentrations for 72 h. In contrast, SMAD4 overexpression apparently sensitized AsPC-1 cells carrying R100T mutation at little protein expression of *SMAD4* gene to gemcitabine (Fig. 4B). The results indicated that SMAD4 contributed to cellular response to gemcitabine in PDAC cells (Page 22, Line 552-558)."

"BxPC3 cells do not express SMAD4 mRNA or protein of owing to gene deletion. Therefore, we used this cell line to generate three types of stably transfected cells containing empty control, mutant and wild-type SMAD4 overexpression vectors after creating the mutant SMAD4 overexpression vector via site-mutagenesis assay (Fig. 7A) to further test the different effects of wild-type and mutant SMAD4 on drug efficacy.....We further evaluated the *in vitro* inhibitory effects of gemcitabine on the growth of BxPC3 cells with empty vector (BxPC3-EV), wild-type (BxPC3-WT) and mutant (BxPC3-Mut) overexpression vectors. As shown in Fig. 7D, BxPC3-WT cells were more sensitive to gemcitabine than BxPC3 mock or BxPC3-EV cells, but BxPC3-Mut cells showed little difference from those two. Gemcitabine or DTLL alone, and their combination were tested for proliferative inhibition of BxPC3-EV, BxPC3-WT and BxPC3-Mut cells by using MTS and CyQUANT assays (Fig. 7E and Fig. 7F), respectively (Page 26, Line 688-704)." "Next, we used the above BxPC3-EV, BxPC3-WT and BxPC3-Mut stably transfected cell lines to generate *in vivo* CDX mouse models for evaluation of antineoplastic efficacy of vehicle, gemcitabine or DTLL alone, and the combination. As shown in Fig. 7H and Table 3, gemcitabine had a minimal inhibitory effect on BxPC3-Mut tumors at 7.46% and DTLL slightly repressed tumor growth by 47.61%, however, the combination of gemcitabine and DTLL had remarkable synergistic efficacy with a tumor inhibition rate of 78.62% in this gemcitabine-resistant xenograft model. In the BxPC3-WT models, gemcitabine or DTLL alone was able to apparently inhibit tumor growth by 65.33% or 59.19%, respectively, suggesting its sensitivity to both drugs. The combination treatment significantly repressed tumor growth by 82.06% (Page 27-28, Line 714-723)." The *in vitro* and *in vivo* results in BxPC-3 cells or xenograft tumors further confirmed our observation with AsPC-1 and MIA PaCa-2 cells that SMAD4 genetic status is responsible for SMAD4 protein level that is consequently contributed to cellular susceptibility to gemcitabine. Furthermore, we investigated DTLL action mechanism on sensitization of gemcitabine response along with SMAD4 genetic or expression status, respectively, thereby to shed light on its therapeutic efficacy *in vitro* and *in vivo*."

"Consequently, all the above findings suggested that DTLL is capable of sensitizing

either gemcitabine-resistant or gemcitabine-sensitive cells to gemcitabine efficacy (or even other chemotherapeutic agents) through distinct action mechanisms according to different SMAD4 profiles. As shown in the schematic illustration model (Fig. 9), the mechanistic studies implied that DTLL combinational treatment might not only prevent neoplastic proliferation via blockage of ATK/mTOR signaling and anti-apoptotic proteins (Bcl-2 and MCL1) mediated by impaired NF-kB function if given to SMAD4-sufficient/gemcitabine-sensitive PDAC cells, but also restore the bioactivity of SMAD4 as a tumor suppressor to trigger its downstream NF-kB-regulated signaling of cell apoptosis in SMAD4-deficient/gemcitabine-resistant tumors. The inhibitory effectiveness of DTLL alone or in combination with gemcitabine revealed that it functions as a double-edged sword in PDAC tumor growth, with totally different drug responsive behaviors and molecular action mechanisms on synergistic efficacy between gemcitabine-resistant and gemcitabine-sensitive PDAC. We speculate that DTLL seems to act as a novel module of the SMAD4 driver that normalizes its function as a tumor suppressor according to SMAD4 protein level and genetic status in PDAC, and then adjusts NF-kB, TRIM33, gemcitabine-metabolic enzymes (such as RRM2) and proteomic profiles of PDAC cells, which explains the synergistic effect on tumor growth of DTLL in combination with gemcitabine (Page 37, Line 995-1013).”

Therefore, we conclude that SMAD4 protein levels is contributed to gemcitabine response and that DTLL in combination with gemcitabine might improve chemotherapeutic efficacy in PDAC. SMAD4 could be a predominant switch or driver for PDAC susceptibility and serve as a predictive marker for determining treatment outcomes to overcome chemoresistance clinically. Our findings may unravel a potential combination therapy for overcoming resistance of PDAC mediated by SMAD4, which paves the way toward a long-sought tumor suppressor reactivation approach and lays the foundation for future precise medicine as the first step.

Overall, there are several conceptual issues that are not addressed. First, -the authors conclude that SMAD4 defines responsiveness to gemcitabine treatment on the basis of correlations between treatment response and protein expression. Loss of function experiments are needed to:

Response to reviewer’s comment: Thank you so much for the comment. To verify our hypothesis, we performed mechanism studies to functionally characterize the effect of SMAD4 expression on gemcitabine response, and differences in DTLL action mechanism of between mutant SMAD4/resistant and wild-type SMAD4/sensitive PDAC cells. Those functional experiments verified our hypothesis and we re-addressed our findings in detail in the text of Results (Page 22, Line 548 – 587; Page 26, Line 688 – Page 33, Line 894) and Discussion sections (Page 34, Line 898 - Page 40, Line 1085). Our point-by-point responses to the comments and suggestions are provided in the below. All changes in our manuscript have been highlighted by using the track changes mode in MSWord. A clean version of the manuscript has also been uploaded for your reference.

i) confirm that SMAD4 is functionally required and its downstream targets needed; ii) Is

resistance to gemcitabine gained upon SMAD4 depletion? Is it SMAD4 causal mediator of the effect?

Response to reviewer's comment: We confirmed that SMAD4 is functionally required for determination of PDAC cell susceptibility to gemcitabine by using its specific siRNA knockdown and overexpression, followed with observation of drug inhibitory effect on *in vitro* tumor cell growth. Furthermore, we observed the effect of SMAD4 levels on PDAC cell susceptibility to gemcitabine by using MIA PaCa-2 and AsPC-1 corresponding CDX animal models. We also evaluated the *in vivo* efficacy of PDX models of human pancreatic cancer. Our findings demonstrated that SMAD4 genetic status is responsible for SMAD4 protein levels, consequently contribute to cellular susceptibility to gemcitabine.

As described in the section of Results, "As shown in Fig. 4A, downregulation of SMAD4 at both mRNA and protein levels in MIA PaCa-2 cells that carry wild-type *SMAD4* gene with high protein levels significantly reduced cell sensitivity to gemcitabine after exposure to various concentrations for 72 h. In contrast, SMAD4 overexpression apparently sensitized AsPC-1 cells carrying R100T mutation at little protein expression of SMAD4 gene to gemcitabine (Fig. 4B). The results indicated that SMAD4 contributed to cellular response to gemcitabine in PDAC cells (Page 22, Line 552-558)." "Evaluation of *in vivo* efficacy in CDX models from AsPC-1 and MIA PaCa-2 cells.....Evaluation of *in vivo* efficacy by PDX models of human pancreatic cancer (Page 23-26, Line 588-685)"

"BxPC-3 cells do not express SMAD4 mRNA or protein of owing to gene deletion. Therefore, we used this cell line to generate three types of stably transfected cells containing empty control, mutant and wild-type SMAD4 overexpression vectors after creating the mutant SMAD4 overexpression vector via site-mutagenesis assay (Fig. 7A) to further test the different effects of wild-type and mutant SMAD4 on drug efficacy.....As shown in Fig. 7D, BxPC3-WT cells were more sensitive to gemcitabine than BxPC3 mock or BxPC3-EV cells, but BxPC3-Mut cells showed little difference from those two. Gemcitabine or DTLL alone, and their combination were tested for proliferative inhibition of BxPC3-EV, BxPC3-WT and BxPC3-Mut cells by using MTS and CyQUANT assays (Fig. 7E and Fig. 7F), respectively (Page 26, Line 688-704)." "Next, we used the above BxPC3-EV, BxPC3-WT and BxPC3-Mut stably transfected cell lines to generate *in vivo* CDX mouse models for evaluation of antineoplastic efficacy of vehicle, gemcitabine or DTLL alone, and the combination. As shown in Fig. 7H and Table 3, gemcitabine had a minimal inhibitory effect on BxPC3-Mut tumors at 7.46% and DTLL slightly repressed tumor growth by 47.61%, however, the combination of gemcitabine and DTLL had remarkable synergistic efficacy with a tumor inhibition rate of 78.62% in this gemcitabine-resistant xenograft model. In the BxPC3-WT models, gemcitabine or DTLL alone was able to apparently inhibit tumor growth by 65.33% or 59.19%, respectively, suggesting its sensitivity to both drugs. The combination treatment significantly repressed tumor growth by 82.06% (Page 27-28, Line 714-723)." The *in vitro* and *in vivo* results in BxPC-3 cells or xenograft tumors further confirmed our observation with AsPC-1 and MIA PaCa-2 cells that SMAD4 genetic status is responsible for SMAD4 protein level that is consequently contributed to cellular susceptibility to gemcitabine. We also demonstrated the synergistic effects of DTLL and gemcitabine combination on inhibition

of BxPC3-WT and BxPC3-Mut cell viability.

We performed functional studies to explore different mechanisms of the gemcitabine response in BxPC-3 cells affected by wild-type and mutant SMAD4 with DTLL induction, as described in the section of Results. “The result demonstrated that the mutant SMAD4 protein was degraded rapidly (Fig. 8A) but dramatically induced by DTLL (Fig. 8B) in BxPC3-Mut cells, whereas wild-type SMAD4 remained much longer and showed a slight increase via DTLL treatment in BxPC3-WT cells (Fig. 8A and 8B), consistent with the results from the AsPC-1 and MIA PaCa-2 lines (Fig. 4D and 4E).....SMAD4 genetic status is responsible for SMAD4 protein expression and that the difference in TRIM33 mediated proteasome degradation between mutant and wild-type proteins partially contributed to their distinguished protein levels. Moreover, DTLL significantly prolonged the half-life time by inhibiting SMAD4 degradation but impacted wild-type and mutant proteins to different degrees, thereby reactivating SMAD4 function with restored proteins, especially in BxPC3-Mut cells, consequently leading to enhanced cellular susceptibility to gemcitabine (Page 28-29, Line 745-765).” “In accordance with the results of TRIM33 alteration over the time course shown in Fig. 8B, the mRNA and protein levels of TRIM33 in BxPC3-Mut cells were decreased but not as significantly as those in BxPC3-WT cells after treatment with DTLL for 24 h (Fig. 8F and 8G). This result suggested that TRIM33 was not regulated by SMAD4, and seemed to be the feedback of either mutant or wild-type SMAD4 expression. We also demonstrated the interaction of TRIM33 with either mutant or wild-type SMAD4 proteins (Fig. 8H). The interaction between wild-type SMAD4 and TRIM33 was significantly inhibited in BxPC3-WT cells after treatment with DTLL, while their interaction was apparently increased in DTLL-treated BxPC3-Mut cells (Fig. 8I). Different levels of TRIM33 expression in both cell lines were decreased but to different degrees (Page 32, Line 843-853).”

There were different cell signaling pathways affected by mutant and wild-type SMAD4 given DTLL induction. “The results from co-IP assay further indicated that both mutant and wild-type SMAD4 proteins were able to interact with P50 and P65 (Fig. 8H), which implied that SMAD4 might impact on the transcriptional bioactivity of NF- κ B that further was responsible for regulating the expression of the above mRNAs and proteins in both BxPC3-WT and BxPC3-Mut cells. Moreover, the interactions of wild-type SMAD4 with P50 and P65 were significantly inhibited in BxPC3-WT cells after treated by DTLL but were apparently increased in DTLL-treated BxPC3-Mut cells (Fig. 8I). This might explain the reason for the difference in tendencies in the downstream gene expression affected by NF- κ B transcriptional activity between those two cell lines after treated by DTLL. Owing to the increased bioactivity of mutant SMAD4-mediated NF- κ B given DTLL, apoptotic BAX and FADD proteins were significantly induced and resulted in cell death (increased cleaved caspase-8). However, the transcriptional bioactivity of NF- κ B promoted by wild-type SMAD4 was inhibited by DTLL, and thus anti-apoptotic proteins (Bcl-2 and MCL1) were downregulated, mainly responsible for the increase in programmed cell death (enhanced cleaved caspase-8) (Page 31-32, Line 828-842).”

“In addition, we further performed a proteomic study to determine if SMAD4 has effects on the expression of gemcitabine-relevant transporters or pharmacokinetic enzymes (Supplementary Table 1A and 1B). As shown in the heatmap (left panel of Fig.

8J), proteomic profiles of BxPC3-Mut and -WT cells without any treatments were quite distinct and showed fewer changes after gemcitabine treated both lines, respectively. DTLL or its combination obviously altered the profile of BxPC3-Mut cells compared to the control or gemcitabine-induced profile, unlike BxPC3-WT cells.....Specifically, the expression of RRM2 was much lower in BxPC3-Mut than BxPC3-WT cells (Ratio Mut/WT = 0.22, $p = 2.24 \times 10^{-6}$), and significantly induced after treated with DTLL and its combination, with a similar altered trend of SMAD4 in those cells. The proteomic results implied that SMAD4 status significantly impacted on the proteomic profiles as a result of chemoresistance properties that DTLL might also affect the proteomic profiles of both lines, and that it altered cellular susceptibility via RRM2 induction regulated by SMAD4 in BxPC3-Mut cells (Page 32-33, Line 868-889).”

All the above suggested differences in action mechanism of DTLL overcoming gemcitabine resistance based on SMAD4 genetic or expression status, thereby to shed light on its therapeutic efficacy. As described in the section of Discussion (Page 37, Line 998-1013), “As shown in the schematic illustration model (Fig. 9), the mechanistic studies implied that DTLL combinational treatment might not only prevent from neoplastic proliferation via blockage of ATK/mTOR signaling and anti-apoptotic proteins (Bcl-2 and MCL1) mediated by impaired NF- κ B function if given to SMAD4-sufficient/gemcitabine-sensitive PDAC cells, but also restore the bioactivity of SMAD4 as a tumor suppressor to trigger its downstream NF- κ B-regulated signaling of cell apoptosis in SMAD4-deficient/gemcitabine-resistant tumors. The inhibitory effectiveness of DTLL alone or in combination with gemcitabine revealed that it functions as a double-edged sword in PDAC tumor growth, with totally different drug responsive behaviors and molecular action mechanisms on synergistic efficacy between gemcitabine-resistant and gemcitabine-sensitive PDAC. We speculate that DTLL seems to act as a novel module of the SMAD4 driver that normalizes its function as a tumor suppressor according to SMAD4 protein level and genetic status in PDAC, and then adjusts NF- κ B, TRIM33, gemcitabine-metabolic enzymes (such as RRM2) and proteomic profiles of PDAC cells, which explains the synergistic effect on tumor growth of DTLL in combination with gemcitabine.”

Therefore, we have demonstrated the function of SMAD4 in gemcitabine response, and different action mechanism of DTLL in mutant SMAD4/resistant and wild-type SMAD4/sensitive PDAC cells. SMAD4 is involved in the synergistic effect of gemcitabine and DTLL combination. DTLL showed an inverse effect on SMAD4 promoting its downstream targets in SMAD4-sufficient and SMAD4-deficient PDAC cells, including NF- κ B-mediated cell signaling (anti-apoptotic Bcl-2 and MCL1, or apoptotic BAX and FADD), gemcitabine-metabolic enzymes (such as RRM2), and interaction with TRIM33 to control its own protein degradation.

The quantification of cell proliferation in the manuscript is based on MTS assay. This assay is based on the cells metabolic activity and hence it may be confounding. An assay based on DNA quantity is required (CyQuant or similar).

Response to reviewer’s comment: Thank you so much for the comment. According to your

suggestion, we detected the drug effect on cell proliferation by using CyQUANT assay that's based on DNA quantity in addition of MTS assay, and modified the text in the sections of Methods and Results as the below.

“CyQUANT cell viability assay: In parallel to the above MTS assay, cell proliferation was confirmed using CyQUANT® NF cell proliferation assay kit (Thermo Fisher Scientific, Waltham, MA, USA) according to the manufacturer's protocol. This assay is based on the measurement of cellular DNA content via fluorescent dye binding. Cells were seeded in a 96-well plate (3,000 cells/well) and then exposed to a series of concentrations of GEM, DTLL, or GEM+ DTLL (10-14mol/L) for 72 h. Culture media was removed and 100 uL of the CyQUANT working solution was applied to the cells and incubated for 60 min at 37°C. The sample fluorescence was measured using a multimodal microplate reader (SpectraMax iD5, Molecular Devices) at 480 nm excitation and 520 nm emission. The assay was performed in triplicate. Cell viability was detected to create drug-response curves and CIs was calculated by the Chou-Talalay method [31, 32] (Page 8, Line 181-192).”

“To investigate whether the combination of two drugs could impact on PDAC cell proliferation, we treated gemcitabine-resistant/SMAD4-deficient AsPC-1 and gemcitabine-sensitive/SMAD4-sufficient MIA PaCa-2 cells with gemcitabine, DTLL and both. The results from both MTS (Fig. 2A) and CyQUANT (Fig. 2B) assays indicated that the combination of gemcitabine with DTLL obviously inhibited AsPC-1 cell growth compared to gemcitabine or DTLL alone. Compared with AsPC-1 cells, gemcitabine alone did show a stronger inhibitory effect in MIA PaCa-2 cells. The combination treatment revealed even more significant inhibition of cell growth in a concentration-dependent manner (Page 18, Line 436-444).” “Gemcitabine or DTLL alone, and their combination were tested for proliferative inhibition of BxPC3-EV, BxPC3-WT and BxPC3-Mut cells by using MTS and CyQUANT assays (Fig. 7E and Fig. 7F), respectively. The results indicated that the combination of gemcitabine with DTLL showed the most significant inhibition rate of tumor cell growth when compared to gemcitabine or DTLL alone, obviously showing a synergistic effect evaluated by CI calculation, not a simple additive effect (Page 27, Line 701-707).” The plots of the above results are shown in Fig. 2B and Fig. 7F, respectively.

In addition, it is unclear to this reviewer what are the concentration units used in Fig 1A, 2A etc. Particularly, given this is not the unit used in the text.

Perform Cyquant assay

Response to reviewer's comment: Thank you so much for the comment. Sorry for this confusion. The unit of drug concentration used in Fig 1A, 2A is mol/L with Log transformation based on 10, as well as that in other plots for *in vitro* cytotoxicity assays. We have changed all those labels as Log (mol/L).

Largely, the main hurdle throughout the rest of the figures is the lack of any causality of the findings. None of the proposed mechanisms is genetically or chemically inhibited. Thus, the reader is left with a series of associations between a potentially interesting drug and various

protein expression markers. To demonstrate that the proposed mechanisms are central to the observation's loss-of-function experiments are a must. The used of short hairpin RNA, CRISPR/cas9 or even ProTAG approaches is needed to unravel how this compound works and which cell signaling mechanisms are required to activate the synergy observed with gemcitabine.

Response to reviewer's comment: Thank you so much for the comment. According to your suggestion, we altered SMAD4 expression by transiently transfecting either specific siRNA or overexpression vector of SMAD4 into pancreatic cancer cells to detect the effect of SMAD4 expression level on PDAC cellular susceptibility to gemcitabine. Downregulation of SMAD4 in MIA PaCa-2 cells carrying wild-type SMAD4 gene resulted in high protein levels and thus significantly reduced cell sensitivity to gemcitabine (Fig. 4A). In contrast, SMAD4 overexpression apparently sensitized AsPC-1 cells to gemcitabine, due to the R100T mutation and little SMAD4 protein expression (Fig. 4B).

Furthermore, we evaluated the antitumor effects of gemcitabine on the *in vitro* growth of BxPC-3 cells with empty vector (BxPC3-EV), wild-type (BxPC3-WT) and mutant (BxPC3-Mut) overexpression vectors. Similar to the above results in Fig. 4, BxPC3-WT cells were more sensitive to gemcitabine than BxPC3 mock or BxPC3-EV cells, but BxPC3-Mut showed little difference (Fig. 7D). The combination treatment of gemcitabine with DTLL inhibited the cell growth of all three lines more obviously than gemcitabine or DTLL alone (Fig. 7E, 7F). Our *in vivo* observations by using those BxPC-3 CDX models further verified our hypothesis that SMAD4 is responsible for gemcitabine response and that DTLL in combination with gemcitabine improved PDAC chemotherapeutic efficacy (Fig. 7H and Table 3). Follow-up mechanistic research was conducted and demonstrated the function of SMAD4 in the gemcitabine response and the different action mechanisms of DTLL in mutant SMAD4/resistant and wild-type SMAD4/sensitive PDAC cells (Fig. 8).

Our findings indicated that SMAD4 was a predominant driver determining the cellular response to gemcitabine in PDAC cells. DTLL combinational treatment might not only prevent neoplastic proliferation via blockage of ATK/mTOR signaling and anti-apoptotic proteins mediated by impaired NF-kB function in SMAD4-sufficient/gemcitabine-sensitive PDAC cells, but also restore the bioactivity of SMAD4 as a tumor suppressor to trigger its downstream NF-kB-regulated signaling of cell apoptosis and enhance the gemcitabine-relative enzyme RRM2 in SMAD4-deficient/gemcitabine-resistant tumors. Our findings may unravel a potential combination therapy for overcoming resistance of PDAC mediated by SMAD4, which paves the way toward a long-sought tumor suppressor reactivation approach and lays the foundation for future precise medicine as the first step.

REVIEWERS' COMMENTS

Reviewer #1 (Remarks to the Author):

The paper has been significantly improved

Reviewer #2 (Remarks to the Author):

1. As far as the responses to my (Reviewer 2) comments are concerned, I am happy with the change in title.

2. However I am still uneasy about your plans for access for other researchers to your antibody so that your results are repeatable by other researchers - if your results cannot be independently replicated, your paper becomes of little value.

Of course it would be unreasonable to expect your lab to provide pure or labelled antibody to all comers. The solution would be to make good producer clones of your antibody easily available by depositing in a reputable cell bank. The cells to be available for a reasonable handling fee on condition that they are used only for research - commercial use only by agreement with your lab
Such a deposit is a common condition of patent application, and I note from your response that you are filing a patent

Ultimately this is for the journal editors to decide, but MY RECOMMENDATION IS THAT THESE CELLS MUST BE DEPOSITED AS DESCRIBED BEFORE THE PAPER CAN BE ACCEPTED AND THAT DETAILS MUST BE WRITTEN INTO THE PAPER.

3. In general I will not comment on the changes made in response to the other Reviewers' comments (although in general I support these comments) except for one area which has resulted in a lot of new material - that is the final proteomics section. I think this should be a separate paper, in a proteomics journal, with a longer and clearer description of the results .

Reviewer #3 (Remarks to the Author):

This is a revised version of the previously submitted manuscript. The manuscript has clearly improved including new models (including syngeneic) and an extended series of observations. As previously stated, the information provided is relevant but mechanistically weak. In this current version this has significantly improved with genetic validations that strengthen the authors claims. See below a few points.

As per my main point, the authors provide an extensive analysis of cells expressing SMAD4, in different mutant forms, as well as using siRNA against SMAD4, they go on to monitor some interesting aspects

such as synergistic response to DTLL in combination with gemcitabine and also other compound. Further the authors used different cell systems and an in vivo approach to confirm the observation.

Overall the manuscript has improved through the revision process and now reads more mechanistic than just a correlation of observations. The only sticky point is that now it reads a bit convoluted and the model (Figure 9) is difficult to digest. A more simplified version would be of help.

REVIEWERS' COMMENTS

Reviewer #1 (Remarks to the Author):

The paper has been significantly improved

Response to reviewer's comment: Thank you very much for the reviewer's comment.

Reviewer #2 (Remarks to the Author):

1. As far as the responses to my (Reviewer 2) comments are concerned, I am happy with the change in title.

Response to reviewer's comment: Thank you very much for the reviewer's comment.

2. However I am still uneasy about your plans for access for other researchers to your antibody so that your results are repeatable by other researchers - if your results cannot be independently replicated, your paper becomes of little value.

Of course it would be unreasonable to expect your lab to provide pure or labelled antibody to all comers. The solution would be to make good producer clones of your antibody easily available by depositing in a reputable cell bank. The cells to be available for a reasonable handling fee on condition that they are used only for research - commercial use only by agreement with your lab. Such a deposit is a common condition of patent application, and I note from your response that you are filing a patent.

Ultimately this is for the journal editors to decide, but MY RECOMMENDATION IS THAT THESE CELLS MUST BE DEPOSITED AS DESCRIBED BEFORE THE PAPER CAN BE ACCEPTED AND THAT DETAILS MUST BE WRITTEN INTO THE PAPER.

Response to reviewer's comment: Thank you very much for the reviewer's comment and suggestions. We have recently made good producer clones of our DTLP (the precursor of DTLL) fusion protein and deposited it into the China Pharmaceutical Culture Collection (CPCC) under Accession Number CCPC 101501 (<http://www.cpcc.ac.cn/>), as re-addressed in the text of the section of Methods (Page 31, Line 828-830). The strain (E. Coli. bacterial BE21 Star™ strain cells) has been assessed for viability and stored by using one of the standard methods in the CPCC, and is now available to the scientific research community, domestic or international. Upon request, CPCC shall distribute the strain without any special restriction.

3. In general I will not comment on the changes made in response to the other Reviewers' comments (although in general I support these comments) except for one area which has resulted in a lot of new material - that is the final proteomics section. I think this should be a separate paper, in a proteomics journal, with a longer and clearer description of the results.

Response to reviewer's comment: Thank you very much for the reviewer's comment and suggestions. We have kept our proteomic findings in the manuscript and deposited the data into a public repository - PRIDE. "All the proteomic raw data and the results files had been uploaded to the iProX Consortium (<https://www.iprox.org/>) with PXD identifiers (PXD031977) (Page 21, Line 578-580)." Accordingly, we have modified the data in the Supplementary Table 3 and 4 (the

previous Supplementary Table 1a and 1b), as shown in the Supplementary Information file.

With regarding to writing a separate paper, we appreciate the reviewer's constructive suggestion. Since the data we used in this manuscript is only one part of the entire proteomic data, we will explore those proteomic data to investigate other potential focuses with another study design for a separate paper in future.

Reviewer #3 (Remarks to the Author):

This is a revised version of the previously submitted manuscript. The manuscript has clearly improved including new models (including syngeneic) and an extended series of observations. As previously stated, the information provided is relevant but mechanistically weak. In this current version this has significantly improved with genetic validations that strengthen the authors claims. Se below a few points.

As per my main point, the authors provide an extensive analysis of cells expressing SMAD4, in different mutant forms, as well as using siRNA against SMAD4, they go on to monitor some interesting aspects such as synergistic response to DTLL in combination with gemcitabine an also other compound. Further the authors used different cell systems and an in vivo approach to confirm the observation.

Overall the manuscript has improved through the revision process and now reads more mechanistic than just a correlation of observations. The only sticky point is that now it reads a bit convoluted and the model (Figure 9) is difficult to digest. A more simplified version would be of help.

Response to reviewer's comment: Thank you very much for the reviewer's comment and suggestions. According to the reviewer's suggestion, we have selected the English language Editing Service (American Journal Experts), and modified the text as highlighted in a track change system to improve our manuscript explicitly. Moreover, we have re-made the model in Figure 9 that allows readers easy to understand, as shown in below.

Additional points:

1. In order to plot Figure 3a and Supplementary figure 4 with a two-segments Y-axis, we have re-analyzed for semiquantification of protein expression by using the GraphPad Prism 8.0 software, and remade these two figures.
2. We have corrected the calculative errors for all values of Tumor volume and Inhibition rate in Supplementary table 1 and 2 (the previous Table 2 and 3).
3. We have modified all the spelling or gramma errors and restated the manuscript to have our manuscript clarified.